# ResponseRank: Data-Efficient Reward Modeling through Preference Strength Learning

**Timo Kaufmann**
LMU Munich, MCML
timo.kaufmann@ifi.lmu.de

**Yannick Metz**
University of Konstanz
yannick.metz@uni-konstanz.de

**Daniel Keim**
University of Konstanz
keim@uni-konstanz.de

**Eyke Hüllermeier**
LMU Munich, MCML, DFKI
eyke@lmu.de

## Abstract

Binary choices, as often used for reinforcement learning from human feedback (RLHF), convey only the *direction* of a preference. A person may choose apples over oranges and bananas over grapes, but *which preference is stronger*? Strength is crucial for decision-making under uncertainty and generalization of preference models, but hard to measure reliably. Metadata such as response times and inter-annotator agreement can serve as proxies for strength, but are often noisy and confounded. We propose ResponseRank to address the challenge of learning from noisy strength signals. Our method uses relative differences in proxy signals to *rank responses to pairwise comparisons by their inferred preference strength*. To control for systemic variation, we compare signals only locally within carefully constructed strata. This enables robust learning of utility differences consistent with strength-derived rankings while making minimal assumptions about the strength signal. Our contributions are threefold: (1) ResponseRank, a novel method that robustly learns preference strength by leveraging locally valid relative strength signals; (2) empirical evidence of improved sample efficiency and robustness across diverse tasks: synthetic preference learning (with simulated response times), language modeling (with annotator agreement), and RL control tasks (with simulated episode returns); and (3) the *Pearson Distance Correlation (PDC)*, a novel metric that isolates cardinal utility learning from ordinal accuracy.

## 1 Introduction

Consider an AI system choosing between a risky action (usually good, occasionally catastrophic) and a safe one (consistently mediocre). Optimal decision-making in such uncertain scenarios requires knowing not just *which* outcome is preferred, but *by how much* – the preference strength. A slight preference for the good outcome over the mediocre one might favor safety, while a strong preference could justify the risk. Knowing preference strength is essential for optimal decision-making when outcomes are uncertain [1]. However, standard reinforcement learning from human feedback (RLHF) typically only collects binary preferences, identifying the preference *order* but not this crucial strength information – the *distance* between utilities.

In addition to the benefits for decision making under uncertainty, learning preference strength can significantly improve reward model generalization [2, 3], providing benefits even in deterministic tasks where the optimal policy only depends on the order of outcome returns. When assuming that preferences adhere to a latent utility function, utility differences provide additional structural information that aids learning this utility [4]. For example, if $A$ is strongly preferred to $C$ but only

39th Conference on Neural Information Processing Systems (NeurIPS 2025).

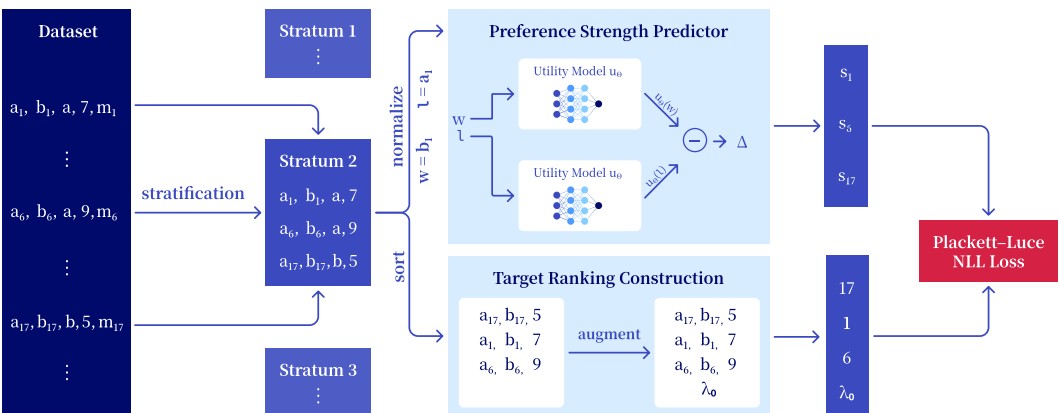

Figure 1: **ResponseRank learns strength-aware preferences from rankings over comparisons** derived from implicit strength signals. We start from a dataset of comparisons, consisting of a pair of objects $(a_i, b_i)$, a preference $(p_i \in \{a, b\})$, a strength signal $(\tau_i$, here scalar response times), and metadata $(m_i)$. (1) Comparisons are *stratified* using metadata $m_i$ (e.g., by annotator ID), ensuring local validity of the signal-strength relationship. The metadata can be discarded after this step. (2) Within each stratum, instances $(a_i, b_i, p_i, \tau_i)$ are *normalized* to $(w_i, l_i, \tau_i)$ format based on the preference label $p_i$, where $w_i$ is the preferred item and $l_i$ is the dispreferred item. (3) *Rank construction* then sorts these normalized tuples (e.g., by ascending response time $\tau_i$) and appends a virtual anchor element with a fixed score of 0 to create target rankings. (4) A preference predictor models these target rankings by predicting *signed utility differences* $s_\theta(w_i, l_i) = u_\theta(w_i) - u_\theta(l_i)$. (5) The $s_\theta$ values (and the anchor's 0) act as latent strengths for a Plackett–Luce model; NLL loss minimization yields parameters $\theta$ of the utility function (or *reward model*) $u_\theta$.

weakly preferred to $B$, we can infer $B \succ C$ without observing that comparison directly. Moreover, strength signals can provide an implicit regularization effect, emphasizing confident preferences and downweighting noisy ones.

Lacking an explicit strength signal, standard RLHF has to rely entirely on the model's generalization between comparisons to learn preference strength. Collecting such explicit signals requires additional annotation effort [5], however, and is susceptible to metacognitive challenges [6]. As an alternative to explicit collection, response metadata can often convey strength information. A motivating example of such implicit strength signals is *response time* (RT), the time taken by annotators to provide their feedback. Faster responses often indicate stronger preferences [7, 8]. While promising, signals such as response times can be difficult to learn from as they are noisy and confounded by factors such as individual annotator speed and task complexity [8, 9]. Consequently, the inverse RT-strength relationship is most reliably observed *locally* within homogeneous contexts, motivating methods that can robustly exploit such signals.

**Contributions.** (1) We introduce ResponseRank (Figure 1), a novel RLHF method that extracts preference strength from any locally-valid relative strength signal by leveraging the *relative order* of strength within homogeneous strata (e.g., per-annotator). This local approach can mitigate systemic noise through stratification and provides a proxy for preference strength designed to improve reward model generalization and decision-making under uncertainty. (2) We *empirically demonstrate* ResponseRank's improved accuracy and generalization in multiple domains. (3) We introduce the *Pearson Distance Correlation* (PDC), a novel metric designed to quantify how well a model captures preference strength.

## 2 Learning preference strength from local rankings

We introduce ResponseRank, a novel method for learning strength-aware preference models from rankings of comparisons. Our method builds on the principle that auxiliary signals can indicate preference strength but are often noisy and confounded. We learn utility differences by leveraging the relative order of these signals within homogeneous partitions (strata).

## 2.1 Problem setting and utility framework

We aim to learn a reward model from human feedback, incorporating strength signals to improve strength modeling and generalization. The input consists of a dataset $\mathcal{D} = \{(a_i, b_i, p_i, \tau_i, m_i)\}_{i=1}^{N}$. Each data point $i$ consists of two items $a_i, b_i \in \mathcal{X}$ being compared, a preference label $p_i$ indicating the preferred item, the strength signal $\tau_i$ (e.g., response time), and optional metadata $m_i$ (e.g., annotator ID, session timestamp).

Our approach is grounded in utility learning, a preference learning approach seeking to learn a *utility function* $u : \mathcal{X} \to \mathbb{R}$ that assigns a scalar value to each item, such that $x \succeq y \iff u(x) \geq u(y)$.[1] The magnitude of the utility difference, $u(x) - u(y)$, quantitatively represents the *preference strength*. In reinforcement learning from human feedback (RLHF), such learned utility functions often serve as reward models guiding policy optimization [10, 11]. The ResponseRank method, detailed next, learns a utility function $u_\theta$ by leveraging both the choice $p_i$ and the strength signal $\tau_i$ to infer not just which item is preferred, but by how much.

## 2.2 The ResponseRank method

ResponseRank starts by constructing strength-ranked comparisons within homogeneous strata (step 1 in Figure 1). It then learns preferences and strength jointly from these rankings using a joint ranking loss (steps 2–5 in Figure 1). We detail this process in the following subsections and discuss ResponseRank's relation to the conventional choice-only Bradley–Terry approach.

### 2.2.1 Constructing strength-ranked comparisons within homogeneous strata

**Stratification.** A key challenge is that strength signals like response times are often confounded globally but reliable locally. Consider response times across different annotators: baseline speed differences make global comparisons misleading, but stratifying by annotator isolates the within-annotator strength relationship. More generally, any factor believed to systematically influence the strength signal can serve as a stratification criterion, enabling flexible encoding of domain knowledge. Examples include task attributes such as text complexity clusters, labeling sessions, or other contextual features.

Within each stratum, the strength-signal relationship is more homogeneous. Rather than explicitly modeling the signal-generation process, we exploit this local validity by leveraging *ordinal* (relative) information *within* each stratum. This ordinal-only approach is inherently robust to outliers and signal magnitude variations, allowing any locally valid strength signal to serve as a proxy for preference strength – *only a monotonic relationship within each stratum is needed*.

**Partitioning and batch packing.** Once strata are formed, ResponseRank optionally applies additional partitioning within each stratum,[2] driven by practical constraints of the Plackett–Luce training objective: (1) Plackett–Luce requires rankings to be co-located within a training batch, necessitating careful packing and potential splitting of rankings into batches; (2) Strength signals may contain ties; partitioning can ensure tie-free rankings for cleaner training signals. (3) Smaller rankings sometimes improve robustness to noisy signals.

We detail algorithms for batch packing as well as size-constrained and tie-avoidant partitioning in Appendix B. Once stratification and partitioning are complete, the final step is sorting comparisons within each partition by the strength signal, yielding the target rankings used in the learning step.

### 2.2.2 Learning preferences and strength jointly from comparison rankings

To learn from the strength rankings constructed in the previous section, we need a model that captures both preference *direction* and *strength*. We adopt Plackett–Luce [12, 13], a probabilistic ranking model that connects latent scores (strengths here) to observed rankings.[3]

---

[1]We use the term utility function; in RLHF literature, this is commonly called the *reward function*.

[2]Globally valid signals may even *only* require partitioning and no data-driven stratification.

[3]Learning from such rankings of comparisons, a form of learning from *second-order preferences*, provides a strong signal for utility differences. Under reasonable assumptions, this is sufficient to identify the utility function up to a linear transformation [14]. A brief theoretical discussion can be found in Appendix F.4.

Table 1: **Target ranking of a single stratum** using illustrative comparisons with response times as the strength signal. Original comparisons ($q_i$) are first *normalized to (winner, loser) format* ($q_i'$), then sorted by *ascending response time $\tau_i$* (faster RTs indicate stronger preferences and are ranked higher). A *virtual anchor element* (latent score 0) is appended to this list for the Plackett–Luce model. The model learns positive utility differences $s_\theta(q') = u_\theta(\text{winner}) - u_\theta(\text{loser})$ ordinally consistent with this ranking (illustrative latent $s(q')$ shown).

| Comparison ($q_i$) | Preference ($p_i$) | Normalized ($q_i'$) | RT ($\tau_i$) | Exemplary $s(q')$ |
|---|---|---|---|---|
| $(a_1, b_1)$ | $a_1 \succ b_1$ | $(a_1, b_1)$ | 1.2s | 1.7 |
| $(a_2, b_2)$ | $a_2 \succ b_2$ | $(a_2, b_2)$ | 1.7s | 1.0 |
| $(a_3, b_3)$ | $b_3 \succ a_3$ | $(b_3, a_3)$ | 1.7s | 0.9 |
| $(a_4, b_4)$ | $b_4 \succ a_4$ | $(b_4, a_4)$ | 2.3s | 0.5 |
| *Virtual Anchor Element* | | | *N/A* | *0 (fixed)* |

The PL model defines the probability distribution of a ranking $\pi = (q_1, q_2, \ldots, q_k)$ of size $k$ as the product of sequentially choosing each comparison. The probability of $q_j$ being chosen next from the set of not-yet-ranked comparisons $\mathcal{C}^j$ is $P_{\text{PL}}(q_j \text{ from } \mathcal{C}^j \mid \theta) = \exp(s_\theta(q_j)) / \left( \sum_{q \in \mathcal{C}^j} \exp(s_\theta(q)) \right)$.

In our case, the latent utility values to be learned are preference strengths (by construction of the ranking). To capture *direction* alongside strength, we require the learned strengths $s_\theta(q)$ to be *signed*, with the sign indicating the preference direction. We achieve this by introducing a *virtual anchor element* with fixed score 0 and add it to each ranking. We normalize all comparisons to $q' = (w_i, l_i)$ format (preferred item first) and place the anchor at the end, resulting in all-positive target values. The Plackett–Luce log likelihood loss then naturally penalizes negative predictions, as they would make the target ranking, in which each comparison $q'$ is ranked above the 0-valued anchor by construction, less likely. Our goal is then to learn utility differences $s_\theta(w_i, l_i) = u_\theta(w_i) - u_\theta(l_i)$ that are positive (to match the normalized format) and ordinally consistent with the strength rankings.

Table 1 illustrates this process with a concrete example, showing how comparisons from a single partition are normalized, ranked by strength, and augmented with the virtual anchor. The rightmost column shows illustrative predicted $s_\theta(q')$ values that are ordinally consistent with the ranking.

We optimize $\theta$ via maximum likelihood estimation, maximizing the joint probability of the observed rankings under the PL model. Concretely, we model the utility function $u$ as a neural network $u_\theta$ that maps items to scalar utilities (or rewards in RLHF), giving rise to utility differences $s_\theta(a_i, b_i) = u_\theta(a_i) - u_\theta(b_i)$. During training, for each ranking, we predict $s_\theta(q')$ for all comparisons and fit $s_\theta$ to the target ranking via negative log-likelihood loss minimization.

### 2.2.3 Relationship to Bradley–Terry

A key property of ResponseRank, as formalized in Theorem 1, is that it gracefully reduces to the standard Bradley–Terry loss in the degenerate case of a single comparison per stratum (proof in Appendix G). This reduction places ResponseRank as a generalization of Bradley–Terry and allows for seamless mixing of strength-ranked and choice-only data: For any datapoint missing a strength signal, we can place it in a stratum of size 1, resulting in the standard Bradley–Terry loss for that comparison.

**Theorem 1** (ResponseRank reduces to BT for a single comparison)**.** *When applying the* ResponseRank *method to a stratum containing only a single normalized comparison $q' = (w, l)$, the target ranking for the Plackett-Luce model is $[q', \lambda_0]$, where $\lambda_0$ is the virtual anchor. Minimizing the NLL for this ranking, given the Plackett-Luce scores $s_\theta(q') = u_\theta(w) - u_\theta(l)$ and $s(\lambda_0) = 0$, is equivalent to minimizing the binary cross-entropy loss of a Bradley-Terry model for the preference $w \succ l$ with item utilities $u_\theta(w)$ and $u_\theta(l)$.*

Intuitively, for multiple comparisons per stratum, each comparison beating the anchor contributes a BT preference term, while the ordering among comparisons encourages stronger preferences to have larger utility differences. See Appendix A.4 for details.

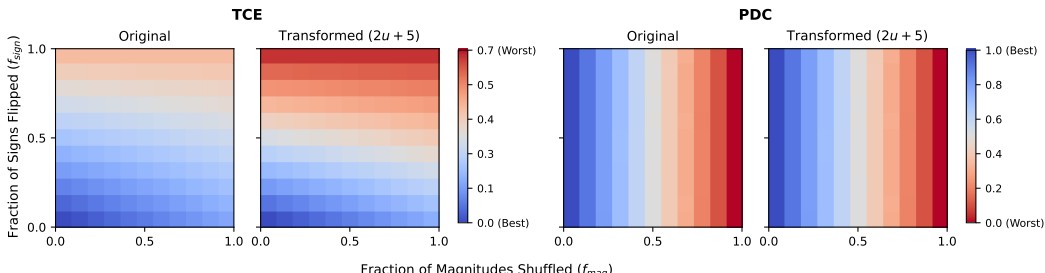

Figure 2: **PDC robustly isolates distance learning, unlike TCE.** Heatmaps compare PDC and TCE on synthetic data under varying levels of ordinal error ($f_{\text{sign}}$, y-axis, flipping signs) and distance information loss ($f_{\text{mag}}$, x-axis, shuffling magnitudes), for original and affine-scaled ($2u + 5$) utilities. PDC remains optimal (blue) when only ordinal information is degraded ($f_{\text{mag}} = 0$), decreases systematically as distance information is lost ($f_{\text{mag}} > 0$), and is consistent across utility scalings and ordinal accuracy, unlike TCE which conflates these aspects.

## 3 The Pearson Distance Correlation

To effectively quantify how well a model learns the *strength* or *distance* between utilities beyond just their ordinal ranking, we introduce the *Pearson Distance Correlation* (PDC). While some existing metrics assess aspects of preference strength, PDC is specifically designed to robustly isolate the learning of cardinal utility distances by satisfying key desirable properties. Further details, comparison with other metrics, and a discussion of these properties are provided in Appendix H.

The core idea of PDC, formalized in Definition 1, is to measure the linear correlation between the *absolute magnitudes* of true utility differences and predicted utility differences. This focus on absolute differences aims to decouple the assessment of distance learning from the correctness of ordinal preference, a property we term *ordinal independence*. Figure 2 visually contrasts PDC with calibration error (TCE, $l_1$ distance between predicted confidences and true likelihoods), illustrating PDC's robustness and key characteristics like ordinal independence and affine invariance.

**Definition 1** (Pearson Distance Correlation). *Let $X, X' \sim_{\text{iid}} p_{\text{data}}$ be independent items sampled from the data distribution $p_{\text{data}}$, and let $u, \hat{u} \colon \mathcal{X} \to \mathbb{R}$ denote the true and predicted utility functions, respectively. Define the true absolute utility difference as $\Delta U_{(X,X')} \coloneqq |u(X) - u(X')|$ and the predicted absolute utility difference as $\Delta \hat{U}_{(X,X')} \coloneqq |\hat{u}(X) - \hat{u}(X')|$. The PDC is the Pearson correlation coefficient between these random variables $\Delta U$ and $\Delta \hat{U}$:*

$$\rho_{\text{PDC}}(u, \hat{u}; p_{\text{data}}) = \text{Corr}(\Delta U, \Delta \hat{U}) = \frac{\text{Cov}(\Delta U, \Delta \hat{U})}{\sqrt{\mathbb{V}[\Delta U] \cdot \mathbb{V}[\Delta \hat{U}]}}. \tag{1}$$

*Variances and covariances are taken over the joint distribution of $(X, X')$ where $X, X' \sim_{iid} p_{\text{data}}$.*

PDC is designed to satisfy several crucial properties for evaluating learned utility distances, as stated in Proposition 1 (details and proof in Appendix H.3).

**Proposition 1** (PDC Properties). *Given a true utility function $u$ and a predicted utility function $\hat{u}$, the PDC satisfies: (i)* Affine Invariance*: Unchanged by positive affine utility transformations. (ii)* Distance Sensitivity*: Monotonically increases with accuracy of predicted difference magnitudes. (iii)* Known Baseline and Scale*: Values of 0 (no distance learning) and 1 (ideal distance learning). (iv)* Ordinal Independence*: Baseline for no distance learning is independent of ordinal accuracy.*

A PDC significantly greater than 0 implies non-trivial learning of utility difference magnitudes. This allows more precise statements about *if*, and *how well*, a model has learned relative preference strengths (see Appendix A for an example). PDC is designed to complement standard ordinal metrics, not replace them. Due to its ordinal independence, a high PDC can occur even with some ordinal mispredictions, provided the magnitudes of these (potentially erroneous) differences still correlate with true magnitudes. Note that real-world datasets generally do not have true utilities, making exact computation impossible. Appendix H.1 discusses this problem and proposes an approximation.

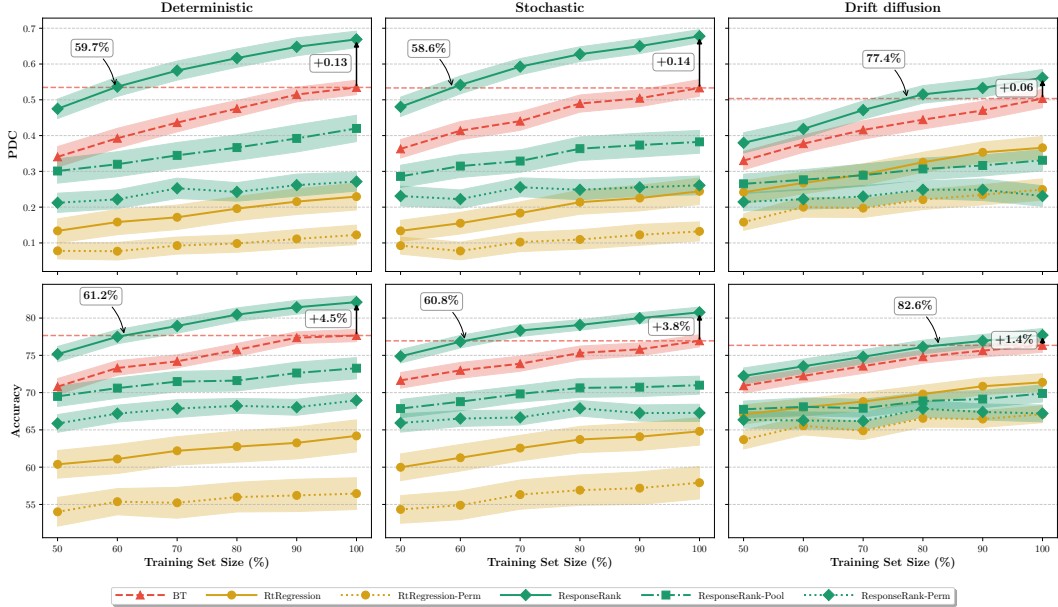

Figure 3: **Synthetic results with RT variability across strata.** Comparison of PDC (top) and accuracy (bottom) on Deterministic, Drift-Diffusion (DDM), and Stochastic datasets as a function of dataset size (100 runs; shading indicates 95% CI assuming normality). BT baseline performance with the full dataset is shown as a dashed red line; ResponseRank's interpolated breakeven point is marked by an arrow. Higher PDC indicates better learned preference strength (utility distances); higher accuracy indicates better ordinal preference predictions. **ResponseRank matches BT performance with only 61-83% of the data, demonstrating the efficiency of the method.** The permutation baseline (ResponseRank-Perm) confirms that performance improvements stem specifically from informative strength signals rather than modeling artifacts.

## 4 Synthetic preference learning experiments

In this section, we evaluate ResponseRank on synthetic pairwise comparison datasets with known ground-truth utility differences and simulated response times. We compare against baselines on metrics measuring preference strength learning (PDC) and ordinal accuracy under controlled conditions.

### 4.1 Experimental setup

**Datasets.** Synthetic datasets are generated by generating item pairs with random attributes, assigning utilities with a known function, and then deriving choices and strength signals in the form of synthetic RTs via a human choice model. To test robustness to mismatched assumptions, we consider two distinct human choice models: (1) a stochastic Bradley–Terry (BT) model extended with a custom link function mapping utility differences to response times, and (2) a joint drift-diffusion model (DDM) simultaneously generating stochastic choices and response times. We set model parameters to yield similar error rates (approximately 8%) in both scenarios, although the DDM exhibits higher noise in the response time generation. A 'Deterministic' setting uses model (1) without choice/RT noise. Data is partitioned into artificial strata (simulating, e.g., different annotators) and we apply a random RT multiplier to each stratum (simulating, e.g., annotator speed differences). The utility function is modeled as a small feed-forward neural network with a single hidden layer (64 neurons, ReLU activation) and optimized with AdamW. Full details and further results are in Appendix C. The source code for our experiments is available at `https://github.com/timokau/response-rank`.

**Baselines.** We compare ResponseRank against: (1) a *Bradley–Terry* (BT) baseline, the de-facto standard for learning utilities from pairwise preferences which considers only preference labels without response time information, (2) *RtRegression*: models RTs as inversely proportional to utility differences via a specific link function; this link function mirrors the one used in generating data for the deterministic and stochastic conditions, but does not use the stratum RT multipliers. (3) two

permutation control baselines (***ResponseRank-Perm***, ***RtRegression-Perm***), which train the respective methods on permuted response times to assess robustness and verify response-time informativeness, (4) ***ResponseRank-Pool***, an ablation that omits stratification in favor of a single global stratum. Detailed information on baselines and modeling assumptions is described in Appendix I.

## 4.2 Research questions and evaluation criteria

We investigate three specific research questions: (Q1) *Preference strength learning:* Does ResponseRank better recover underlying utility differences compared to baselines? (Q2) *Robustness across data generating processes:* Does ResponseRank maintain performance when trained with datasets generated from different assumptions? (Q3) *Choice prediction accuracy:* Does incorporating response-time information via ResponseRank improve ordinal preference prediction accuracy relative to baselines?

We quantitatively evaluate these aspects using two complementary metrics: (1) *Pearson Distance Correlation (PDC):* measuring correlation between predicted and true utility differences (Section 3). High PDC indicates superior recovery of cardinal utility distances. (2) *Choice Accuracy:* predicting held-out preference choices; higher accuracy represents a better ordinal utility model.

## 4.3 Results and discussion

Figure 3 shows the performance of ResponseRank and the BT baseline as a function of dataset size. Results demonstrate consistent improvements across most settings, indicating the robustness and effectiveness of the ResponseRank approach.

**Preference strength recovery (Q1).** ResponseRank achieves significantly higher median PDC than BT, demonstrating improved recovery of utility difference magnitudes. The ResponseRank-Perm control performs notably worse than ResponseRank, confirming improvements stem from informative RT signals, not just artifacts of the training procedure. Nonetheless ResponseRank-Perm still demonstrates reasonable performance, highlighting our method's robustness to noise in response times and bad stratification.

**Robustness to data-generating model assumptions (Q2).** ResponseRank consistently outperforms BT and permutation baselines across all data-generation paradigms (Deterministic, DDM, Stochastic) in median PDC and accuracy (Figure 8). This indicates ResponseRank robustly exploits relative RT signals even when data-generating mechanisms vary. The advantage is largest when the strength signal is cleanest (Deterministic) and decreases with signal noise (Stochastic, then DDM).

**Improving ordinal predictions via response time (Q3).** Incorporating RTs via ResponseRank significantly improves ordinal choice prediction accuracy across scenarios. This improvement, despite RTs not directly adding ordinal labels, highlights that learning better cardinal models through RTs enhances ordinal generalization. This suggests that ResponseRank could also benefit deterministic scenarios or domains less obviously requiring cardinal utilities, like language-model fine-tuning from pairwise human preferences, by learning more accurate and generalizable preference models.

## 4.4 Comparison to regression-based modeling and robustness analysis

In Figure 3, RtRegression underperforms both ResponseRank and the BT baseline. This is because RtRegression does not model inter-stratum variability: If there were no variability and the link between the strength signal and utility differences was *perfectly known*, RtRegression would be a viable alternative. Detailed results for this setting are available in Appendix C. ResponseRank-Pool similarly does not take this variability into account, but its performance remains slightly above the baseline. This indicates that the ResponseRank approach is more robust than RtRegression, likely because of its much weaker assumptions about the RT-strength relationship. This highlights the practical robustness of the proposed ranking-based approach, which avoids rigid link-function assumptions entirely.

Permuting RTs (ResponseRank-Perm) significantly degrades ResponseRank's performance, confirming RTs' role in the method. However, accuracy remains above chance and PDC remains above zero, indicating robustness to even extreme amounts of noise in the raw timing data. ResponseRank's effectiveness even on the drift diffusion dataset with noisy response times (Figure 3) demonstrates its robustness to noisy, yet informative, response time signals.

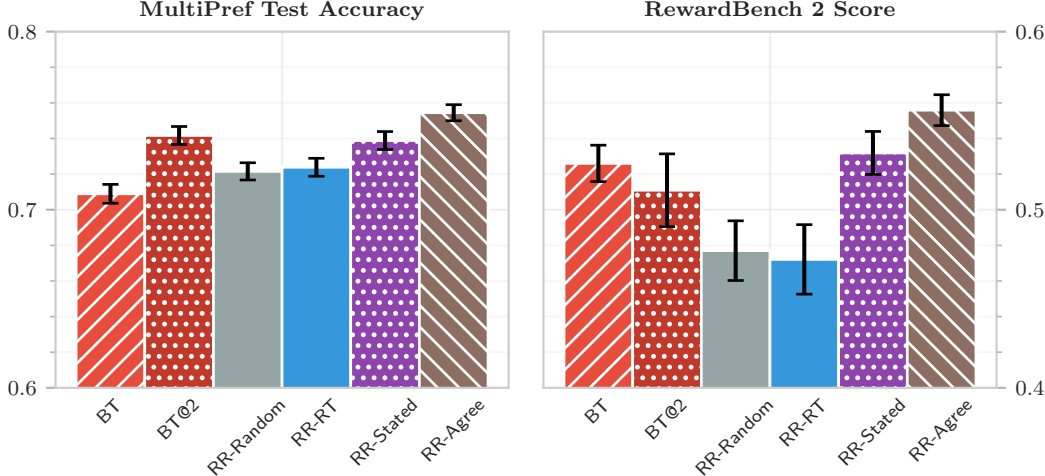

Figure 4: **Response time approach comparison.** We compare BT baseline (3 epochs), *BT@2* (2 epochs), *RR-Random* (*ablation*: using random strength rankings of size 2), *RR-RT* (using 8 length buckets with size-2 constraint), *RR-Stated*, and *RR-Agree* on both MultiPref test accuracy and RewardBench 2 performance. We report mean and 95% CI (assuming normality) across 30 seeds with distinct train/test splits.

Our experiments support ResponseRank's effectiveness and robustness. The empirical evaluation confirms considerable advantages over baselines in both cardinal and ordinal preference modeling, effective across diverse assumptions about human choice processes and providing tangible benefits even for purely ordinal prediction accuracy. These results indicate promising potential for broader applications, such as training language models, by improving model generalization through the incorporation of implicit cardinal information.

## 5    Language modeling experiments on MultiPref

To validate our method's applicability to real-world language modeling data, we conducted experiments on the MultiPref preference dataset [15]. This dataset provides text preferences with rich metadata. It spans 227 distinct crowdworkers, 5,323 unique prompts, 10,461 distinct responses, and four annotations per instance from workers of different expertise. It includes rich data on annotator expertise, preference strength (clear or slight), per-category preferences (helpful, truthful, harmless), and justifications for the preferences.

**Experimental setup.** We compare the *BT* baseline, not using any strength information, with several variants of ResponseRank using distinct strength signals: *RR-Random* ablates the impact of strength by using random strength rankings of size 2, *RR-RT* uses *response time* with stratification by annotator, *RR-Stated* uses binary *stated strength* of the preference (slight/clear) as given by the annotator with stratification by annotator identity, and *RR-Agree* uses the inter-annotator agreement of the 4 annotations for each comparison in the MultiPref dataset without stratification.

Our experimental design aims to validate that incorporating preference strength information improves reward model performance. In addition to the MultiPref binarized test set, we evaluate on the out-of-distribution RewardBench 2 dataset, which correlates strongly with downstream performance [16].

**Results.** Figure 4 shows that, given a relatively clean strength signal such as inter-annotator agreement (RR-Agree), ResponseRank improves both *in-distribution accuracy* and *out-of-distribution Reward-Bench performance* (+3.0 pp improvement over BT) compared to the BT baseline. Experiments with smaller subsets of the training data (Figure 10 in Appendix D) further show that RR-Agree matches final BT performance in RewardBench with approximately 30% *fewer preference pairs*, demonstrating its data efficiency.

RR-RT performs poorly, indicating that response time is not a useful signal of preference strength in this dataset. This may be because MultiPref's annotation protocol includes additional annotations (strength, categories, justifications) that extend response times and obscure the RT-strength relationship. Future work could investigate whether RT provides useful strength signals in datasets with simpler annotation protocols, or explore whether improved filtering and stratification could extract meaningful signals from MultiPref.

RR-Stated, which uses explicit strength annotations (slight vs. clear), outperforms RR-Random and RR-RT, confirming that ResponseRank benefits from informative signals. However, gains over BT are modest ($+0.6$ pp on RewardBench, not statistically significant, $p = 0.414$), likely because rankings require responses with different strength labels (limited by label imbalance) and self-reported strength annotations are inherently noisy.

**Training robustness.** We observe that training with the BT loss for two epochs is better for MultiPref performance while three epochs (our primary baseline) improve RewardBench performance. Analysis of RewardBench subcategories (Figure 11 in Appendix D) shows that the epoch 3 gains concentrate in the Focus category (measuring prompt adherence). We hypothesize that continued training simultaneously learns transferable skills (off-topic detection) useful for the RewardBench Focus category while also overfitting to other patterns – gains benefit Focus, but costs dominate on MultiPref test. ResponseRank variants avoid this tradeoff, demonstrating a regularizing effect of the ranking loss. Even RR-Random, despite fitting to entirely random strength signals, exceeds three-epoch BT performance on MultiPref test; this does not generalize to RewardBench, however, where informative strength signals are necessary.

# 6 RL control experiments

While our language modeling results demonstrate improved reward model generalization on out-of-distribution benchmarks, a remaining question is whether learned preference strength translates to other domains and improved downstream policy performance. To assess this, we conducted additional experiments in the control domain using simulated episode returns as strength signals. We train reward models on preference data generated using ground-truth environment rewards and use them to optimize policies via reinforcement learning, then evaluate the resulting policies on the actual task. This allows us to test whether capturing preference strength genuinely improves the policies that agents learn, rather than merely improving reward model statistics.

**Experimental setup and strength signal.** Following the methodology of Metz et al. [17], we tested three MuJoCo environments (HalfCheetah, Swimmer, Walker2d) and Highway merge-v0, comparing the BT baseline with ResponseRank using 5000 queries (4000 training/1000 validation) and averaging results over 5 runs per configuration. As return differences are globally valid, we used random partitions of size 16 to construct the rankings.

**Results.** Results detailed in Appendix E show that ResponseRank-reward models generally increase reward model accuracy, which translates to downstream policy improvements across environments. While this general trend is clear, the experiments in this domain are small scale and the differences often fall within the confidence interval. We encourage future work on applying and tuning ResponseRank to control domains at a larger scale.

# 7 Related work

Our work builds on rich literature in preference learning, auxiliary strength signals, and human feedback. In this section, we concisely highlight works most relevant to learning preference strength and the evaluation thereof. Appendix J expands on this and further discusses ranking-based utility learning, explicit strength specification, and adaptive losses. We give particular attention to response times as a strength signal in this discussion, as we consider them particularly promising due to their ready availability and their well-established relationship with preference strength.

**Learning from choices.** Learning latent utility functions from observed choices is a common approach in preference learning, widely used in RLHF [10, 18, 19]. Standard models like the Bradley-Terry model [20], when combined with proper scoring rules, can implicitly learn some aspects of utility differences, reflecting preference strength in choice probabilities. However, robustly

inferring preference strength solely from choices is challenging when comparisons are seen only once, as is typical in large-scale RLHF. This limitation motivates the use of additional signals, such as RTs, inter-annotator agreement, or categorical strength ratings, to capture preference strength more directly.

**Response time as a strength signal.** Response times as a readily available signal with an established relationship with preference strength are a key motivator for ResponseRank. Prior methods that integrate RTs often employ explicit cognitive process models like the Drift Diffusion Model (DDM) [e.g., 21, 2] or make strong assumptions about a direct, *global* relationship between RT magnitude and preference strength [e.g., 22].[4] ResponseRank differs fundamentally: it assumes signal-strength correlations (e.g., RT-strength) hold only *locally* within strata (e.g., per-annotator) and leverages only their monotonic *ordinal* ranking rather than modeling the full relationship. This local, relative-ranking approach enhances robustness against confounders, avoids the need for explicit DDM modeling or its associated complexities, and bypasses restrictive data requirements (e.g., repeated queries or fixed item sets [2, 22]), rendering it suitable for typical large-scale RLHF data.

**Evaluating Preference Strength.** While empirical evidence suggests that using signals such as RTs can improve overall model performance and generalization [3, 2, 21], robustly quantifying how well *preference strength* itself is learned remains an open challenge. Standard metrics like choice prediction accuracy [e.g., used by 2] or general calibration measures like the Brier score [e.g., 21] only indirectly reflect success in capturing strength, not specifically isolating learned utility *distances* or satisfying key invariances desirable for this particular evaluative task. To address this, we propose the Pearson Distance Correlation (PDC) (Section 3) as a novel metric tailored for evaluating learned preference strength by directly addressing utility distances and satisfying key invariances.

## 8 Discussion and conclusion

We introduced ResponseRank, a novel method to learn preference strength from locally valid relative signals of preference strength, enhancing RLHF without extra annotation.

**Limitations and Future Work.** In this work, we demonstrated ResponseRank's effectiveness across synthetic datasets, real-world language modeling, and RL control tasks. While response times proved challenging in MultiPref – likely due to the complex annotation protocol – inter-annotator agreement demonstrated strong performance, validating the approach with informative signals. Future work should explore datasets with simpler annotation protocols where RT may prove effective, improved noise-handling techniques (such as refined stratification) to extract weak strength signals, and alternative low-cost strength proxies. Our approach also discards tied preferences, though these could be incorporated as additional indicators of small preference differences. We validated ResponseRank-trained reward models on RewardBench (which correlates with downstream performance) and demonstrated improved policies in RL control tasks; validation in large-scale RLHF for language model policies remains important future work.

**Broader applications.** The core principle of ResponseRank – inferring signal strength from localized, relative rankings of noisy auxiliary data – extends beyond RTs in RLHF. It could improve human-AI alignment by capturing preference intensity from various implicit signals (e.g., hesitation) in applications like recommender systems. This approach could also generally enhance supervised learning by estimating label confidence or informativeness from contextual cues, aiding active learning or calibration.

**Conclusion.** ResponseRank robustly extracts cardinal preference information from noisy strength signals, and our proposed Pearson Distance Correlation (PDC) metric specifically quantifies this. Our experiments show ResponseRank significantly improves preference strength recovery (via PDC) and ordinal accuracy over choice-only baselines, indicating better sample efficiency and generalization. Key future work remains identifying cheaply available sources of strength signal or further investigating the use of response times. By enabling a more granular understanding of human preferences, ResponseRank contributes to developing AI systems that better align with the nuanced spectrum of human values, crucial for building more reliable and beneficial AI.

---

[4]Theoretical foundations of the RT–strength relationship and detailed comparison with prior methods using RTs are provided in Appendix F and Appendix J.

## Acknowledgments and Disclosure of Funding

The authors acknowledge support and computational resources from the Munich Center of Machine Learning (MCML). Additional computational resources were provided by several computing centers. For the CPU-based synthetic experiments, the authors gratefully acknowledge the computational and data resources provided by the Leibniz Supercomputing Centre (www.lrz.de). For the GPU-based experiments, the authors gratefully acknowledge the scientific support and resources of the AI service infrastructure LRZ AI Systems provided by the Leibniz Supercomputing Centre (LRZ) of the Bavarian Academy of Sciences and Humanities (BAdW), funded by Bayerisches Staatsministerium für Wissenschaft und Kunst (StMWK). The authors further acknowledge support by the state of Baden-Württemberg through *bwHPC* providing compute and GPU resources.

This work has also received funding from the European Union's Horizon Europe research and innovation programme under the Marie Sklodowska-Curie grant agreement No 101073307. The work at the University of Konstanz was partially funded by the Deutsche Forschungsgemeinschaft (DFG, German Research Foundation) – Project-ID 251654672 – TRR 161.

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

# Appendix

## A  Implicit learning of preference strength in standard choice models

Standard pairwise preference labels ($a \succ b$) used in reinforcement learning from human feedback (RLHF) primarily yield ordinal information (i.e., $u(a) \geq u(b)$). However, common RLHF methodologies often implicitly capture cardinal preference strength (the magnitude of $u(a) - u(b)$) through several interconnected mechanisms. This implicit capture of information is suggested by the empirical success of RLHF methods [e.g., 10]. If utility differences were learned entirely arbitrarily, preserving only preference order but not magnitude, these methods would likely struggle to learn effective and generalizable policies.

### A.1  Mechanisms of implicit strength learning

Several factors contribute to this implicit learning of preference strength. First, *probabilistic choice models* like the Bradley–Terry (BT) model [20] or Plackett–Luce models [12] inherently link choice probabilities to latent utility differences. In such models, for example the BT model, the probability $p(a \succ b)$ can be expressed as a function of the utility difference $u(a) - u(b)$, such as $\sigma(u(a) - u(b))$. This means that larger utility differences yield choice probabilities closer to 0 or 1, thereby encoding preference strength within the choice probability distribution itself. Second, training these models typically involves minimizing a *proper scoring rule*, such as cross-entropy loss [23]. This objective incentivizes the model to accurately predict the true choice probabilities. Since these probabilities are dependent on utility differences, the training process implicitly pushes the model to learn the underlying differences necessary to match the observed choice patterns [24]. Crucially, while the absolute utility values $u(x)$ from such models are typically identifiable[5] only up to an arbitrary additive shift, the resulting learned utility *differences* $u(a) - u(b)$ are shift-invariant and become meaningful representations of preference strength and choice probability [24].

Furthermore, *inductive biases* from the modeling approach assist in learning strength. The core assumption of a consistent utility function enforces transitivity. The use of smooth function approximators, such as neural networks, encourages generalization and interpolation of relative utility differences across similar item pairings. For example, if $u(a) > u(b)$ and $b \approx c$, smoothness in the utility approximator promotes learning that $u(a) > u(c)$. This capability is crucial, as proper scoring rules alone may not recover true probabilities when only ever observing unique training instances with hard, binary labels.

The Bradley–Terry model serves as a clear illustration of these principles in action. It is a popular model for pairwise comparisons, where the probability of preferring item $i$ over item $j$ is typically defined as:

$$p(i; u_i, u_j) = \frac{\exp(u_i)}{\exp(u_i) + \exp(u_j)}, \tag{2}$$

where $u_i$ and $u_j$ represent the utilities of items $i$ and $j$ respectively. This expression is akin to a pairwise logistic regression model, where the utility difference effectively serves as a feature. In many practical RLHF scenarios, we only observe a single outcome for each comparison (e.g., $i$ is chosen over $j$), meaning the training labels are degenerate ($p_i \in \{0, 1\}$). In principle, fitting a model to such degenerate labels without any regularization could theoretically push the learned absolute utility differences towards infinity, as the model continually seeks to improve the likelihood for the observed certain outcome. This issue parallels the challenge of fitting a classifier to perfectly separable hard labels, which can lead to overconfident and poorly calibrated models that assign extreme probabilities [25, 26]. In both preference learning and classification, the goal extends beyond mere accuracy in the primary prediction (the choice or the class label) to also obtaining meaningful secondary information, such as preference strength or a well-calibrated probability. The aforementioned inductive biases, particularly smoothness from neural network approximators, help to regularize the learning process and yield finite, meaningful utility differences.

---

[5]See Appendix A.3 for limitations of this identifiability in the RLHF setting.

## A.2 Empirical evidence and limitations

Our own experimental results provide empirical evidence for this implicit learning of distance by standard models. As demonstrated in Figure 3 in the main paper, the Bradley–Terry baseline consistently achieves a Pearson Distance Correlation (PDC) that is significantly and robustly above zero across various synthetic datasets and conditions. As established in Appendix H, a PDC value greater than zero robustly indicates that the model has learned non-trivial information about the true underlying utility differences.

Despite these mechanisms allowing for implicit strength capture, it is crucial to recognize their *limitations*. The preference strength information learned in this manner is indirect and can be weak, noisy, or highly contingent on the specific validity of model assumptions, the quality of the data, and the chosen architectural details. The inherent indirectness and potential unreliability of this implicitly learned strength motivate the development of methods like ResponseRank. ResponseRank aims to leverage more direct and potentially richer signals of preference strength, such as human response times, for more robust reward modeling. Indeed, while our results confirm that standard BT models do implicitly learn preference strength to some degree, these same results (Figure 3 in Appendix C) also clearly demonstrate that ResponseRank consistently achieves superior performance in capturing these cardinal utility differences, highlighting the benefits of its explicit, RT-informed approach.

## A.3 Identifiability

We briefly discuss identifiability of utilities in choice models and how ResponseRank compares to Bradley–Terry.

**Classical results.** In the Bradley–Terry model with a fixed finite set of items, the MLE of the item utilities is identifiable up to an additive constant under the condition that the comparison graph is strongly connected [27], and is consistent given sufficient data [28].

**ResponseRank and antisymmetry of strength.** ResponseRank builds on the Plackett–Luce model, a generalization of BT to rankings. Plackett–Luce item strengths are identifiable up to additive shift under similar conditions as BT [29]. For ResponseRank, we further constrain this model as we rank *comparisons* and prescribe that item strengths (each comparison being an item) take the form $s_{(a,b)} = u(a) - u(b)$. This introduces antisymmetry ($s_{(a,b)} = -s_{(b,a)}$), which prevents additive shift of the strength, making item strengths exactly identifiable. Strength implies utility up to shift (e.g., fixing any reference item $x$, we have $u(a) = s_{(a,x)} + u(x)$). Thus, ResponseRank has the same identifiability properties for item utilities and preference strength as BT. Note, however, that in the general case the identified utilities are not the same as those BT would identify from the pairwise comparisons alone, as ResponseRank uses more information to better estimate strength from finite data.

**Structured setting (RLHF).** The above identifiability results primarily consider the *unstructured* setting, where we estimate individual item utilities from comparisons among a fixed set of items. In RLHF, items are typically unique prompt-response pairs, and we estimate parameters of a utility function approximator instead of individual item utilities. This utility function makes use of *structured* items with feature vectors (e.g., embeddings of prompt-response pairs). This setting is called BT-Regression by Sun et al. [30]. Classical identifiability results still inform what can be learned: utilities only up to shift, and meaningful comparisons only between items or features connected by similar training comparisons. If the dataset contains only within-domain comparisons (e.g., poems vs. poems, code vs. code, but never across), cross-domain utility comparison is not grounded. ResponseRank, like BT, inherits these limitations, though the strength-based ranking provides additional signal that may help constrain learned utilities under limited data.

## A.4 Structural interpretation via rank-breaking

To build intuition for what the ResponseRank loss captures beyond the single-comparison case (Theorem 1), it is helpful to analyze its full *rank breaking*, the decomposition of each ranking into all implied pairwise comparisons. This analysis is meaningful because the quasi-MLE obtained by applying Bradley–Terry to a full rank-breaking targets the same parameters as the full PL MLE [31, Proposition 3.2].

The full breaking of a ResponseRank ranking consists of two types of pairs: (1) pairs between a comparison and the anchor element, and (2) pairs between two non-anchor comparisons. Each pair of type (1) has the same structure as the single-comparison case in Theorem 1 and therefore contributes a standard BT loss term for the corresponding comparison's preference direction. Pairs of type (2) further constrain the relative magnitudes of preference strengths within each stratum: if normalized comparison $q'_i$ is ranked above $q'_j$, the loss encourages $s_\theta(q'_i) > s_\theta(q'_j)$. Thus, ResponseRank can be viewed as Bradley–Terry on the preference directions, augmented with additional pairwise constraints on the order of strengths.

## B    Ranking construction details

While conceptually the primary means of constructing rankings is stratification, ensuring that the strength signal is comparable within each stratum, practical considerations often necessitate additional constraints: (1) Strata may have *tied strength signals* (e.g., two comparisons with identical response time from the same annotator), requiring partitioning into smaller rankings to avoid ties. (2) Strata may be large, while smaller rankings may be preferable in some cases (see Appendix D.6 for discussion). (3) Each ranking needs to be entirely co-located in a single on-device batch during training to compute the ranking loss, imposing memory constraints on ranking size. We address the first two considerations with a *tie-breaking partitioning strategy* within each stratum and the third with a *batch packing* algorithm.

### B.1    Tie-breaking partitioning strategy

The ranking loss requires preferences within each partition to have distinct strength values. Additionally, it is often desirable to split these strata further, as they may contain ties in strength values and smaller rankings may prove more robust to noisy signals (see Appendix D.6). Therefore, after stratification (if applicable), we further partition the data within each stratum into tie-free partitions with an optional maximum size $s$. Given a stratum of size $n$ we compute the number of partitions $k = \lceil n/s \rceil$ if a maximum size $s$ is specified or $k = m$, where $m$ is the size of the largest tie group, otherwise. We group preferences by strength value, then distribute each group across the $k$ partitions in a round-robin fashion. As long as $k$ is at least the size of the largest tie group, this guarantees that no two preferences within a partition have the same strength value. This maintains balanced partition sizes while maximizing the spread of strength values within each partition.

### B.2    Batch packing

The Plackett-Luce loss computation requires entire rankings to be co-located within single on-device training batches, as the probability calculation depends on the full set of comparisons within each stratum. Although the resulting batches are not i.i.d., as they contain comparisons from the same stratum (partially mediated by packing strata randomly into batches), this has precedent in techniques like length-based grouping for efficient GPU utilization [32] and has little performance impact in practice (see Figure 10 in Appendix D). The batch size additionally constrains maximum ranking length, which is further analyzed in Appendix D.5.

We designed a greedy 'largest first best fit' approach to pack the partitioned rankings into fixed-size batches for training:

- Initialize sufficient empty batches of fixed size to hold all comparisons. All batches will be filled to capacity, except possibly the last one.
- Process partition fragments in descending order of size (implemented using a max-heap). Shuffle each fragment's comparisons to randomize split points.
- Place the fragment in the batch with the least remaining space that can still accommodate it, if one exists. Otherwise place the largest prefix that fits and reinsert the remaining suffix into the heap.

While this leads to non-i.i.d. batches, we observe no adverse effects on training performance[6].

---

[6]Validated by comparing BT with random vs. RR-Agree batching, with RR-Agree batching showing a minor *improvement* in performance: MultiPref test +0.9 pp ($p < 0.001$), RewardBench 2 +0.4 pp ($p = 0.536$).

Table 2: Parameters for the synthetic data generation models.

| Model | Parameter | Value |
|---|---|---|
| Bradley-Terry & Log-Normal | Choice temperature | 0.2 |
| | Response time SD | 0.4 |
| | Minimum response time | 0 |
| | Maximum response time | 10 |
| | Partition RT variability | 3.0 |
| Drift-Diffusion | Decision threshold | 1.2 |
| | Non-decision time | 0.3 |
| | Drift rate multiplier | 1.0 |
| | Noise standard deviation | 0.4 |
| | Drift rate variability | 0.0 |
| | Starting point variability | 0.0 |
| | Non-decision time variability | 0.0 |
| | Time increment | 0.001 |
| | Number of partitions | 5 |
| | Partition RT variability | 3.0 |

## C   Supplemental details on synthetic experiments

In this appendix we first provide additional details on the synthetic experimental setup, including details on the synthetic dataset and training procedure. We then provide additional plots and numeric results for reproducibility.

### C.1   Detailed experimental setup

We use three synthetic datasets in our experiments: a deterministic dataset where the choice always favors the item with the higher utility, a stochastic dataset where the choice is made according to a Bradley–Terry model parameterized by the true utility, and a drift-diffusion dataset where choice and response times are generated jointly with a drift-diffusion model parameterized by the utility difference as the drift rate. The parameters for both models are shown in Table 2.

We generate a synthetic dataset of pairwise comparisons annotated with a choice label and a response time in two steps: query generation and labeling.

#### C.1.1   Query generation

This process generates pairs of objects for comparison. It can be further subdivided into *object generation* and *pairing*. We start by generating random objects, which are represented as vectors of 20 attributes each sampled from a uniform distribution over $[0, 1]$. The number of features and the dimensionality of the feature space can be adjusted to control the complexity of the dataset. We then generate queries from these objects by randomly sampling pairs of objects $(a_i, b_i)$.

#### C.1.2   Labeling

We start by generating a random utility function $u$, modelled as a neural network with random weights. The random utility function simulates a complex but learnable relationship between object attributes and utility and uses two hidden layers (one more than the reward predictor) with 64 neurons each and ReLU nonlinearity. It takes an object $x$ as input and outputs a scalar utility value $u(x)$ that is a function of the object's attributes and is guaranteed to be learnable by a neural network. The architecture of the utility function can be adjusted to control the complexity of the dataset. We then synthesize preference labels $p_i$ and response times $\tau_i$ based on the utility function. To test the robustness of ResponseRank to different preference and response time models, we consider two different models for generating the preference labels and response times, one generating preferences and response times independently, and one generating them jointly. We discuss the models in more detail below.

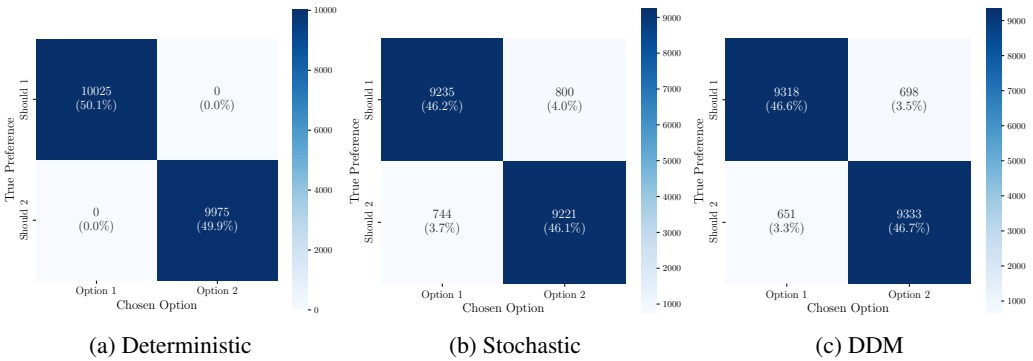

|   | (a) Deterministic | (b) Stochastic | (c) DDM |

Figure 5: Confusion matrices of the synthetic datasets.

**Independent labeling with Bradley–Terry and Log-Normal.** We generate preference labels $p_i$ based on the Bradley–Terry model and response times $\tau_i$ from a log-normal distribution with a hyperbolic link function. Concretely, we sample $p_i$ from a Bernoulli distribution with parameter $p = \sigma((u(a_i) - u(b_i))/\theta)$, where $\sigma$ is the sigmoid function and $\theta$ is the choice temperature. For response times, we first compute the mean using a hyperbolic link function (matching Equation (14) and repeated here for convenience)

$$\text{link}(\Delta u; \text{rt}_{\min}, \text{rt}_{\max}) := \text{rt}_{\min} + \frac{\text{rt}_{\max} - \text{rt}_{\min}}{|\Delta u| + 1}.$$

with $\text{rt}_{\min} = 0$ and $\text{rt}_{\max} = 10$. Then, we sample $\tau_i$ from a log-normal distribution with this mean and a fixed standard deviation parameter. The log-normal distribution ensures that response times are always positive while allowing for a right-skewed distribution commonly observed in human response time data. Note that in this model, the response time depends only on the absolute utility difference, so 'correct' and 'incorrect' responses are not differentiated in their response times.

**Partition-based Variability.** To better simulate real-world data collection scenarios, we introduce an additional source of systematic variability in response times through a partition-based approach. Each comparison is randomly assigned to one of 5 partitions, representing different data sources, annotators, or experimental conditions.

Within each partition, all response times are scaled by a partition-specific multiplier. These multipliers are sampled from a normal distribution with mean $1.0$ and standard deviation $3.0$, then converted to absolute values to ensure they remain positive. This creates systematic variation in response times between partitions while maintaining the same underlying preference patterns, simulating scenarios where different data sources might exhibit different response time characteristics.

**Joint labeling with Drift-Diffusion.** We jointly generate preference labels $p_i$ and response times $\tau_i$ based on the drift-diffusion model. We derive the drift rate from the utility difference, i.e., the choice experiment comparing $a_i$ and $b_i$ is modeled as a drift-diffusion process with drift rate $v = u(a_i) - u(b_i)$. Since all items are generated from the same distribution and passed through the same utility function, the drift rate has an expected value of $0$. Additionally, we normalize the standard deviation to $1$. All other parameters, such as the decision threshold, non-decision time, and noise standard deviation, are fixed to constant values as shown in Table 2. We assume symmetric decision thresholds, which intuitively control how cautious the decision-maker is. We chose these parameters to ensure a reasonable resulting joint distribution of response times and preference labels given the synthetic utilities and normalized drift rates.

These datasets can be parameterized, e.g., by re-scaling utilities and their differences, resulting in different levels and types of noise. The parameters were manually set to values resulting in reasonable noise levels, which are reported in this section.

### C.1.3 Dataset analysis

To characterize the synthetic datasets, we evaluated the noise levels affecting both ordinal preferences and response time signals (Figures 5 and 6). The confusion matrices (Figure 5) confirm the expected ordinal preference accuracy: perfect accuracy for the deterministic dataset, and approximately $8\%$

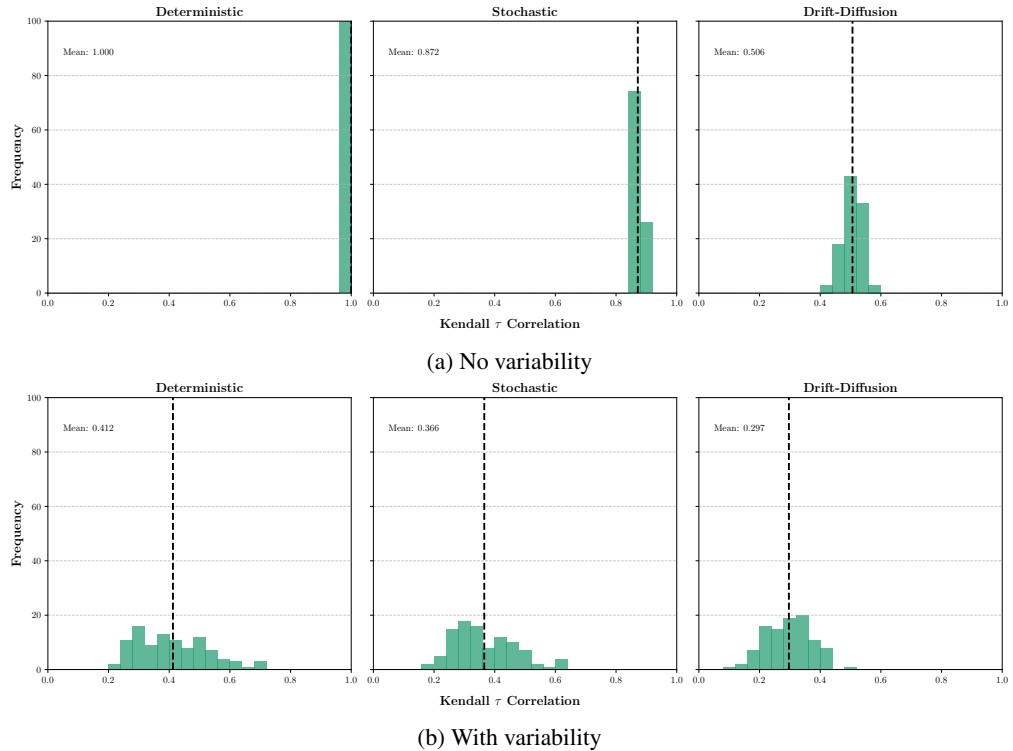

(a) No variability

(b) With variability

Figure 6: Histograms of the Kendall-tau correlation between the strength ranking implied by response times and the ground-truth strength ranking on synthetic datasets.

symmetric error rates for both the Stochastic and DDM datasets due to their probabilistic choice mechanisms.

Figure 6 assesses the fidelity of response time as an indicator of preference strength. It displays the distribution of Kendall's tau correlations (across 100 datasets) between the ranking derived from simulated RTs and the ranking derived from true utility differences. Kendall's tau is used here as it quantifies the correlation of ranks (the *order*), which is the specific information ResponseRank aims to leverage from RTs. The results highlight several points: (1) Under ideal (deterministic, no variability) conditions, the RT ranking perfectly matches the true strength ranking (by design). (2) Introducing inter-stratum variability weakens this correlation, demonstrating the challenge of using raw RTs globally, assuming such variability is present in real-world datasets. (3) The inherent noise in the Stochastic and DDM generation processes further reduces the correlation, with the Stochastic process preserving the rank order more reliably than the DDM. (4) Inter-stratum variability consistently degrades the RT-strength signal across dataset types.

### C.1.4 Assumptions

Our method makes minimal assumptions about human behavior. It requires only that (1) preferences approximately follow some utility function (standard Bradley-Terry assumption) and (2) we can approximately rank preference groups by strength. The only assumption about response times is an inverse monotonic relationship with preference strength. Our synthetic data, while far from perfectly realistic, reflects these properties: preferences generated from utility functions (extensively validated through BT model usage) and response times following inverse monotonic relationships (supported by psychology literature [7]). Going beyond this, one of our settings follows the drift-diffusion model, which has been extensively validated in psychological research [33], providing an approximation of response time behavior that goes beyond the monotonic relationship. We choose base utility functions randomly, which while not aligned with specific human preferences, is reasonable since the model should learn any utility function matching our two key assumptions.

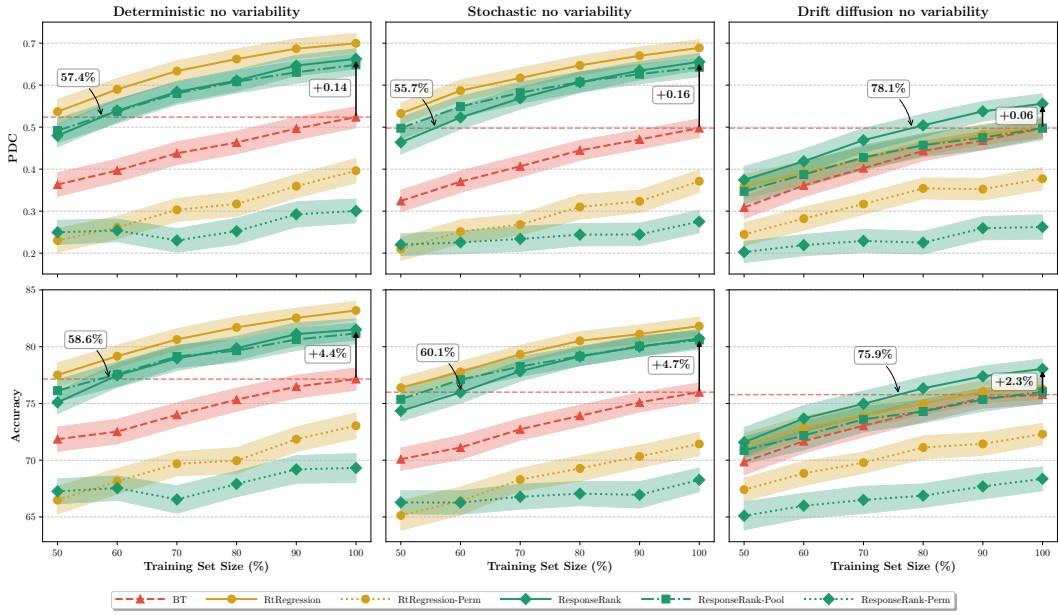

Figure 7: **Synthetic results without RT variability across strata.** Comparison of PDC (top) and accuracy (bottom) on Deterministic, Drift-Diffusion (DDM), and Stochastic datasets as a function of dataset size (100 runs; shading indicates $95\%$ CI assuming normality). Here we ablate the effect of inter-stratum variability by omitting stratum-RT multipliers. BT baseline performance with the full dataset is a dashed red line; ResponseRank's interpolated breakeven point is marked by an arrow. Higher PDC indicates better learned preference strength (utility distances); higher accuracy indicates better ordinal preference predictions.

## C.2 Training

We run 100 trials per condition. Each trial uses an independently generated dataset consisting of 50 training examples, 200 test examples, and 20 features per example. We optimize a small feed-forward neural network with a single hidden layer of size 64 and ReLU activation. Due to the small dataset size, we use full-batch gradient descent, processing all training examples in each update. Each model is trained for 200 gradient steps (AdamW optimizer, no early stopping, learning rate 0.001, weight decay 0.01). This avoids batch packing considerations described in Appendix B.2, which apply only to larger-scale settings. We optimize the standard cross-entropy loss for the BT baseline, MSE for RtRegression, and negative log-likelihood for ResponseRank. Each loss is normalized by dividing by the batch size (i.e., the full training set size).

## C.3 Additional results

Here we present additional experiments supporting the ones in the main body as well as additional detail and visualizations for the main experiments.

**Results without inter-stratum RT variability.** In the main experiments, we multiply response times in each stratum with a stratum-specific random multiplier to simulate effects such as individual annotator speeds. Here we ablate this choice by presenting results without this variability multiplier. This simulates a setting where all annotators display the exact same response time patterns. We include these results to demonstrate RtRegression's behavior under idealized conditions and the cost of unnecessary stratification on ResponseRank.

Figure 7 shows that, when inter-stratum RT variability is absent, stratification offers no benefit compared to an ResponseRank variant using a single global stratum (ResponseRank-Pool). This is expected, as strata have no difference in this setting and smaller rankings offer less grounding. Both variants continue to outperform the BT baseline.

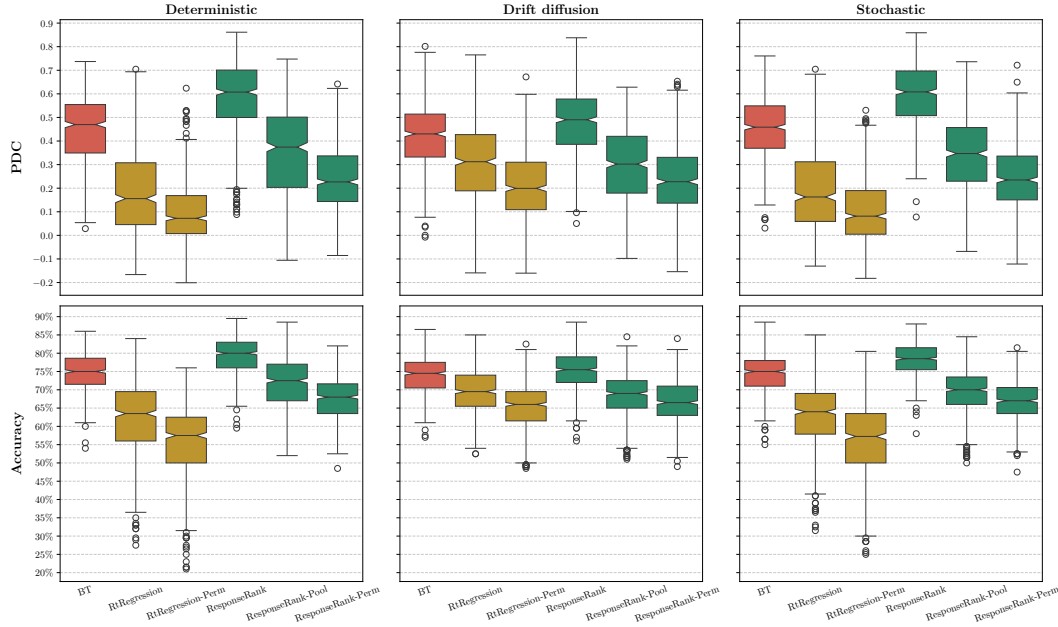

Figure 8: Performance comparison on synthetic datasets when **inter-stratum RT variability is introduced**. Boxplots show PDC (top) and accuracy (bottom) across 100 runs for deterministic, drift-diffusion, and stochastic data. This condition simulates diverse annotator speeds or task complexities affecting RTs across different strata. Higher PDC indicates better learned preference strength (utility distances); higher accuracy indicates better ordinal preference predictions.

Table 3: PDC performance across datasets with response time variability over 100 trials. Format: mean $\pm$ std [$\pm$ 95% CI half-width]. Best values per column are bolded.

| Dataset / Learner | 50% | 60% | 70% | 80% | 90% | 100% |
|---|---|---|---|---|---|---|
| **Deterministic** | | | | | | |
| BT | 0.34 ± 0.15 [± 0.03] | 0.39 ± 0.14 [± 0.03] | 0.44 ± 0.13 [± 0.03] | 0.48 ± 0.12 [± 0.02] | 0.51 ± 0.11 [± 0.02] | 0.53 ± 0.10 [± 0.02] |
| RtRegression | 0.13 ± 0.17 [± 0.03] | 0.16 ± 0.18 [± 0.04] | 0.17 ± 0.17 [± 0.03] | 0.20 ± 0.17 [± 0.03] | 0.22 ± 0.18 [± 0.04] | 0.23 ± 0.19 [± 0.04] |
| RtRegression-Perm | 0.08 ± 0.12 [± 0.02] | 0.08 ± 0.12 [± 0.02] | 0.09 ± 0.12 [± 0.02] | 0.10 ± 0.12 [± 0.02] | 0.11 ± 0.13 [± 0.03] | 0.12 ± 0.14 [± 0.03] |
| ResponseRank | **0.47** ± 0.14 [± 0.03] | **0.54** ± 0.14 [± 0.03] | **0.58** ± 0.14 [± 0.03] | **0.62** ± 0.13 [± 0.03] | **0.65** ± 0.13 [± 0.03] | **0.67** ± 0.12 [± 0.02] |
| ResponseRank-Pool | 0.30 ± 0.17 [± 0.03] | 0.32 ± 0.18 [± 0.04] | 0.34 ± 0.18 [± 0.04] | 0.37 ± 0.18 [± 0.04] | 0.39 ± 0.19 [± 0.04] | 0.42 ± 0.19 [± 0.04] |
| ResponseRank-Perm | 0.21 ± 0.14 [± 0.03] | 0.22 ± 0.13 [± 0.03] | 0.25 ± 0.15 [± 0.03] | 0.24 ± 0.13 [± 0.03] | 0.26 ± 0.16 [± 0.03] | 0.27 ± 0.14 [± 0.03] |
| **Drift diffusion** | | | | | | |
| BT | 0.33 ± 0.13 [± 0.03] | 0.38 ± 0.12 [± 0.02] | 0.42 ± 0.13 [± 0.03] | 0.44 ± 0.13 [± 0.03] | 0.47 ± 0.13 [± 0.03] | 0.50 ± 0.13 [± 0.03] |
| RtRegression | 0.24 ± 0.16 [± 0.03] | 0.27 ± 0.15 [± 0.03] | 0.29 ± 0.15 [± 0.03] | 0.33 ± 0.14 [± 0.03] | 0.35 ± 0.15 [± 0.03] | 0.37 ± 0.16 [± 0.03] |
| RtRegression-Perm | 0.16 ± 0.12 [± 0.02] | 0.20 ± 0.14 [± 0.03] | 0.20 ± 0.13 [± 0.03] | 0.22 ± 0.14 [± 0.03] | 0.23 ± 0.14 [± 0.03] | 0.25 ± 0.15 [± 0.03] |
| ResponseRank | **0.38** ± 0.14 [± 0.03] | **0.42** ± 0.13 [± 0.03] | **0.47** ± 0.13 [± 0.03] | **0.52** ± 0.12 [± 0.02] | **0.53** ± 0.13 [± 0.03] | **0.56** ± 0.12 [± 0.02] |
| ResponseRank-Pool | 0.27 ± 0.14 [± 0.03] | 0.28 ± 0.14 [± 0.03] | 0.29 ± 0.16 [± 0.03] | 0.31 ± 0.16 [± 0.03] | 0.32 ± 0.15 [± 0.03] | 0.33 ± 0.15 [± 0.03] |
| ResponseRank-Perm | 0.21 ± 0.13 [± 0.03] | 0.22 ± 0.14 [± 0.03] | 0.23 ± 0.16 [± 0.03] | 0.25 ± 0.13 [± 0.02] | 0.25 ± 0.15 [± 0.03] | 0.23 ± 0.15 [± 0.03] |
| **Stochastic** | | | | | | |
| BT | 0.36 ± 0.13 [± 0.03] | 0.41 ± 0.13 [± 0.03] | 0.44 ± 0.13 [± 0.03] | 0.49 ± 0.12 [± 0.02] | 0.50 ± 0.12 [± 0.02] | 0.53 ± 0.12 [± 0.02] |
| RtRegression | 0.13 ± 0.15 [± 0.03] | 0.16 ± 0.15 [± 0.03] | 0.18 ± 0.16 [± 0.03] | 0.21 ± 0.17 [± 0.03] | 0.23 ± 0.18 [± 0.04] | 0.24 ± 0.19 [± 0.04] |
| RtRegression-Perm | 0.09 ± 0.12 [± 0.02] | 0.08 ± 0.13 [± 0.02] | 0.10 ± 0.14 [± 0.03] | 0.11 ± 0.14 [± 0.03] | 0.12 ± 0.15 [± 0.03] | 0.13 ± 0.14 [± 0.03] |
| ResponseRank | **0.48** ± 0.14 [± 0.03] | **0.54** ± 0.13 [± 0.03] | **0.59** ± 0.12 [± 0.02] | **0.63** ± 0.10 [± 0.02] | **0.65** ± 0.11 [± 0.02] | **0.68** ± 0.09 [± 0.02] |
| ResponseRank-Pool | 0.29 ± 0.16 [± 0.03] | 0.31 ± 0.16 [± 0.03] | 0.33 ± 0.16 [± 0.03] | 0.36 ± 0.17 [± 0.03] | 0.37 ± 0.17 [± 0.03] | 0.38 ± 0.16 [± 0.03] |
| ResponseRank-Perm | 0.23 ± 0.14 [± 0.03] | 0.22 ± 0.13 [± 0.03] | 0.26 ± 0.13 [± 0.03] | 0.25 ± 0.14 [± 0.03] | 0.26 ± 0.13 [± 0.03] | 0.26 ± 0.13 [± 0.03] |

Table 4: Accuracy performance across datasets with response time variability over 100 trials. Format: mean ± std [± 95% CI half-width]. Best values per column are bolded.

| Dataset / Learner | 50% | 60% | 70% | 80% | 90% | 100% |
|---|---|---|---|---|---|---|
| **Deterministic** | | | | | | |
| BT | 70.8 ± 5.5% [± 1.1] | 73.3 ± 5.0% [± 1.0] | 74.2 ± 4.4% [± 0.9] | 75.7 ± 4.4% [± 0.9] | 77.4 ± 4.0% [± 0.8] | 77.7 ± 4.2% [± 0.8] |
| RtRegression | 60.4 ± 9.5% [± 1.9] | 61.1 ± 9.9% [± 2.0] | 62.2 ± 9.7% [± 1.9] | 62.8 ± 10.6% [± 2.1] | 63.3 ± 11.0% [± 2.2] | 64.2 ± 11.0% [± 2.2] |
| RtRegression-Perm | 54.0 ± 9.8% [± 2.0] | 55.4 ± 9.0% [± 1.8] | 55.2 ± 10.6% [± 2.1] | 56.0 ± 10.1% [± 2.0] | 56.2 ± 11.0% [± 2.2] | 56.5 ± 10.9% [± 2.2] |
| ResponseRank | **75.2** ± 5.3% [± 1.0] | **77.5** ± 4.8% [± 1.0] | **78.9** ± 5.0% [± 1.0] | **80.5** ± 4.6% [± 0.9] | **81.5** ± 4.7% [± 0.9] | **82.1** ± 4.3% [± 0.9] |
| ResponseRank-Pool | 69.5 ± 6.9% [± 1.4] | 70.6 ± 7.0% [± 1.4] | 71.5 ± 7.0% [± 1.4] | 71.6 ± 7.1% [± 1.4] | 72.6 ± 7.4% [± 1.5] | 73.3 ± 7.3% [± 1.4] |
| ResponseRank-Perm | 65.9 ± 6.1% [± 1.2] | 67.2 ± 5.8% [± 1.1] | 67.9 ± 6.1% [± 1.2] | 68.2 ± 5.0% [± 1.0] | 68.0 ± 5.5% [± 1.1] | 69.0 ± 5.4% [± 1.1] |
| **Drift diffusion** | | | | | | |
| BT | 70.9 ± 5.0% [± 1.0] | 72.2 ± 4.6% [± 0.9] | 73.6 ± 5.0% [± 1.0] | 74.8 ± 4.4% [± 0.9] | 75.6 ± 4.4% [± 0.9] | 76.3 ± 4.7% [± 0.9] |
| RtRegression | 67.1 ± 6.6% [± 1.3] | 68.0 ± 5.8% [± 1.2] | 68.8 ± 6.1% [± 1.2] | 69.8 ± 6.0% [± 1.2] | 70.9 ± 5.8% [± 1.1] | 71.4 ± 5.9% [± 1.2] |
| RtRegression-Perm | 63.7 ± 6.4% [± 1.3] | 65.5 ± 6.2% [± 1.2] | 64.9 ± 6.0% [± 1.2] | 66.5 ± 6.1% [± 1.2] | 66.5 ± 5.7% [± 1.1] | 67.1 ± 5.8% [± 1.2] |
| ResponseRank | **72.2** ± 5.5% [± 1.1] | **73.5** ± 4.9% [± 1.0] | **74.8** ± 4.8% [± 1.0] | **76.1** ± 4.7% [± 0.9] | **76.9** ± 4.3% [± 0.9] | **77.7** ± 4.5% [± 0.9] |
| ResponseRank-Pool | 67.8 ± 5.6% [± 1.1] | 68.1 ± 6.1% [± 1.2] | 67.9 ± 6.1% [± 1.2] | 68.9 ± 5.8% [± 1.1] | 69.1 ± 5.9% [± 1.2] | 69.9 ± 5.8% [± 1.2] |
| ResponseRank-Perm | 66.3 ± 5.6% [± 1.1] | 66.3 ± 5.8% [± 1.2] | 66.2 ± 5.9% [± 1.2] | 67.8 ± 5.7% [± 1.1] | 67.4 ± 5.5% [± 1.1] | 67.2 ± 5.8% [± 1.2] |
| **Stochastic** | | | | | | |
| BT | 71.6 ± 5.4% [± 1.1] | 73.0 ± 5.4% [± 1.1] | 73.9 ± 5.2% [± 1.0] | 75.3 ± 4.7% [± 0.9] | 75.8 ± 4.4% [± 0.9] | 76.9 ± 4.2% [± 0.8] |
| RtRegression | 60.0 ± 9.1% [± 1.8] | 61.3 ± 9.1% [± 1.8] | 62.6 ± 8.7% [± 1.7] | 63.7 ± 9.0% [± 1.8] | 64.1 ± 9.4% [± 1.9] | 64.8 ± 9.4% [± 1.9] |
| RtRegression-Perm | 54.3 ± 9.5% [± 1.9] | 54.9 ± 9.8% [± 1.9] | 56.3 ± 10.0% [± 2.0] | 56.9 ± 10.4% [± 2.1] | 57.2 ± 11.2% [± 2.2] | 57.9 ± 11.1% [± 2.2] |
| ResponseRank | **74.9** ± 4.6% [± 0.9] | **76.8** ± 4.3% [± 0.8] | **78.3** ± 4.0% [± 0.8] | **79.1** ± 3.6% [± 0.7] | **80.0** ± 3.7% [± 0.7] | **80.8** ± 3.6% [± 0.7] |
| ResponseRank-Pool | 67.9 ± 6.2% [± 1.2] | 68.8 ± 6.2% [± 1.2] | 69.8 ± 5.6% [± 1.1] | 70.6 ± 6.5% [± 1.3] | 70.7 ± 6.4% [± 1.3] | 71.0 ± 6.2% [± 1.2] |
| ResponseRank-Perm | 65.9 ± 6.3% [± 1.2] | 66.5 ± 5.4% [± 1.1] | 66.7 ± 5.1% [± 1.0] | 67.9 ± 5.3% [± 1.0] | 67.3 ± 5.5% [± 1.1] | 67.3 ± 5.9% [± 1.2] |

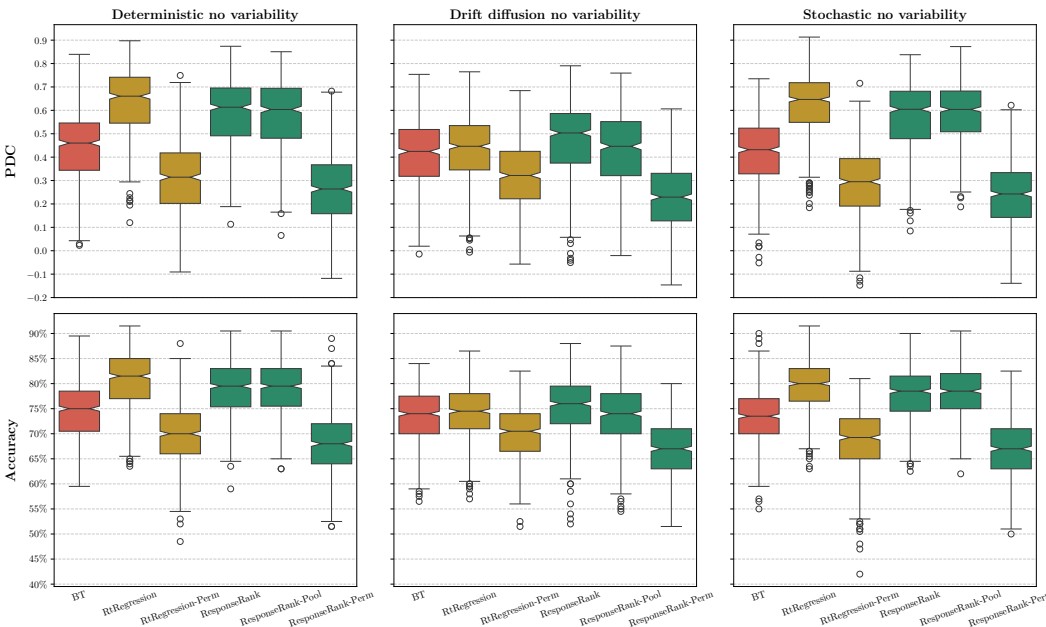

Figure 9: Performance comparison on synthetic datasets when **inter-stratum RT variability is absent**. Boxplots show PDC (top) and accuracy (bottom) across 100 runs for deterministic, drift-diffusion, and stochastic data. In this condition, all strata have identical RT patterns (no inter-stratum variability). Higher PDC indicates better learned preference strength (utility distances); higher accuracy indicates better ordinal preference predictions.

Table 5: PDC performance across datasets without response time variability. Values are reported as mean $\pm$ standard deviation over 100 trials.

| Dataset / Learner | 50% | 60% | 70% | 80% | 90% | 100% |
|---|---|---|---|---|---|---|
| **Deterministic no variability** | | | | | | |
| BT | 0.36 ± 0.14 [± 0.03] | 0.40 ± 0.14 [± 0.03] | 0.44 ± 0.14 [± 0.03] | 0.46 ± 0.14 [± 0.03] | 0.50 ± 0.14 [± 0.03] | 0.52 ± 0.13 [± 0.03] |
| RtRegression | **0.54** ± 0.15 [± 0.03] | **0.59** ± 0.13 [± 0.03] | **0.63** ± 0.13 [± 0.03] | **0.66** ± 0.12 [± 0.02] | **0.69** ± 0.12 [± 0.02] | **0.70** ± 0.12 [± 0.02] |
| RtRegression-Perm | 0.23 ± 0.15 [± 0.03] | 0.26 ± 0.14 [± 0.03] | 0.30 ± 0.14 [± 0.03] | 0.32 ± 0.15 [± 0.03] | 0.36 ± 0.14 [± 0.03] | 0.40 ± 0.15 [± 0.03] |
| ResponseRank | 0.48 ± 0.14 [± 0.03] | 0.54 ± 0.14 [± 0.03] | 0.58 ± 0.13 [± 0.03] | 0.61 ± 0.13 [± 0.03] | 0.65 ± 0.12 [± 0.02] | 0.66 ± 0.12 [± 0.02] |
| ResponseRank-Pool | 0.49 ± 0.15 [± 0.03] | 0.54 ± 0.15 [± 0.03] | 0.58 ± 0.14 [± 0.03] | 0.61 ± 0.14 [± 0.03] | 0.63 ± 0.13 [± 0.03] | 0.65 ± 0.13 [± 0.03] |
| ResponseRank-Perm | 0.25 ± 0.15 [± 0.03] | 0.25 ± 0.14 [± 0.03] | 0.23 ± 0.14 [± 0.03] | 0.25 ± 0.16 [± 0.03] | 0.29 ± 0.15 [± 0.03] | 0.30 ± 0.14 [± 0.03] |
| **Drift diffusion no variability** | | | | | | |
| BT | 0.31 ± 0.14 [± 0.03] | 0.36 ± 0.14 [± 0.03] | 0.40 ± 0.13 [± 0.03] | 0.44 ± 0.12 [± 0.02] | 0.47 ± 0.11 [± 0.02] | 0.50 ± 0.12 [± 0.02] |
| RtRegression | 0.36 ± 0.14 [± 0.03] | 0.39 ± 0.14 [± 0.03] | 0.42 ± 0.14 [± 0.03] | 0.46 ± 0.12 [± 0.02] | 0.48 ± 0.12 [± 0.02] | 0.50 ± 0.11 [± 0.02] |
| RtRegression-Perm | 0.24 ± 0.14 [± 0.03] | 0.28 ± 0.14 [± 0.03] | 0.32 ± 0.13 [± 0.03] | 0.35 ± 0.13 [± 0.03] | 0.35 ± 0.13 [± 0.03] | 0.38 ± 0.14 [± 0.03] |
| ResponseRank | **0.37** ± 0.16 [± 0.03] | **0.42** ± 0.14 [± 0.03] | **0.47** ± 0.15 [± 0.03] | **0.50** ± 0.14 [± 0.03] | **0.54** ± 0.12 [± 0.02] | **0.56** ± 0.12 [± 0.02] |
| ResponseRank-Pool | 0.35 ± 0.15 [± 0.03] | 0.39 ± 0.15 [± 0.03] | 0.43 ± 0.15 [± 0.03] | 0.46 ± 0.14 [± 0.03] | 0.48 ± 0.15 [± 0.03] | 0.50 ± 0.14 [± 0.03] |
| ResponseRank-Perm | 0.20 ± 0.13 [± 0.03] | 0.22 ± 0.13 [± 0.03] | 0.23 ± 0.14 [± 0.03] | 0.22 ± 0.14 [± 0.03] | 0.26 ± 0.14 [± 0.03] | 0.26 ± 0.15 [± 0.03] |
| **Stochastic no variability** | | | | | | |
| BT | 0.32 ± 0.13 [± 0.03] | 0.37 ± 0.13 [± 0.03] | 0.41 ± 0.13 [± 0.03] | 0.44 ± 0.12 [± 0.02] | 0.47 ± 0.12 [± 0.02] | 0.50 ± 0.11 [± 0.02] |
| RtRegression | **0.53** ± 0.13 [± 0.03] | **0.59** ± 0.13 [± 0.03] | **0.62** ± 0.12 [± 0.02] | **0.65** ± 0.12 [± 0.02] | **0.67** ± 0.11 [± 0.02] | **0.69** ± 0.10 [± 0.02] |
| RtRegression-Perm | 0.21 ± 0.14 [± 0.03] | 0.25 ± 0.14 [± 0.03] | 0.27 ± 0.13 [± 0.03] | 0.31 ± 0.15 [± 0.03] | 0.32 ± 0.14 [± 0.03] | 0.37 ± 0.14 [± 0.03] |
| ResponseRank | 0.46 ± 0.14 [± 0.03] | 0.52 ± 0.14 [± 0.03] | 0.57 ± 0.13 [± 0.03] | 0.61 ± 0.12 [± 0.02] | 0.63 ± 0.11 [± 0.02] | 0.66 ± 0.10 [± 0.02] |
| ResponseRank-Pool | 0.50 ± 0.13 [± 0.03] | 0.55 ± 0.13 [± 0.03] | 0.58 ± 0.13 [± 0.03] | 0.61 ± 0.12 [± 0.02] | 0.63 ± 0.12 [± 0.02] | 0.64 ± 0.12 [± 0.02] |
| ResponseRank-Perm | 0.22 ± 0.13 [± 0.03] | 0.23 ± 0.14 [± 0.03] | 0.23 ± 0.15 [± 0.03] | 0.24 ± 0.14 [± 0.03] | 0.24 ± 0.14 [± 0.03] | 0.28 ± 0.14 [± 0.03] |

Table 6: Accuracy performance across datasets without response time variability. Values are reported as mean $\pm$ standard deviation over 100 trials.

| Dataset / Learner | 50% | 60% | 70% | 80% | 90% | 100% |
|---|---|---|---|---|---|---|
| **Deterministic no variability** | | | | | | |
| BT | 71.9 ± 5.5% [± 1.1] | 72.5 ± 5.6% [± 1.1] | 74.0 ± 5.4% [± 1.1] | 75.3 ± 5.2% [± 1.0] | 76.5 ± 5.1% [± 1.0] | 77.2 ± 5.0% [± 1.0] |
| RtRegression | **77.5** ± 5.5% [± 1.1] | **79.2** ± 4.9% [± 1.0] | **80.6** ± 4.9% [± 1.0] | **81.7** ± 4.8% [± 1.0] | **82.5** ± 4.4% [± 0.9] | **83.2** ± 4.1% [± 0.8] |
| RtRegression-Perm | 66.5 ± 6.2% [± 1.2] | 68.1 ± 5.7% [± 1.1] | 69.7 ± 5.4% [± 1.1] | 70.0 ± 5.8% [± 1.2] | 71.8 ± 5.6% [± 1.1] | 73.0 ± 5.7% [± 1.1] |
| ResponseRank | 75.1 ± 5.0% [± 1.0] | 77.5 ± 5.3% [± 1.1] | 79.0 ± 5.5% [± 1.1] | 79.9 ± 5.4% [± 1.1] | 81.1 ± 5.1% [± 1.0] | 81.5 ± 4.7% [± 0.9] |
| ResponseRank-Pool | 76.1 ± 4.9% [± 1.0] | 77.6 ± 5.5% [± 1.1] | 79.1 ± 5.0% [± 1.0] | 79.7 ± 5.1% [± 1.0] | 80.6 ± 4.9% [± 1.0] | 81.2 ± 4.7% [± 0.9] |
| ResponseRank-Perm | 67.3 ± 5.6% [± 1.1] | 67.5 ± 5.5% [± 1.1] | 66.5 ± 6.2% [± 1.2] | 67.9 ± 5.8% [± 1.1] | 69.2 ± 6.2% [± 1.2] | 69.3 ± 6.5% [± 1.3] |
| **Drift diffusion no variability** | | | | | | |
| BT | 69.8 ± 5.8% [± 1.1] | 71.7 ± 5.3% [± 1.0] | 73.0 ± 5.0% [± 1.0] | 74.3 ± 4.8% [± 1.0] | 75.5 ± 4.5% [± 0.9] | 75.8 ± 4.3% [± 0.9] |
| RtRegression | 71.3 ± 5.5% [± 1.1] | 72.9 ± 5.1% [± 1.0] | 73.9 ± 4.9% [± 1.0] | 75.0 ± 5.1% [± 1.0] | 76.0 ± 4.8% [± 1.0] | 76.3 ± 4.7% [± 0.9] |
| RtRegression-Perm | 67.4 ± 5.2% [± 1.0] | 68.8 ± 5.5% [± 1.1] | 69.8 ± 4.7% [± 0.9] | 71.1 ± 5.3% [± 1.0] | 71.4 ± 5.1% [± 1.0] | 72.3 ± 4.8% [± 1.0] |
| ResponseRank | **71.6** ± 6.6% [± 1.3] | **73.7** ± 5.9% [± 1.2] | **75.0** ± 5.7% [± 1.1] | **76.3** ± 5.2% [± 1.0] | **77.4** ± 4.6% [± 0.9] | **78.0** ± 4.5% [± 0.9] |
| ResponseRank-Pool | 70.9 ± 6.2% [± 1.2] | 72.2 ± 6.4% [± 1.3] | 73.6 ± 5.7% [± 1.0] | 74.3 ± 5.2% [± 1.0] | 75.3 ± 5.3% [± 1.0] | 76.0 ± 5.0% [± 1.0] |
| ResponseRank-Perm | 65.1 ± 6.3% [± 1.3] | 66.0 ± 5.6% [± 1.1] | 66.5 ± 6.1% [± 1.2] | 66.9 ± 5.3% [± 1.0] | 67.7 ± 5.6% [± 1.1] | 68.4 ± 5.4% [± 1.1] |
| **Stochastic no variability** | | | | | | |
| BT | 70.1 ± 5.0% [± 1.0] | 71.1 ± 5.4% [± 1.1] | 72.7 ± 4.9% [± 1.0] | 73.9 ± 4.8% [± 1.0] | 75.1 ± 4.3% [± 0.8] | 76.0 ± 4.3% [± 0.9] |
| RtRegression | **76.4** ± 4.5% [± 0.9] | **77.8** ± 4.7% [± 0.9] | **79.3** ± 4.1% [± 0.8] | **80.5** ± 4.1% [± 0.8] | **81.1** ± 4.2% [± 0.8] | **81.8** ± 4.0% [± 0.8] |
| RtRegression-Perm | 65.1 ± 6.7% [± 1.3] | 66.4 ± 6.3% [± 1.2] | 68.3 ± 5.1% [± 1.0] | 69.3 ± 5.7% [± 1.1] | 70.3 ± 5.0% [± 1.0] | 71.4 ± 5.3% [± 1.0] |
| ResponseRank | 74.3 ± 4.6% [± 0.9] | 76.0 ± 4.8% [± 1.0] | 77.8 ± 4.5% [± 0.9] | 79.1 ± 4.2% [± 0.8] | 80.0 ± 4.3% [± 0.9] | 80.7 ± 4.0% [± 0.8] |
| ResponseRank-Pool | 75.4 ± 4.6% [± 0.9] | 77.1 ± 4.3% [± 0.9] | 78.2 ± 4.5% [± 0.9] | 79.2 ± 4.3% [± 0.9] | 80.0 ± 4.6% [± 0.9] | 80.6 ± 4.2% [± 0.8] |
| ResponseRank-Perm | 66.3 ± 5.2% [± 1.0] | 66.3 ± 5.3% [± 1.0] | 66.8 ± 5.6% [± 1.1] | 67.1 ± 5.4% [± 1.1] | 67.0 ± 5.9% [± 1.2] | 68.3 ± 5.3% [± 1.1] |

We further find that in this setting, where RtRegression assumes a perfectly accurate model of the strength-RT relationship in the deterministic and stochastic datasets, it performs comparably to ResponseRank. This is to be expected, but not reflective of realistic settings.

We further include a boxplot of performances with the full dataset (Figures 8 and 9) and numerical results for the experiments presented in the main body (Tables 3 to 6). Boxplots (Figures 8 and 9), visualize median (center line), interquartile ranges (IQR; boxes), 95% confidence intervals of the median (notches), and outliers (dots).

# D   Supplemental details on MultiPref experiments

In this section we share additional results and details on our experiments with the MultiPref dataset, initially discussed in Section 5.

## D.1   Data processing

MultiPref is a dataset of 10,461 instances (5,323 unique prompts), each consisting of a prompt and two completions [15]. For each instance, *four distinct preference annotations* are collected: two from expert annotators (defined as having a graduate degree in the relevant domain) and two from normal crowdworkers, drawn from a pool of 227 total annotators. Each annotation includes *extensive metadata*: sub-preferences for helpfulness, truthfulness, and harmlessness; explicit preference strength (clear vs. slight); and time spent annotating.

**Filtering and splits.** We preprocess the dataset by filtering samples where one of the compared texts exceeds the maximum sequence length of 1,024 tokens resulting in a final dataset of 9,846 samples. We shuffle this filtered dataset and then split 2,000 samples off as a test set, resulting in *distinct splits for different random seeds*. We report all results across 30 random seeds.

**Annotation sampling.** We randomly sample one of the four annotations per comparison to simulate typical RLHF data collection for the training set. For evaluation, we use MultiPref's official `human_overall_binarized` labels, which aggregates annotations by majority vote. Tied preferences are ignored for both training and evaluation, resulting in smaller effective train and test set size varying slightly depending on the sampled annotations determined by the seed (26.32% of annotations in the full length-filtered dataset are ties).

**Fractional training sizes.** To simulate different training set sizes (reported as fractions in the line charts), we select a random subset of the training set, keeping the test set unchanged. This does not affect results reported in bar charts, which all use the full training set (fraction 1.0). Uniformly sampling individual comparisons could result in unrealistic annotation patterns with very few or even only one comparison per annotator. This would undermine stratification by annotator identity (Appendix D.2), which relies on sufficient within-annotator data. We therefore perform *annotator-aware sampling*: after annotation sampling, we group comparisons by annotator, shuffle these groups, and sequentially include groups until reaching the target fraction. All but the last group are included completely.

## D.2 Strength signals and stratification

Stratification groups comparisons into homogeneous subsets where the relationship between the strength signal and preference strength is locally valid, mitigating confounds such as individual annotator speed and task complexity. This is analogous to blocking in experimental design [34], where units are grouped to reduce unexplained variance. We consider three distinct strength signals from the MultiPref dataset:

**Response time.** We derive strength from the total time spent annotating each comparison, assuming shorter times indicate stronger preferences. Since annotation time may depend on individual annotator speed and reading time, we stratify by annotator identity and text length. We first discretize length by grouping instances into 8 global length buckets. Strata are then formed by the Cartesian product of annotator identity and length bucket. After stratification, we form rankings of size $\leq 2$ within each stratum using the partitioning strategy (Appendix B.1) to minimize the effect of noisy response times. Some resulting strata may have size one (2.0% of comparisons on average); those reduce to Bradley–Terry (Theorem 1). We refer to this variant as **RR-RT**. As discussed in Section 5, the response time signal captures not only the preference decision time, but also time spent on additional annotations and justifications, likely confounding its relationship with preference strength.

**Stated strength.** We use the explicit binary preference strength annotation (slight vs. clear) as the strength signal. Since this produces only two distinct strength values, most rankings are limited to size 2 (pairs of preferences with different stated strengths). As stated-strength judgments may be annotator-specific, we stratify by annotator identity before forming rankings. Within each stratum, we pair comparisons with different stated strengths using the partitioning strategy (Appendix B.1). As strengths are imbalanced (more 'slight' than 'clear'), many comparisons cannot be paired and remain singletons (33.1% of comparisons), which reduce to Bradley–Terry (Theorem 1). We refer to this variant as **RR-Stated**.

**Inter-annotator agreement.** We compute weighted inter-annotator agreement as our primary strength signal. Each of the four annotations per instance indicates either a clear or slight preference for one of the two responses. Given $N$ annotations in total, including $n_t$ ties, $n_c^+$ strong and $n_s^+$ slight preferences for the chosen response, and $n_c^-$ clear and $n_s^-$ slight preferences for the rejected response, we compute the agreement as

$$\text{Agreement} = \left(1.0 \cdot (n_c^+ - n_c^-) + 0.5 \cdot (n_s^+ - n_s^-)\right)/N \,.$$

We do not use any stratification for inter-annotator agreement. As MultiPref has $N = 4$ annotations per instance, the agreement score can only take a few discrete values, however, requiring *tie splitting* through the partitioning strategy (Appendix B.1). This results in rankings of size 4.58 on average. We refer to this variant as **RR-Agree**.

Table 7: Hyperparameters and training configuration for MultiPref experiments. We use the same hyperparameters for BT and all RR variants.

| Hyperparameter | Value |
|---|---|
| Learning Rate | 15e-6 |
| LR scheduler | Linear with 0.05 warmup ratio |
| Gradient Clipping | 1.0 |
| Weight Decay | 0.1 |
| AdamW optimizer settings ($\beta_1$, $\beta_2$, $\epsilon$) | (0.9, 0.999, 1e-08) |
| On-Device Batch Size | 16 |
| Gradient Accumulation Steps | 4 |
| Max Sequence Length (tokens) | 1024 |
| Dropout | no |
| Precision | bfloat16 |
| Epochs | 3 |

## D.3 Detailed experimental setup

**Model and hyperparameters.** Reward models are initialized from `Llama-3.1-8B-Instruct` [35] with a single classification head that extracts a scalar reward prediction from the last hidden state, fine-tuned in full using both BT and ResponseRank losses (negative log-likelihood). For both losses, we normalize by the batch size (number of comparisons). We use equal hyperparameters (Table 7) for both methods, supported by the hyperparameter sensitivity analysis in Appendix D.8.

**Evaluation.** We evaluate reward models on both in-distribution and out-of-distribution test data. The in-distribution test data is a held-out portion of the MultiPref dataset (2,000 samples) using aggregated labels (`human_overall_binarized`), providing an in-distribution evaluation with respect to prompts and responses. For out-of-distribution evaluation, we primarily rely on RewardBench 2 as it correlates strongly with downstream performance [16]. In Tables 8 to 10, we additionally report results for the original RewardBench benchmark [36]. As we did for the MultiPref dataset, we filter RewardBench for samples where both responses are within the 1024 token limit.

## D.4 Details on main experiments

**Performance vs. training set size.** Matching the synthetic experiments in Section 4, we investigate performance as a function of training set size to determine to which degree the best-performing strength-aware methods (primarily RR-Agree, with RR-Stated shown for comparison) improve sample efficiency. Figure 10 shows that RR-Agree outperforms BT by 3.0 percentage points on RewardBench 2. Additionally, RR-Agree matches final BT performance on that benchmark with approximately 30% fewer preference pairs, demonstrating its high potential for data-efficient preference learning.

**Numerical results.** To support reproducibility, we report numerical results for the experiments in the main text as well as the training data size experiments in Tables 8 to 10. We also report results for the original RewardBench benchmark (version 1) for additional validation. Note that reported confidence intervals assume normality and are not corrected for multiple comparisons.

**RewardBench category breakdown.** The RewardBench 2 benchmark consists of seven distinct categories, each measuring different aspects of reward model performance [16]. Figure 11 shows the performance of all the variants studied in the main text broken down by category as a function of training set size. It shows that variants that make use of an effective strength signal (agreement and, to a lesser degree, stated strength) outperform the BT baseline primarily in the *Safety*, *Ties*, and *Math* categories while the *Focus* category is most impacted by BT training duration (2 vs. 3 epochs). Performance increases on the particularly difficult Ties category highlight that agreement signals can help learn better-calibrated reward models.

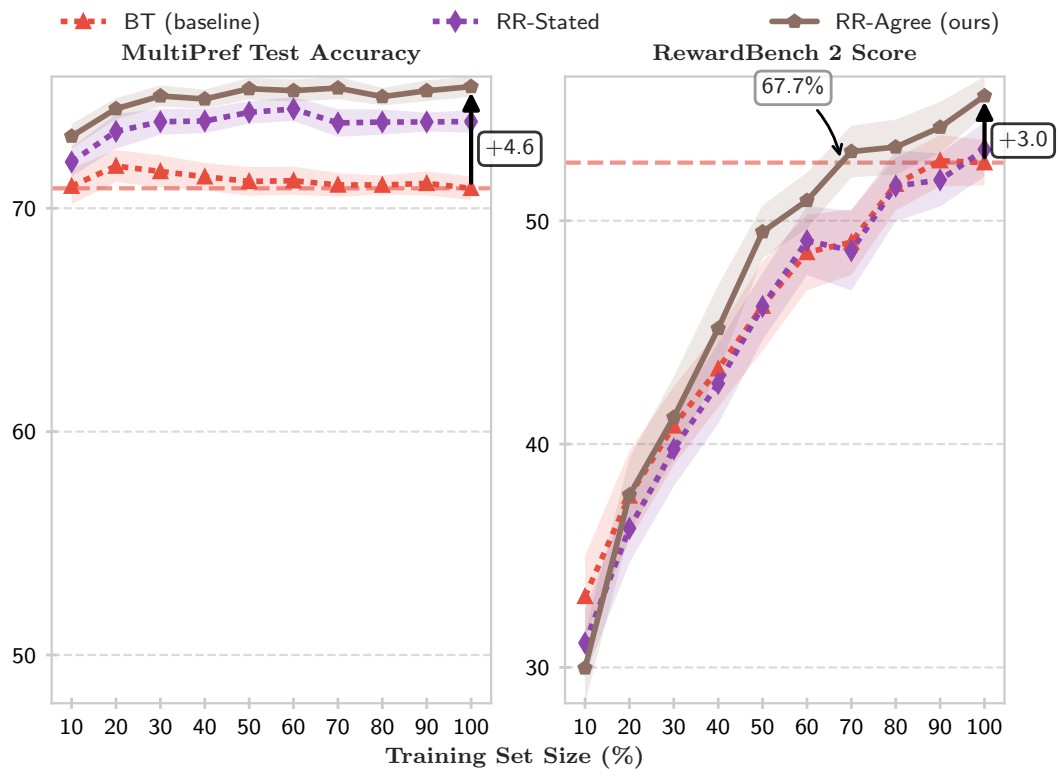

Figure 10: **Performance vs. training set size.** As a function of training set size (percent of entire training set), we show: MultiPref test set accuracy (in-distribution). RewardBench 2 score (out-of-distribution). 30 runs; shading indicates 95% CI assuming normality.

Table 8: MultiPref test set accuracy. Format: mean ± std [± 95% CI half-width]. Best values per column are bolded.

| Learner | 10% | 20% | 30% | 40% | 50% |
|---|---|---|---|---|---|
| BT | $71.0 \pm 2.2\%$ [± 0.8] | $71.9 \pm 1.9\%$ [± 0.7] | $71.6 \pm 1.9\%$ [± 0.7] | $71.4 \pm 1.7\%$ [± 0.6] | $71.2 \pm 1.7\%$ [± 0.6] |
| BT@2 | $71.8 \pm 1.7\%$ [± 0.6] | $73.5 \pm 1.6\%$ [± 0.6] | $73.6 \pm 1.7\%$ [± 0.6] | $73.6 \pm 1.7\%$ [± 0.6] | $74.2 \pm 1.6\%$ [± 0.6] |
| RR-Random | – | – | – | – | – |
| RR-RT | $71.3 \pm 1.7\%$ [± 0.6] | $71.9 \pm 1.9\%$ [± 0.7] | $72.2 \pm 1.7\%$ [± 0.6] | $72.7 \pm 1.9\%$ [± 0.7] | $72.5 \pm 1.6\%$ [± 0.6] |
| RR-Stated | $72.1 \pm 1.7\%$ [± 0.6] | $73.4 \pm 2.1\%$ [± 0.8] | $73.9 \pm 1.5\%$ [± 0.6] | $73.9 \pm 1.5\%$ [± 0.5] | $74.3 \pm 1.4\%$ [± 0.5] |
| RR-Agree | $\mathbf{73.2} \pm 1.5\%$ [± 0.6] | $\mathbf{74.5} \pm 1.3\%$ [± 0.5] | $\mathbf{75.0} \pm 1.3\%$ [± 0.5] | $\mathbf{74.9} \pm 1.1\%$ [± 0.4] | $\mathbf{75.4} \pm 1.3\%$ [± 0.5] |

| Learner | 60% | 70% | 80% | 90% | 100% |
|---|---|---|---|---|---|
| BT | $71.2 \pm 1.7\%$ [± 0.6] | $71.0 \pm 1.4\%$ [± 0.5] | $71.0 \pm 1.1\%$ [± 0.4] | $71.1 \pm 1.4\%$ [± 0.5] | $70.9 \pm 1.4\%$ [± 0.5] |
| BT@2 | $74.3 \pm 1.2\%$ [± 0.5] | $74.2 \pm 1.3\%$ [± 0.5] | $74.5 \pm 1.3\%$ [± 0.5] | $74.3 \pm 1.1\%$ [± 0.4] | $74.2 \pm 1.3\%$ [± 0.5] |
| RR-Random | – | – | – | – | $72.2 \pm 1.3\%$ [± 0.5] |
| RR-RT | $72.5 \pm 1.5\%$ [± 0.6] | $72.6 \pm 1.5\%$ [± 0.5] | $72.1 \pm 1.6\%$ [± 0.6] | $72.1 \pm 1.6\%$ [± 0.6] | $72.4 \pm 1.4\%$ [± 0.5] |
| RR-Stated | $74.4 \pm 1.4\%$ [± 0.5] | $73.8 \pm 1.6\%$ [± 0.6] | $73.9 \pm 1.5\%$ [± 0.6] | $73.9 \pm 1.1\%$ [± 0.4] | $73.9 \pm 1.4\%$ [± 0.5] |
| RR-Agree | $\mathbf{75.3} \pm 1.3\%$ [± 0.5] | $\mathbf{75.4} \pm 1.3\%$ [± 0.5] | $\mathbf{75.0} \pm 1.0\%$ [± 0.4] | $\mathbf{75.3} \pm 1.3\%$ [± 0.5] | $\mathbf{75.4} \pm 1.2\%$ [± 0.5] |

Table 9: RewardBench version 1 score. Format: mean ± std [± 95% CI half-width]. Best values per column are bolded.

| Learner | 10% | 20% | 30% | 40% | 50% |
|---|---|---|---|---|---|
| BT | **65.2** ± 5.1% [± 1.9] | 69.1 ± 5.3% [± 2.0] | 70.4 ± 5.1% [± 1.9] | 72.8 ± 4.1% [± 1.5] | 74.6 ± 4.0% [± 1.5] |
| BT@2 | 61.9 ± 4.5% [± 1.7] | 64.9 ± 5.2% [± 1.9] | 69.0 ± 4.9% [± 1.8] | 69.4 ± 6.0% [± 2.2] | 71.9 ± 4.8% [± 1.8] |
| RR-Random | – | | – | | |
| RR-RT | 61.1 ± 4.4% [± 1.6] | 64.1 ± 4.5% [± 1.7] | 66.7 ± 4.7% [± 1.7] | 68.6 ± 4.2% [± 1.6] | 71.2 ± 5.0% [± 1.9] |
| RR-Stated | 64.0 ± 5.3% [± 2.0] | 68.3 ± 4.8% [± 1.8] | 70.8 ± 4.6% [± 1.7] | 72.3 ± 4.1% [± 1.5] | 74.5 ± 3.8% [± 1.4] |
| RR-Agree | 62.4 ± 5.1% [± 1.9] | **69.5** ± 4.5% [± 1.7] | **71.3** ± 5.0% [± 1.9] | **74.4** ± 4.6% [± 1.7] | **77.0** ± 2.8% [± 1.0] |

| Learner | 60% | 70% | 80% | 90% | 100% |
|---|---|---|---|---|---|
| BT | 76.3 ± 3.0% [± 1.1] | 76.3 ± 3.0% [± 1.1] | 77.4 ± 3.2% [± 1.2] | 77.9 ± 3.2% [± 1.2] | 78.4 ± 2.3% [± 0.9] |
| BT@2 | 73.5 ± 5.4% [± 2.0] | 74.3 ± 4.5% [± 1.7] | 75.8 ± 3.1% [± 1.2] | 77.8 ± 3.7% [± 1.4] | 77.8 ± 3.7% [± 1.4] |
| RR-Random | – | – | – | – | 76.3 ± 2.9% [± 1.1] |
| RR-RT | 71.4 ± 3.5% [± 1.3] | 72.4 ± 4.1% [± 1.5] | 73.4 ± 3.6% [± 1.3] | 72.8 ± 4.1% [± 1.5] | 74.0 ± 5.2% [± 1.9] |
| RR-Stated | 77.0 ± 3.3% [± 1.2] | 75.6 ± 3.1% [± 1.2] | 78.4 ± 2.2% [± 0.8] | 78.3 ± 2.5% [± 0.9] | 79.2 ± 2.2% [± 0.8] |
| RR-Agree | **78.0** ± 2.6% [± 1.0] | **79.1** ± 2.3% [± 0.9] | **78.6** ± 2.8% [± 1.0] | **79.2** ± 1.9% [± 0.7] | **80.0** ± 2.0% [± 0.7] |

Table 10: RewardBench 2 score. Format: mean ± std [± 95% CI half-width]. Best values per column are bolded.

| Learner | 10% | 20% | 30% | 40% | 50% |
|---|---|---|---|---|---|
| BT | **33.2** ± 4.8% [± 1.8] | 37.7 ± 5.4% [± 2.0] | 40.8 ± 4.7% [± 1.8] | 43.4 ± 4.6% [± 1.7] | 46.2 ± 5.4% [± 2.0] |
| BT@2 | 29.9 ± 3.6% [± 1.3] | 32.9 ± 5.3% [± 2.0] | 36.8 ± 4.7% [± 1.7] | 38.8 ± 6.0% [± 2.2] | 41.6 ± 5.2% [± 1.9] |
| RR-Random | – | | – | | |
| RR-RT | 29.6 ± 3.5% [± 1.3] | 32.0 ± 4.3% [± 1.6] | 34.7 ± 4.7% [± 1.8] | 35.8 ± 4.1% [± 1.5] | 40.2 ± 4.9% [± 1.8] |
| RR-Stated | 31.1 ± 4.0% [± 1.5] | 36.2 ± 4.2% [± 1.6] | 39.8 ± 4.5% [± 1.7] | 42.7 ± 4.8% [± 1.8] | 46.2 ± 4.0% [± 1.5] |
| RR-Agree | 30.0 ± 4.2% [± 1.6] | **37.7** ± 4.4% [± 1.6] | **41.2** ± 4.9% [± 1.8] | **45.2** ± 5.2% [± 1.9] | **49.5** ± 3.2% [± 1.2] |

| Learner | 60% | 70% | 80% | 90% | 100% |
|---|---|---|---|---|---|
| BT | 48.6 ± 4.5% [± 1.7] | 49.0 ± 3.9% [± 1.5] | 51.6 ± 3.1% [± 1.2] | 52.7 ± 3.1% [± 1.1] | 52.6 ± 2.7% [± 1.0] |
| BT@2 | 44.7 ± 5.9% [± 2.2] | 46.9 ± 4.5% [± 1.7] | 48.8 ± 4.4% [± 1.6] | 49.9 ± 5.7% [± 2.1] | 51.1 ± 5.5% [± 2.0] |
| RR-Random | – | – | – | – | 47.7 ± 4.5% [± 1.7] |
| RR-RT | 41.2 ± 4.5% [± 1.7] | 43.2 ± 4.6% [± 1.7] | 45.2 ± 3.9% [± 1.5] | 44.0 ± 5.2% [± 1.9] | 47.2 ± 5.2% [± 2.0] |
| RR-Stated | 49.1 ± 4.1% [± 1.5] | 48.7 ± 4.8% [± 1.8] | 51.6 ± 4.2% [± 1.6] | 51.8 ± 3.2% [± 1.2] | 53.2 ± 3.2% [± 1.2] |
| RR-Agree | **50.9** ± 3.4% [± 1.3] | **53.1** ± 3.1% [± 1.2] | **53.3** ± 3.3% [± 1.2] | **54.2** ± 3.1% [± 1.1] | **55.6** ± 2.3% [± 0.9] |

## D.5 Validation experiments

Here we present additional experiments and ablations to validate our choices. All experiments report the mean performance across 30 seeds and indicate the 95% CI (assuming normality) in the error bars.

**Impact of ranking size.** As ties are prevalent in agreement scores, tie splitting and batch packing (Appendix B) result in rankings of at most size 5 (average size 4.58). To investigate the effect of ranking size on RR-Agree, we additionally test variants that limit ranking sizes to at most 2, 3, and 4 (resulting in average sizes of 2.00, 2.67, and 4.00 after batch packing[7]). Figure 12 shows performance increases with ranking size, demonstrating that larger rankings convey more information about preference strength. All variants, even with minimal rankings of size 2, outperform the BT baseline with the jump in performance between no strength signal (BT) and small rankings being particularly pronounced.

**Agreement signal with directional information.** Agreement scores are by design independent of the chosen preference, only reflecting the degree of consensus among annotators. This choice was made to simulate realistic settings where only a single annotator's choice is available, with agreement serving as a proof of concept for a strong strength signal. As a consequence of this choice, cases may arise where the sampled annotation disagrees with the majority opinion, yet the agreement score remains high (e.g., when 3 out of 4 annotators prefer the response rejected by the chosen annotation). To assess the impact of this design choice, we additionally evaluate a variant using *signed agreement* as the strength signals, where agreement is multiplied with +1 if the chosen response aligns with the mean opinion (where slight preferences are weighted as 0.5) and −1 otherwise. This is equivalent to using the mean preference as the strength signal, and we call this variant ***RR-MeanPref***. Figure 13

---

[7]As 3 does not divide the batch size, this leads to increased fragmentation during batch packing.

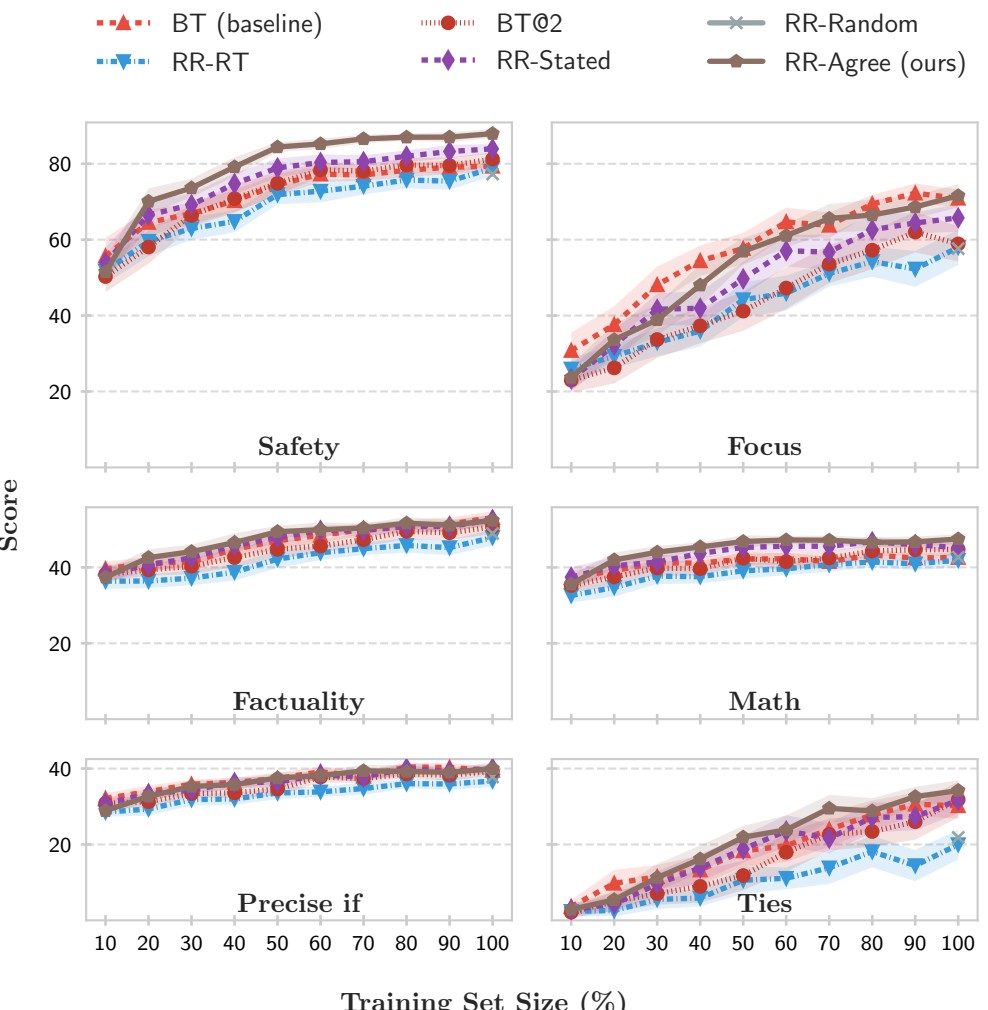

Figure 11: **RewardBench category breakdown.** As a function of training set size (percent of entire training set), we show RewardBench 2 performance for each category composing the benchmark. 30 runs; shading indicates $95\%$ CI assuming normality.

compares RR-Agree with RR-MeanPref. As MeanPref can take more distinct values than Agreement, it leads to larger average ranking sizes (average 7.03 compared to 4.58 for RR-Agree). We therefore again compare several different configurations with limited ranking sizes for RR-MeanPref (effective sizes: 2.00 for size 2, 4.00 for size 4, 4.80 for size 6). We observe that, while RR-MeanPref slightly increases performance on the in-distribution MultiPref test set, it underperforms RR-Agree on the out-of-distribution RewardBench 2 benchmark, indicating potential benefits of slightly noisy rank signals for generalization. The effects in both directions are small, however, and should not be over-interpreted. Note that agreement and MeanPref differ rarely as the majority opinion is sampled most of the time. We again observe increasing performance with increasing ranking size, though this effect plateaus at size 4 (the difference between size 6 and full size is not statistically significant: $+0.3$ pp, $p = 0.519$ on RewardBench, $-0.1$ pp, $p = 0.369$).

**BT with agreement scores as soft labels.** As agreement scores are a particularly clean and informative strength signal that gives a clear cardinal target, it would be natural to use them directly as soft labels in the BT loss. We therefore evaluate two alternatives: **BT-MeanPref**, which uses the mean opinion as the target preference probability like RR-MeanPref; and **BT-Agree**, which uses $p^* = 0.5 + \text{Agreement}/2$ as the choice probability for the chosen response. BT-MeanPref therefore uses a stronger signal than agreement, which is stronger than the binary preference observed by

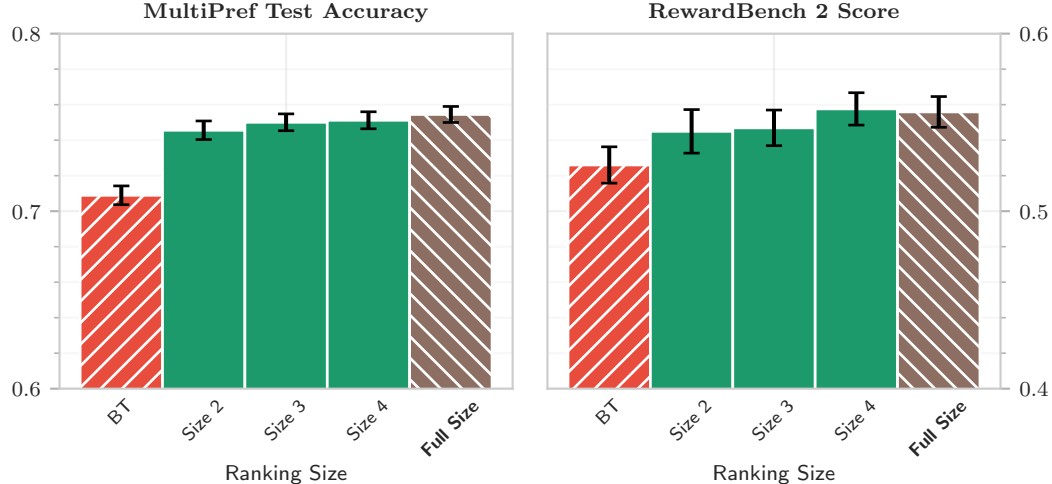

Figure 12: **Impact of ranking size for RR-Agree.** Effective partition sizes after batch packing: 2.00 (size 2), 2.67 (size 3), 4.00 (size 4), 4.58 (full size).

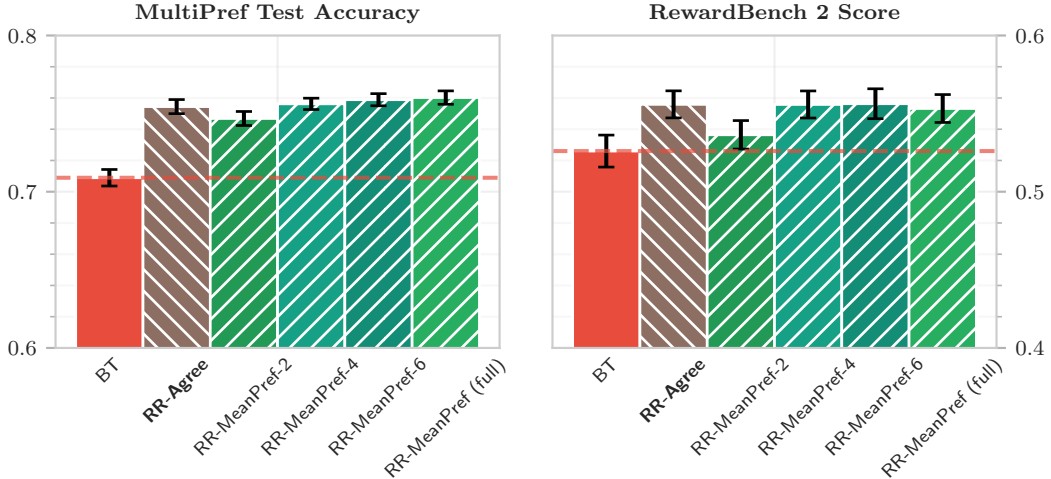

Figure 13: **RR-Agree vs. RR-MeanPref ranking comparison.** We compare RR-Agree with different ranking size configurations of RR-MeanPref. The dashed baseline represents the BT model.

standard BT. Figure 14 compares these approaches, showing a modest improvement of BT-Agree over BT in both MultiPref test ($+0.2$ pp, $p = 0.179$) and RewardBench ($+0.5$ pp, $p = 0.314$). While BT-MeanPref leads to a further improvement on MultiPref test ($+0.3$ pp vs. BT, $p = 0.057$), it under-performs the BT baseline on RewardBench ($-0.4$ pp, $p = 0.694$), indicating potential overfitting to the training distribution. All these effects are small and not statistically significant. They highlight, however, how difficult soft labels can be to use effectively.

### D.6 Robustness analysis

**Robustness to noisy strength signals.** To assess the robustness of RR-Agree to noisy preference data, we conducted experiments with varying levels of data corruption using partial shuffle noise. This method randomly selects a specified percentage of the preference annotations from the training dataset (prior to stratification) and shuffles their strength values among themselves.[8] 0% represents clean data and 100% represents completely randomized preferences. Figure 15 shows performance of

---

[8]While more realistic ranking noise models could be considered, partial shuffle noise provides a simple and interpretable way to control the noise level. More realistic alternatives such as Mallows would operate on a

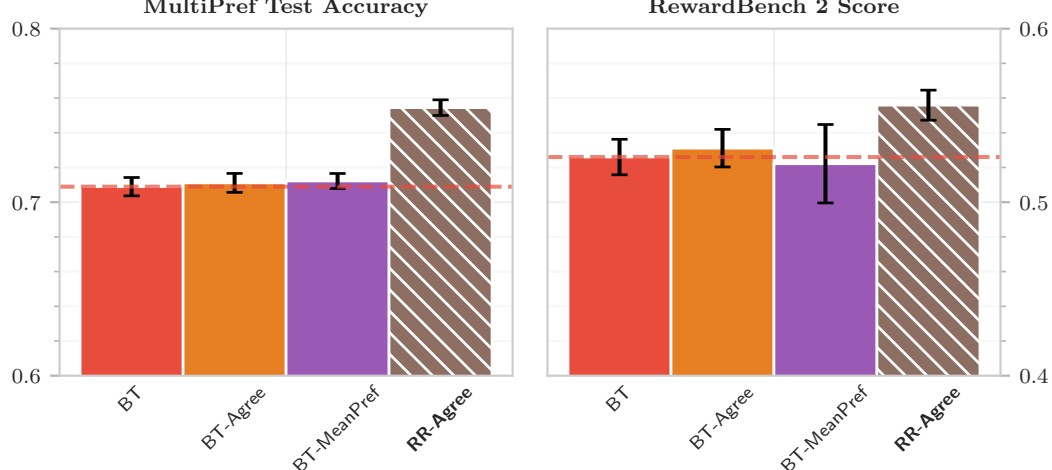

Figure 14: **BT with agreement scores as soft labels.** We compare BT baseline, BT-Agree (BT loss with agreement scores as targets), BT-MeanPref (BT loss with mean preference as targets), and RR-Agree (ResponseRank with agreement-based ranking).

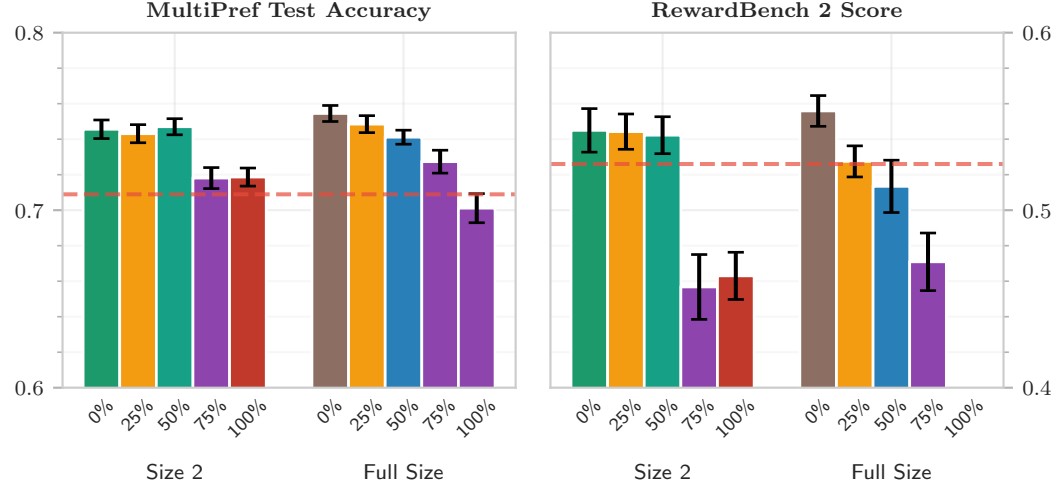

Figure 15: **Robustness to noisy strength signals (RR-Agree).** Comparison of two variants of RR-Agree, one with full ranking size and one limited to size 2, under varying levels of partial shuffle noise.

RR-Agree and the 'size 2' variant which uses the same strength signal with smaller rankings across various noise levels. It shows that RR-Agree falls below baseline performance with 50% noise, while the smaller rankings are much more robust to noise. This motivates our choice of ranking size 2 for the response time experiments with assumed noisy strength signals.

We further confirm the effect of ranking size on noise sensitivity in Figure 16, where we compare RR-Random configurations (random, uninformative strength information) with rankings of different sizes. We observe that performance degrades sharply with increasing ranking size, demonstrating that larger rankings make the ResponseRank loss more sensitive to noise in the strength signal.

**Robustness to missing strength data.** To evaluate the robustness of RR-Agree to missing rank data, we conducted experiments with different filter fractions applied to the training data. We systematically varied the fraction of ranks retained during training. As shown in Figure 17, RR-Agree performance

---

rank level (after stratification) instead of on a value level, making comparison between different ranking sizes challenging, and have less interpretable parameters.

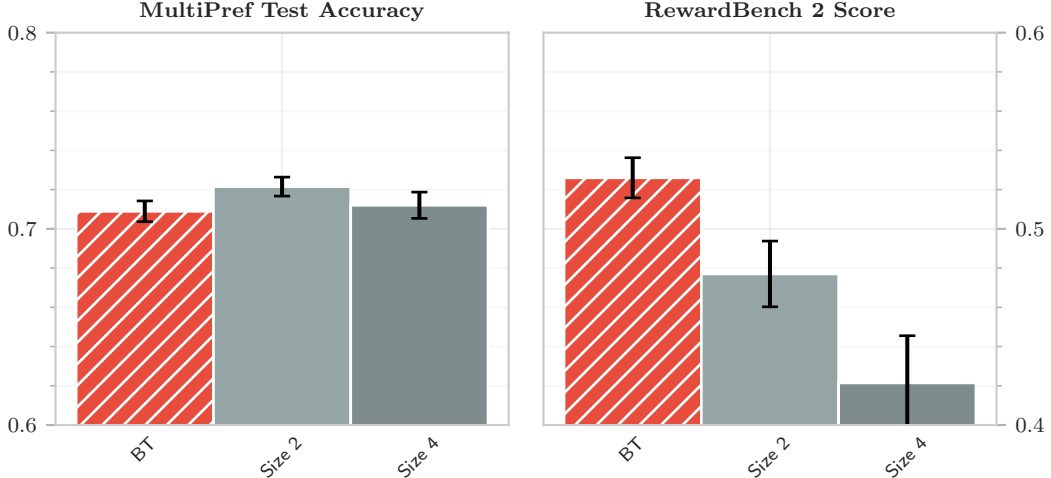

Figure 16: **Robustness to noisy strength signals (RR-Random).** Comparison of RR-Random with different ranking sizes (2 vs 4) using completely random strength signals.

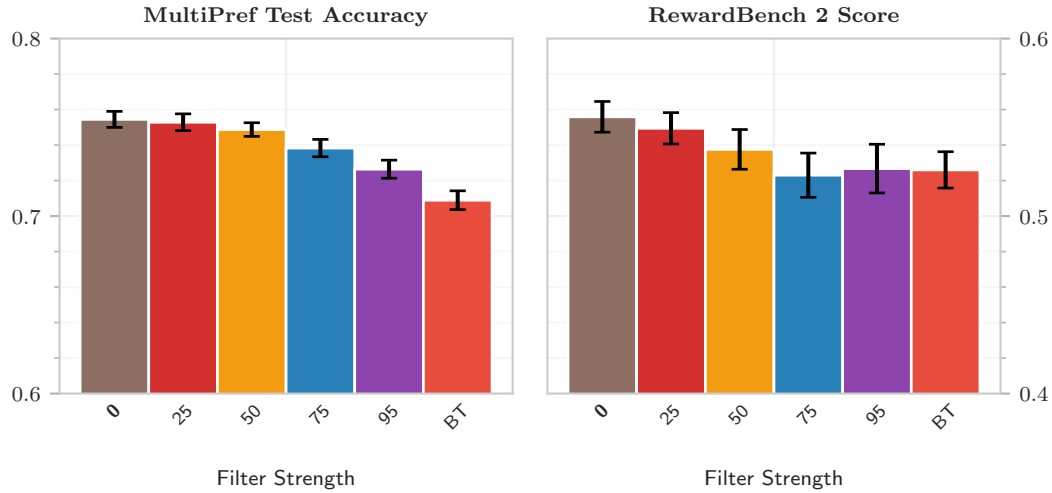

Figure 17: **Robustness to missing strength data.** Filter robustness analysis comparing RR-Agree variants with different filter fractions applied to the training data.

degrades gracefully to BT performance. While very little strength information is sufficient for performance increases on the MultiPref test set, performance regresses much faster on RewardBench 2 and reaches baseline performance at around 75% missing data.

## D.7 Signal-specific stratification analysis

**Impact of stratification on response time signal.** As individual annotators are likely to have different baseline speeds, we stratify by annotator identity in our response time experiments (RR-RT). Since longer texts naturally require more time to read and evaluate, we additionally apply length stratification to ensure fair comparisons: examples are grouped by text length into buckets, and response times are only compared within each bucket. Here we ablate both choices. Figure 18 shows results for RR-RT with different levels of length stratification (fewer buckets means coarser stratification). It shows very little impact of stratification on RR-RT performance, further supporting that the response time signal is weak in this dataset, independent of stratification choice. This may also question the assumption that longer texts necessarily require more time to evaluate, as annotators may skim or skip parts of longer texts. The MultiPref evaluation shows a slight benefit of stratification

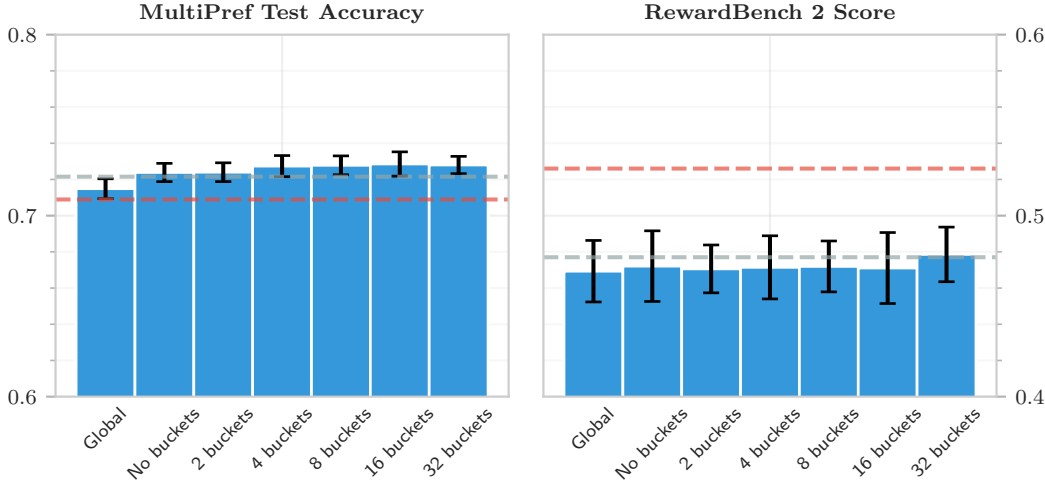

Figure 18: **Impact of stratification on response time signal (length buckets).** Comparison of different stratification approaches (no stratification, annotator-only, annotator + 2/4/8/16/32 length buckets) for the RT-based approach. Reference lines show BT baseline (red) and RR-Random (gray).

(4 bucket is better than annotator-only which is better than no stratification), but this is not reflected in RewardBench performance. It is notable that the no-stratification condition underperforms the random baseline (RR-Random) on MultiPref test. One possible explanation is that systematically fast annotators provide low-quality annotations (classified as strong by the RT signal) or longer texts (with longer RTs) being systematically classified as weak. We choose 8 length buckets for our main experiments as a compromise between granularity and sufficient data per bucket, but the figure indicates that annotator-only stratification or even no stratification would perform similarly.

To put these empirical results into context, we analyze correlations in the dataset to understand the relationship between response time, preference strength, and text length. Here we treat 'consensus', which has empirically proven a reliable strength signal, as a proxy for the unknown true preference strength. We find that *RT is strongly correlated with length* (within-annotator median $\rho = +0.47$), supporting the intuition that longer texts take longer to evaluate. The overall correlation between RT and consensus is weak (global $\rho = -0.05$; negative values indicate that lower RT corresponds to higher strength). *Stratifying by annotator improves the correlation slightly* (within-annotator median $\rho = -0.08$), demonstrating that controlling for annotator-specific effects helps. However, when additionally stratifying by length (8 buckets), the correlation weakens to $\rho = -0.05$, showing that *length stratification does not help* in this dataset. This is because the already-weak RT signal operates largely through text length: Length itself has a weak correlation with consensus (within-annotator length-consensus median $\rho = -0.07$), indicating that longer texts tend to have weaker preferences. Since RT and length are strongly correlated, controlling for length removes part of the predictive power that RT has. This analysis shows that RT is a weak strength signal in this dataset and, when used, annotator-only stratification is preferable. Note, however, that RT variance is not fully explained by length, and datasets with simpler annotation protocols (e.g., forced binary choices with immediate feedback) may exhibit stronger RT-strength correlations.

A large part of the difference of performance across different length stratification choices may be explained by changes in ranking structure: Stratification increases singleton fractions from 0.0001 (no stratification) to 0.02 (annotator-only), then progressively to 0.04 (2 buckets), 0.08 (4), 0.14 (8), 0.26 (16), and 0.42 (32).

**Impact of stratification on stated strength.** We evaluate RR-Stated, an alternative approach that uses explicitly stated preference strength from annotators rather than inter-annotator agreement. We compare two variants: the standard RR-Stated approach with local stratification, RR-Stated (global) which uses global ranking across all examples, and RR-Stated (partialshuffle100) which applies full shuffling noise to test robustness. As shown in Figure 19, the stated strength approach provides useful signal but underperforms compared to the agreement-based method.

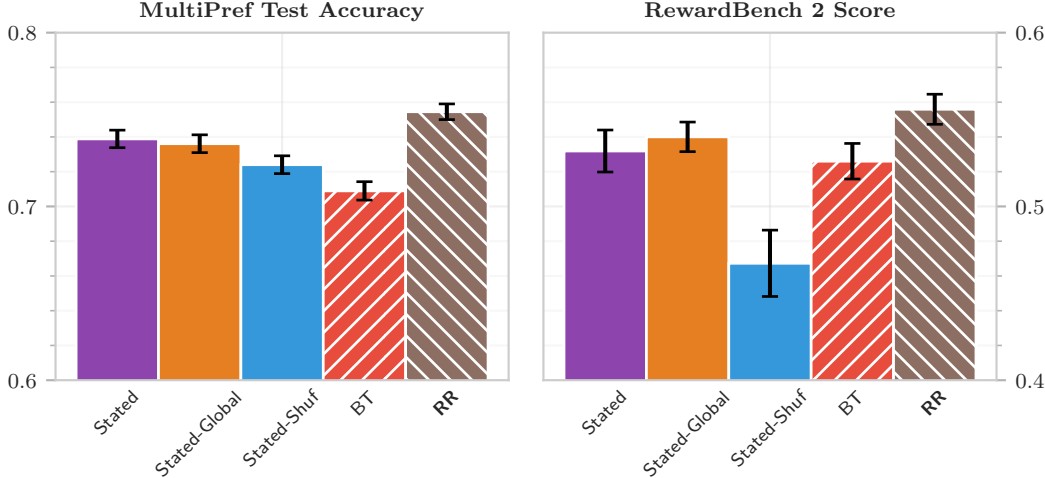

Figure 19: **Stated preference strength ablation.** Comparison of RR-Stated variants that use explicit annotator-stated preference strength, compared to BT baseline and RR-Agree which uses inter-annotator agreement. The stated strength approach provides signal but underperforms the agreement-based method.

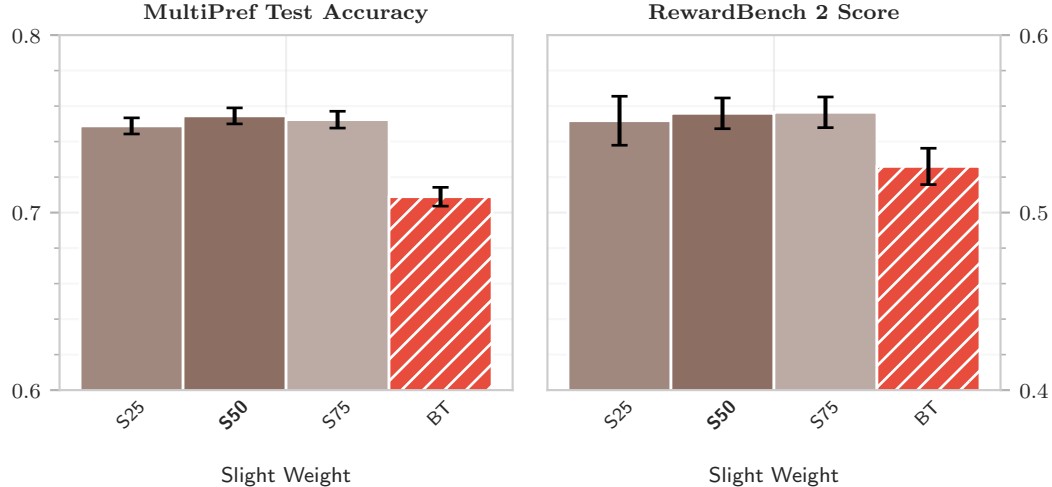

Figure 20: **Weighting of slight preferences.** Comparison of RR-Agree variants with varying weight for slight preferences, showing robustness to different choices.

**Weighting of slight preferences.** In our main experiments we assign slight preferences weight 0.5 and clear preferences weight 1.0 for the agreement computation. This choice is somewhat arbitrary and depends on each annotator's individual calibration of 'slight'. Here we compare two alternative variants where we assign slight preferences weight 0.25 and 0.75 respectively. Figure 20 shows that performance varies slightly across these variants, but is largely robust to this choice.

### D.8 Hyperparameter sensitivity

To evaluate the robustness of both BT and RR-Agree across different training configurations, we conducted a hyperparameter sensitivity analysis. We systematically vary learning rate (Figure 21), gradient accumulation steps (Figure 22), gradient clipping (Figure 23), number of training epochs (Figure 24), warmup ratio (Figure 25), and weight decay (Figure 26). The values used in our main experiments are **bolded**. While some hyperparameters have meaningful impact (particularly

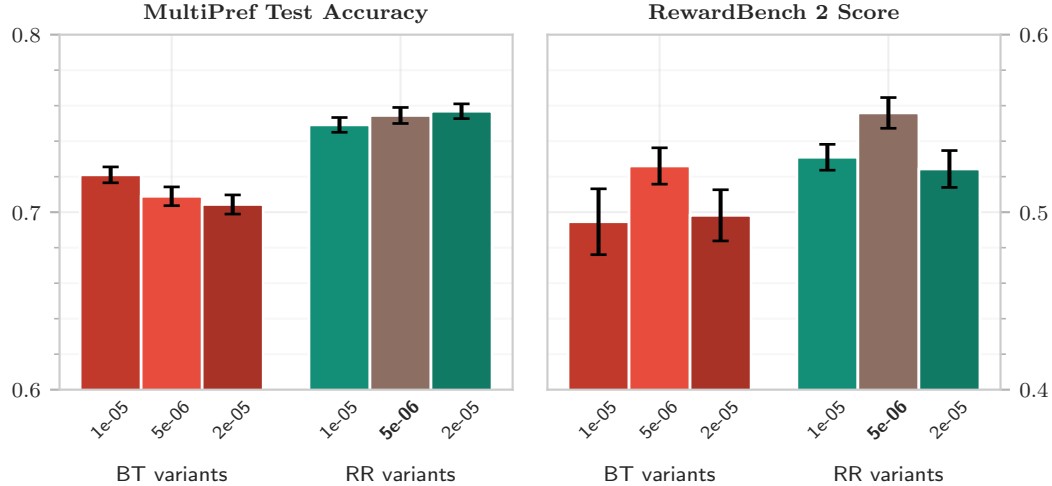

Figure 21: **Learning rate** sensitivity analysis.

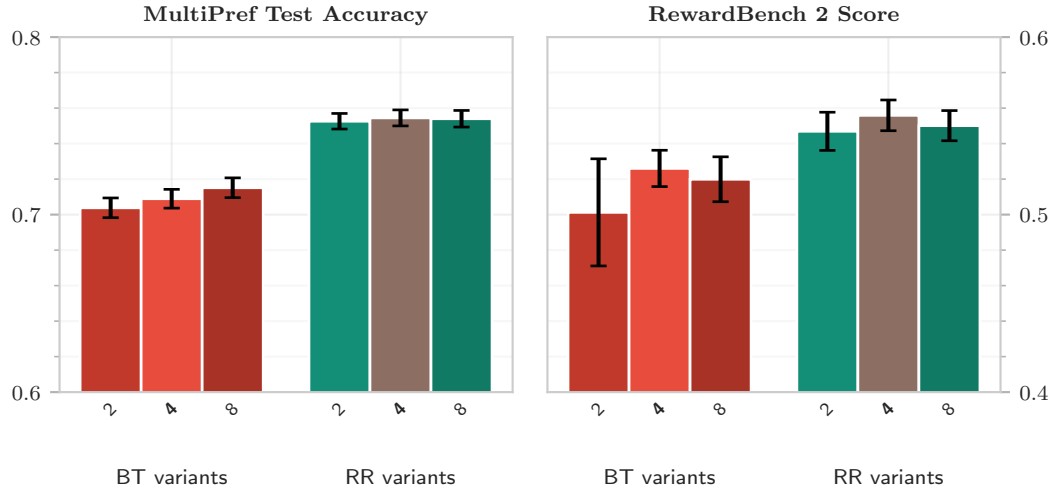

Figure 22: **Gradient accumulation steps** sensitivity analysis.

those related to the amount of optimization, e.g., epochs and learning rate), both methods perform reasonably across a range of values.

The main hyperparameters were selected through exploratory manual tuning during development, using RewardBench 2 performance as feedback. While this could lead to optimistic absolute performance estimates, the scope for such overfitting is limited: several hyperparameters have minimal impact, and the tuning was small-scale exploratory rather than systematic optimization. Identical settings are used for BT and all ResponseRank variants, and the sensitivity analysis confirms that these are favorable for both methods, ensuring fair comparison.

## E Supplemental details on RL control experiments

Our experiments show that ResponseRank improves reward model accuracy across synthetic datasets and human preference datasets. In an additional series of experiments, we investigate the effectiveness of our approach for downstream reinforcement learning performance. We investigate this in a control experiment with synthetic preference labels and preference strength information.

**Setup.** We are interested in the performance of RL agents trained with learned reward models based on the *ResponseRank* loss. We evaluate downstream performance in *MuJoCo* [37], a continuous

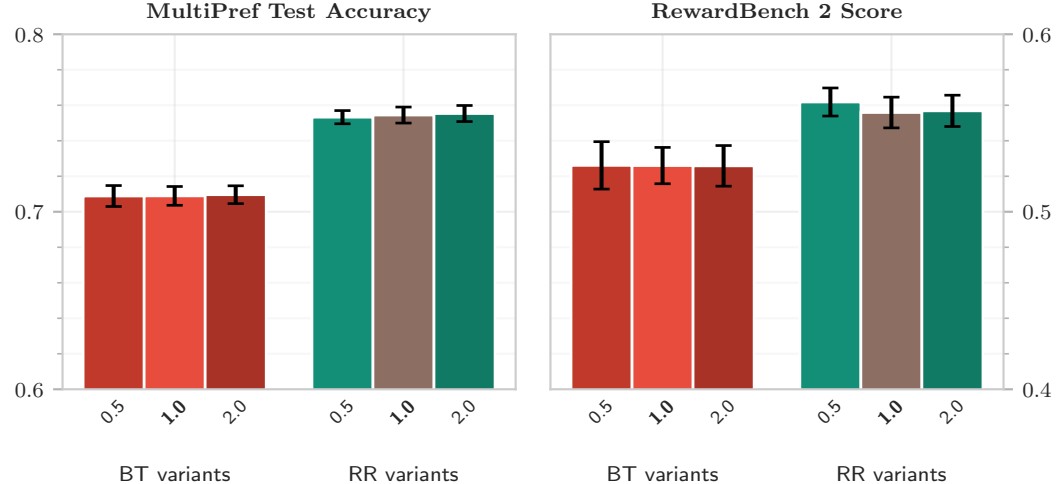

Figure 23: **Gradient clipping** sensitivity analysis.

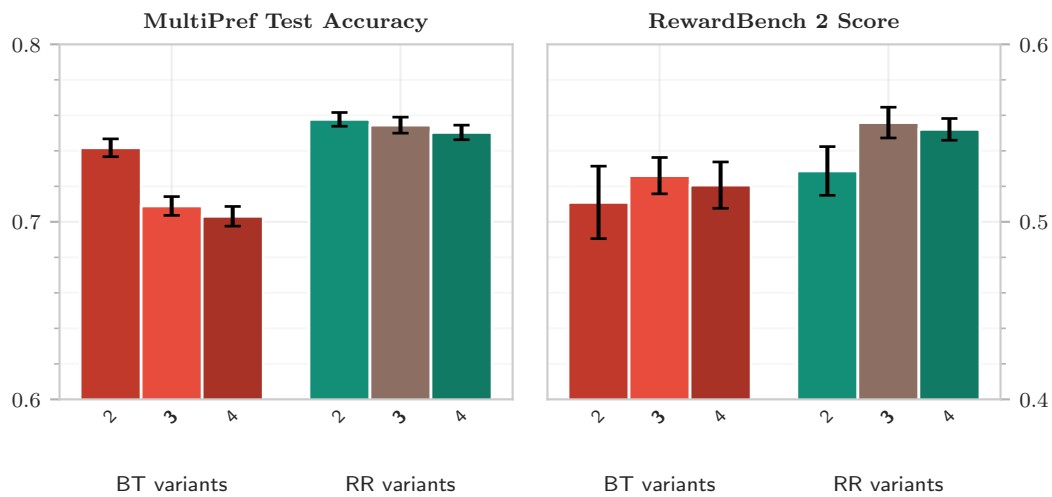

Figure 24: **Training epochs** sensitivity analysis.

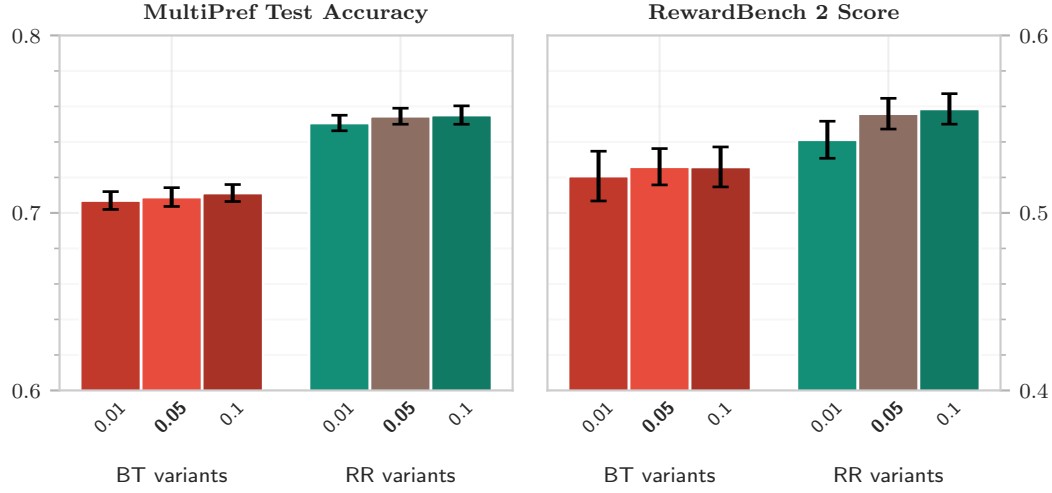

Figure 25: **Warmup ratio** sensitivity analysis.

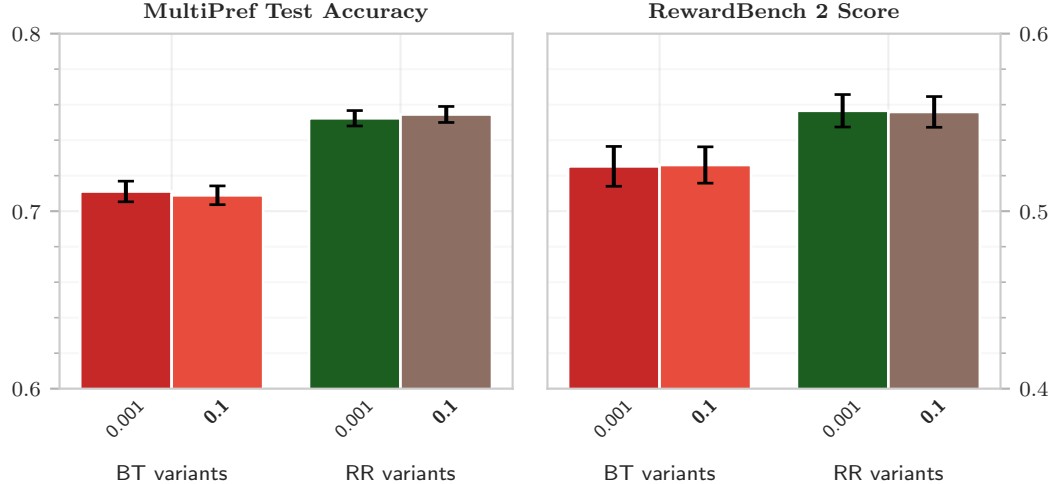

Figure 26: **Weight decay** sensitivity analysis.

control task, and *highway-env* [38], a benchmark environment with a discrete action space. Both are well-established environments in RL research. As the RL training algorithm, we use *PPO* [39] from the *Stable-Baselines3* implementation [40]. For downstream RL evaluation, we use the same hyperparameters from Stable Baselines3 Zoo across all experimental conditions.

**Generating synthetic preference and strength labels.** We follow methodology from Metz et al. [17]. We collect a dataset of trajectories based on expert models (trained via standard RL on ground-truth rewards) that generate trajectories sampled across multiple checkpoints, ensuring diverse policy coverage and skill levels for our offline dataset, similar to [41]. Then we randomly sample 5000 segment pairs of length 50 (truncated at episode boundaries) for reward model training. Out of these 5000 pairs, 4000 are used as the reward model training set, and 1000 samples are used as the validation set. We generate synthetic preference and preference-strength labels from ground truth-rewards $r_{gt}(s, a)$ for state-action pairs in the environment. Given an annotated trajectory pair $(a_i, b_i)$ and discounted return $R_{gt}(seg) = \sum_{t \in 0:50} \lambda^t r_{gt}(s_t, a_t)$ we synthesize the preference using

$$a_i \succ b_i \Leftrightarrow R_{gt}(a_i) \geq R_{gt}(b_i)$$

and the strength as $s_\theta(a_i, b_i) = R_{gt}(a_i) - R_{gt}(b_i)$. Clearer preference differences correspond to lower simulated response times. Note that since only the order of pseudo RTs matters, negative values are allowed, and any preference strength estimate could substitute for response time. The discount factor $\lambda$ is specific to an environment and fixed across all training configurations. We then randomly construct rankings of size 16 (matching our on-device batch size, avoiding the need for batch packing) for reward model training using the ResponseRank loss.

**Noise** To test robustness to reward noise, we also experimented with perturbed rewards. Here, we sample rewards from a truncated Gaussian distribution with mean equal to the retrieved environment reward and scale $\sigma = \alpha \cdot \sigma_r$, where $\alpha$ is the noise factor (from tables) and $\sigma_r$ is the dataset reward standard deviation. Truncation ensures samples stay within the observed reward range, following prior work [17], though it's not essential to our approach. Higher noise levels reduce both preference accuracy and strength reliability.

**Training** As reward models, we use 6-layer MLPs with 256 hidden units, processing concatenated state and action vectors (one-hot encoded for discrete action spaces). We optimize with AdamW [42] with a learning rate of $1e - 5$, weight decay enabled, batch size of 16, and early stopping on a validation holdout set (patience=5). We train the reward model on the pre-collected data, and do not use further online training of the reward model during RL training.

**Reward Model Training** In Table 11, we report reward model accuracies, i.e., prediction accuracy for identifying the preferred segment. We find that ResponseRank outperforms the BT-baseline in most configurations, but that accuracies only differ slightly. We hypothesize that the improved downstream RL performance (see Figure 27), is due to the learned reward models predicting a better shaped reward function compared to the Bradley-Terry baseline.

Table 11: Validation accuracies of reward models for different environments and noise levels. ResponseRank reward models exhibit higher reward model accuracy, but differences are minor and generally fall within the confidence interval.

| Environment / noise | RR | BT |
|---|---|---|
| HalfCheetah-v5 / noise=0 | **0.998** ± 0.001 [± 0.001] | 0.995 ± 0.005 [± 0.006] |
| HalfCheetah-v5 / noise=0.1 | **0.982** ± 0.004 [± 0.006] | 0.980 ± 0.004 [± 0.006] |
| HalfCheetah-v5 / noise=0.25 | **0.932** ± 0.004 [± 0.005] | 0.930 ± 0.005 [± 0.007] |
| HalfCheetah-v5 / noise=0.5 | **0.864** ± 0.003 [± 0.003] | 0.857 ± 0.011 [± 0.016] |
| Swimmer-v5 / noise=0 | **0.979** ± 0.012 [± 0.016] | 0.948 ± 0.009 [± 0.013] |
| Swimmer-v5 / noise=0.1 | **0.920** ± 0.018 [± 0.026] | 0.904 ± 0.020 [± 0.028] |
| Swimmer-v5 / noise=0.25 | **0.812** ± 0.028 [± 0.039] | 0.799 ± 0.027 [± 0.037] |
| Swimmer-v5 / noise=0.5 | **0.716** ± 0.033 [± 0.046] | 0.714 ± 0.029 [± 0.041] |
| Walker2d-v5 / noise=0 | **1.000** ± 0.001 [± 0.001] | 0.996 ± 0.002 [± 0.003] |
| Walker2d-v5 / noise=0.1 | **0.990** ± 0.002 [± 0.002] | 0.988 ± 0.001 [± 0.002] |
| Walker2d-v5 / noise=0.25 | **0.955** ± 0.008 [± 0.011] | 0.952 ± 0.008 [± 0.011] |
| Walker2d-v5 / noise=0.5 | **0.884** ± 0.005 [± 0.007] | 0.883 ± 0.004 [± 0.005] |
| merge-v0 / noise=0 | 0.843 ± 0.012 [± 0.017] | **0.845** ± 0.010 [± 0.014] |
| merge-v0 / noise=0.1 | **0.829** ± 0.013 [± 0.018] | 0.828 ± 0.013 [± 0.018] |
| merge-v0 / noise=0.25 | **0.798** ± 0.020 [± 0.028] | 0.789 ± 0.018 [± 0.025] |
| merge-v0 / noise=0.5 | **0.714** ± 0.010 [± 0.014] | 0.712 ± 0.010 [± 0.013] |

Table 12: RL control with reward noise (final reward).

| Environment / noise | BT | RR |
|---|---|---|
| HalfCheetah-v5 / noise=0 | 5083.9 ± 621.0 [± 862.1] | **5345.0** ± 311.9 [± 433.0] |
| HalfCheetah-v5 / noise=0.1 | **5014.9** ± 595.3 [± 826.4] | 4946.7 ± 493.5 [± 685.2] |
| HalfCheetah-v5 / noise=0.25 | 4847.7 ± 567.5 [± 787.9] | **5448.2** ± 526.5 [± 730.9] |
| HalfCheetah-v5 / noise=0.5 | 4491.9 ± 533.8 [± 741.1] | **4890.7** ± 767.4 [± 1065.3] |
| Swimmer-v5 / noise=0 | 21.2 ± 23.8 [± 33.0] | **167.7** ± 145.2 [± 201.6] |
| Swimmer-v5 / noise=0.1 | 32.2 ± 12.9 [± 17.9] | **81.8** ± 107.1 [± 148.7] |
| Swimmer-v5 / noise=0.25 | 9.8 ± 27.1 [± 37.7] | **83.4** ± 122.8 [± 170.4] |
| Swimmer-v5 / noise=0.5 | 58.5 ± 51.8 [± 71.9] | **81.1** ± 128.7 [± 178.7] |
| Walker2d-v5 / noise=0 | **2531.5** ± 570.8 [± 792.4] | 2363.9 ± 1183.3 [± 1642.7] |
| Walker2d-v5 / noise=0.1 | 2526.2 ± 730.5 [± 1014.1] | **2742.2** ± 1164.8 [± 1617.0] |
| Walker2d-v5 / noise=0.25 | **2978.8** ± 539.8 [± 749.4] | 2042.0 ± 1295.0 [± 1797.7] |
| Walker2d-v5 / noise=0.5 | **2391.6** ± 965.8 [± 1340.8] | 1636.8 ± 758.4 [± 1052.8] |
| merge-v0 / noise=0 | 11.6 ± 0.3 [± 0.5] | **12.1** ± 0.2 [± 0.3] |
| merge-v0 / noise=0.1 | 11.2 ± 0.2 [± 0.3] | **12.0** ± 0.4 [± 0.6] |
| merge-v0 / noise=0.25 | 11.4 ± 0.5 [± 0.6] | **12.1** ± 0.6 [± 0.9] |
| merge-v0 / noise=0.5 | 11.2 ± 0.6 [± 0.8] | **12.4** ± 0.3 [± 0.4] |

**Downstream RL Results** Figure 27 shows that ResponseRank generally outperforms BT models. For merge-v0, ResponseRank consistently outperforms despite similar reward model accuracy, suggesting better reward shaping. In all environments, the maximum reward achieved during training is higher for the RL agent with access to the ResponseRank reward model. For all but one environment, also the final reward is higher.

**Noise robustness.** The results above were measured with perfect environment rewards. To measure robustness to noise, we also benchmarked with different noise levels (following Metz et al. [17]), which perturbs the underlying ground-truth reward and in turn extracted response times and preferences. Figure 27 (numerical results in Tables 12 and 13) shows that ResponseRank outperforms the BT-model in low-noise scenarios, and only underperforms when overall RL performance deteriorates significantly. In environments that are generally robust to noise (such as merge-v0 or HalfCheetah), ResponseRank also stays stable.

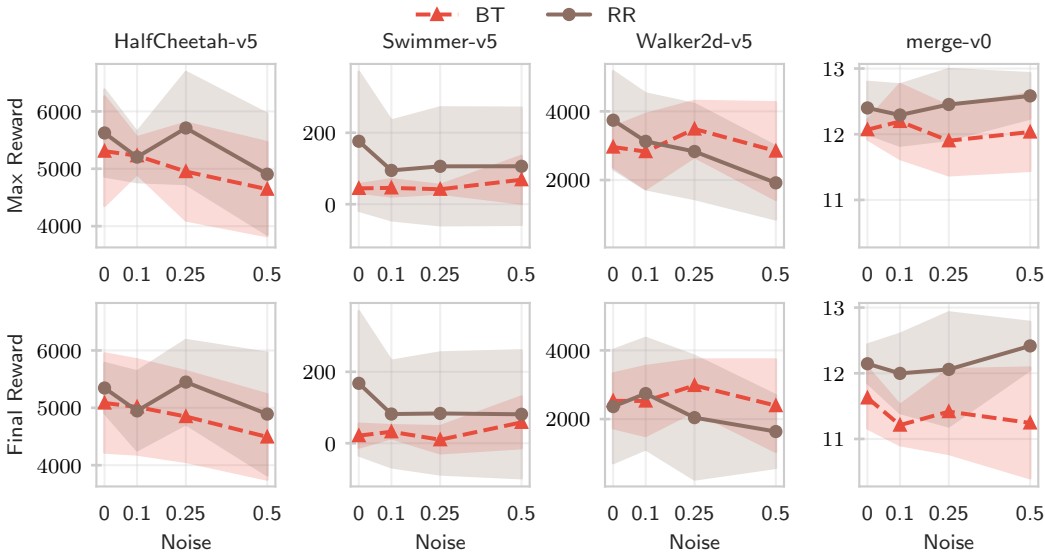

Figure 27: **RL control with reward noise.** Showing final and maximum reward (mean and 95% CI) across 5 seeds as a function of noise.

Table 13: RL control with reward noise (max reward).

| Environment / noise | BT | RR |
|---|---|---|
| HalfCheetah-v5 / noise=0 | $5306.8 \pm 685.6\ [\pm 951.8]$ | $\mathbf{5623.6} \pm 544.2\ [\pm 755.4]$ |
| HalfCheetah-v5 / noise=0.1 | $\mathbf{5227.7} \pm 230.0\ [\pm 319.3]$ | $5203.7 \pm 312.5\ [\pm 433.8]$ |
| HalfCheetah-v5 / noise=0.25 | $4951.4 \pm 611.6\ [\pm 849.1]$ | $\mathbf{5710.4} \pm 701.3\ [\pm 973.6]$ |
| HalfCheetah-v5 / noise=0.5 | $4644.0 \pm 588.5\ [\pm 816.9]$ | $\mathbf{4905.3} \pm 756.5\ [\pm 1050.2]$ |
| Swimmer-v5 / noise=0 | $44.3 \pm 8.2\ [\pm 11.3]$ | $\mathbf{176.3} \pm 139.9\ [\pm 194.2]$ |
| Swimmer-v5 / noise=0.1 | $45.7 \pm 17.0\ [\pm 23.7]$ | $\mathbf{94.6} \pm 100.3\ [\pm 139.2]$ |
| Swimmer-v5 / noise=0.25 | $41.8 \pm 8.4\ [\pm 11.6]$ | $\mathbf{106.0} \pm 118.6\ [\pm 164.6]$ |
| Swimmer-v5 / noise=0.5 | $68.6 \pm 47.8\ [\pm 66.3]$ | $\mathbf{106.2} \pm 117.8\ [\pm 163.5]$ |
| Walker2d-v5 / noise=0 | $2961.1 \pm 419.9\ [\pm 582.9]$ | $\mathbf{3741.9} \pm 1024.6\ [\pm 1422.3]$ |
| Walker2d-v5 / noise=0.1 | $2828.0 \pm 785.2\ [\pm 1090.1]$ | $\mathbf{3126.1} \pm 1001.5\ [\pm 1390.3]$ |
| Walker2d-v5 / noise=0.25 | $\mathbf{3486.3} \pm 585.2\ [\pm 812.4]$ | $2831.4 \pm 993.7\ [\pm 1379.4]$ |
| Walker2d-v5 / noise=0.5 | $\mathbf{2844.1} \pm 1020.5\ [\pm 1416.6]$ | $1918.2 \pm 764.5\ [\pm 1061.3]$ |
| merge-v0 / noise=0 | $12.1 \pm 0.1\ [\pm 0.1]$ | $\mathbf{12.4} \pm 0.3\ [\pm 0.4]$ |
| merge-v0 / noise=0.1 | $12.2 \pm 0.4\ [\pm 0.6]$ | $\mathbf{12.3} \pm 0.3\ [\pm 0.5]$ |
| merge-v0 / noise=0.25 | $11.9 \pm 0.4\ [\pm 0.5]$ | $\mathbf{12.5} \pm 0.4\ [\pm 0.5]$ |
| merge-v0 / noise=0.5 | $12.0 \pm 0.4\ [\pm 0.6]$ | $\mathbf{12.6} \pm 0.2\ [\pm 0.3]$ |

**Training Curves** In Figure 28 to Figure 31, we report the training curves for the RL agent training with both the BT-baseline and ResponseRank variants. The ResponseRank reward models improve learning performance in most cases. However, the results for the *Swimmer-v5* environment show very high variance for the ResponseRank reward models. Note that shaded areas in the training curve plots indicate standard deviation, not 95% CI as in most other plots. This indicates training stability and complements the 95% CIs used in Figure 27.

**Discussion of RL Results** Our results provide a strong first indication that the improved reward modeling performance translates to downstream RL performance. However, we acknowledge that this investigation is limited so far and requires additional experiments, ideally across more scenarios and environments. Larger scale validation in the control domain remains important future work.

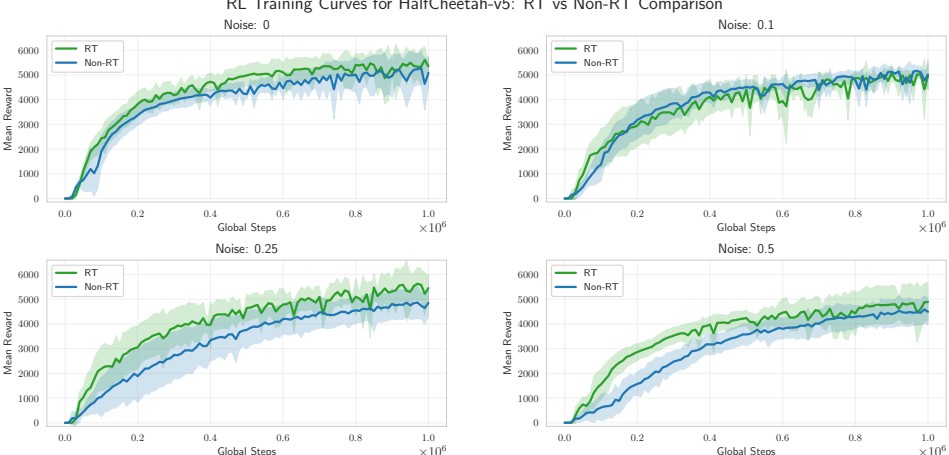

Figure 28: Training curves for *HalfCheetah-v5* across different noise levels. Showing mean and std across 5 seeds.

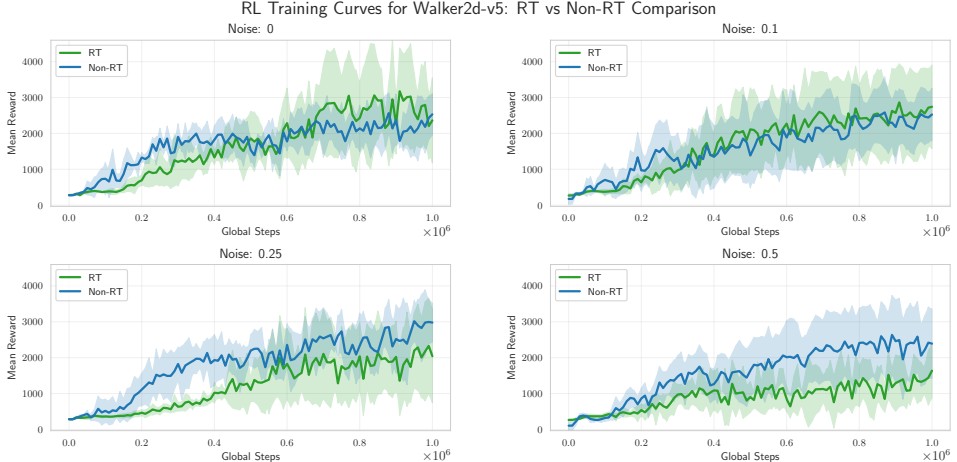

Figure 29: Training curves for *Walker2d-v5* across different noise levels. Showing mean and std across 5 seeds.

# F  Background on response times and preference learning

This section discusses the foundational models and empirical evidence that support using response times (RTs) as a signal for preference strength. It also addresses the complexities inherent in this relationship and draws conceptual parallels to learning strength in other contexts.

## F.1  Foundational support for RTs as preference signals

The premise that RTs offer valuable signals for inferring preference strength is supported by a body of theoretical work, cognitive modeling, and empirical evidence. Humans demonstrably infer others' preference strength from RTs in various contexts, from simple inference tasks [43, 44] to strategic interactions like bargaining games [45].

A key theoretical justification is that decision time can reveal latent aspects of preferences beyond the choice outcome itself [46, 47]. Alós-Ferrer et al. [47], for instance, provide formal grounding, demonstrating through analysis of RT distributions that preferences $(u(x) \geq u(y))$ can be learned with weaker noise assumptions than choice-only data typically require. Their findings indicate that RTs can help recover basic preferences, generalize preference order under symmetric noise assumptions, and even generalize relative choice probabilities under stricter Fechnerian noise assumptions. This

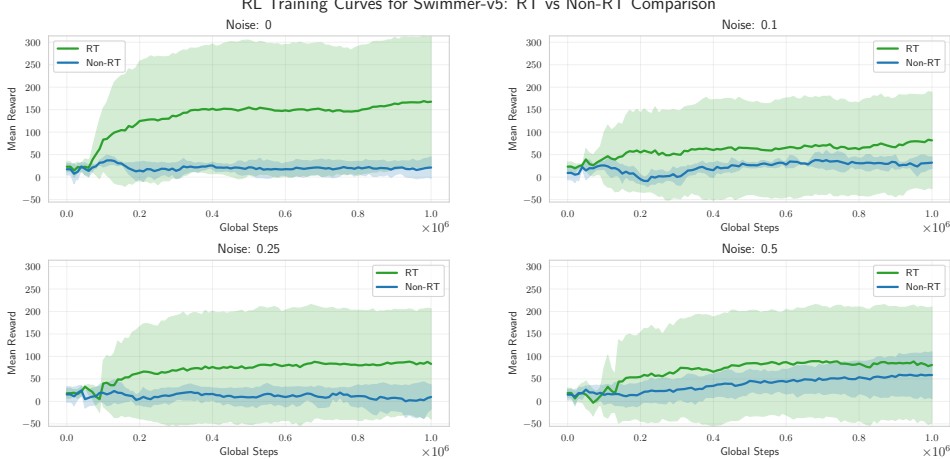

Figure 30: Training curves for *Swimmer-v5* across different noise levels. Showing mean and std across 5 seeds.

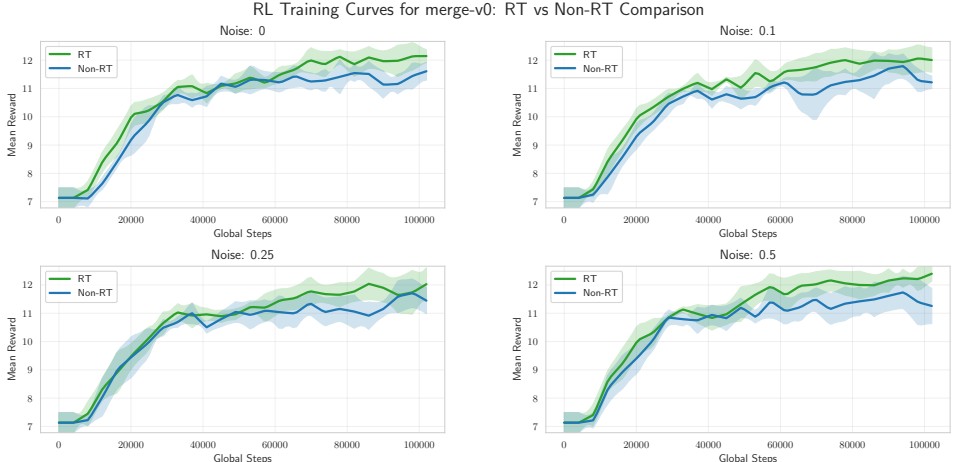

Figure 31: Training curves for *merge-v0* across different noise levels. Showing mean and std across 5 seeds.

informational content of RTs is further highlighted by empirical findings showing that RTs often correlate with latent utility differences learned by models trained only on binary comparisons (implicitly learning strength, as discussed in Appendix A), suggesting RTs reflect these underlying differences [21] and may help learn them more data-efficiently. While ResponseRank does not estimate full RT distributions, this fundamental insight that RTs inherently contain information about underlying utility differences supports their use as a signal for preference strength.

The drift-diffusion model (DDM) [48, 33] provides a well-supported cognitive model explaining this link. It describes the decision process as a stochastic accumulation of evidence over time until a decision threshold is reached. The DDM posits that stronger preferences lead to faster evidence accumulation (higher drift rate) and thus quicker decisions. This inverse relationship is observed empirically; for example, faster responses frequently indicate stronger preferences in tasks like food choice [7], while more cognitive effort (longer RTs) is common when decision-makers are close to indifference in decisions under risk [8]. In value-based DDM applications [7], the *drift rate* $v$, indicating evidence accumulation speed and direction, is often modeled as a linear function of the utility difference between two options, $v = \beta(u_a - u_b)$. Here, $u_a$ and $u_b$ are item utilities, and $\beta$ is a sensitivity parameter. This model predicts that larger utility differences (stronger preferences) result in higher drift rates and, consequently, faster RTs, providing a theoretical justification for using RTs as a proxy for preference strength.

Moreover, the ability to learn preference strength, as opposed to just ordinal preferences, has been shown to enhance model generalization and robustness. From a theoretical point of view, assuming the existence of item utilities allows for generalization to unseen comparisons. Utility differences, often implied by RTs, provide additional structural information about the underlying utility function that aids learning beyond simple preference order. For instance, incorporating RTs to better model utility differences can lead to improved out-of-sample prediction in choice tasks [3] and better performance in preference-based bandit settings [2]. Humans similarly use RT-implied utility differences to infer outcomes of unseen comparisons beyond simple transitivity [43]. Understanding preference strength is also critical for robust value alignment, particularly when choice probabilities are near degenerate values (0 or 1), where outcomes become highly sensitive to small perturbations in learned utilities [4]. These benefits underscore the value of methods capable of extracting cardinal preference information.

## F.2 Complexities and confounding factors in using RTs

Despite foundational support, the relationship between RT and preference strength is complex and subject to numerous confounding influences. Within the DDM framework itself, observed RTs are affected not only by drift rate (utility difference) but also by parameters like decision threshold (caution), non-decision time (e.g., stimulus encoding), and evidence accumulation starting point. These parameters can vary significantly across individuals (e.g., differing annotator speeds) and tasks.

External contextual influences further complicate the picture. User fatigue, experience and boredom, or variations in financial incentives can impact RTs [8]. Task complexity is a major factor; for instance, RTs can be dominated by the reading time required for lengthy texts or by the inherent difficulty of discriminating between complex stimuli [9]. This overall variability makes absolute RTs globally noisy indicators of preference strength.

The nature of the RT-strength relationship can also be nuanced. Factors like decision difficulty, uncertainty, and cognitive load can mean slower RTs sometimes correlate with stronger deliberation over highly valued but conflicting options [49]. Campbell et al. [49] highlights that RT is linked not only to utility differences but also to error variance and cognitive processing strategies. The use of RTs extends to various contexts for inferring cognitive processes; for example, Montero-Porras et al. [50] show that in some settings of the iterated Prisoner's Dilemma, extended deliberation times (slower RTs) are associated with higher cooperation rates, indicating RTs can signal underlying strategy or intent – further showing the complex nature of the signal.

Furthermore, the typical inverse RT-strength link may not hold universally. Under certain high-variance experimental conditions, increased deliberation time might not lead to improved (or even worsen) decision accuracy [51]. In strategic settings, individuals might consciously alter their RTs to influence an observer's perception, irrespective of their true preference strength [45]. Even under ideal conditions, the marginal information that RTs provide beyond choice consistency varies with preference strength: RTs are most informative when preferences are strong and choices alone are near-degenerate, but contribute less when preferences are weak and choices already vary informatively [2].

The design of ResponseRank is fundamentally motivated by the need to navigate these complexities. By using *relative* RTs within carefully constructed, homogeneous strata (e.g., per-annotator or per-session), ResponseRank aims to control for many of these DDM-internal and external confounders. The assumption is that within such local contexts, confounding influences are more likely to be constant or vary less, allowing relative RT differences to more reliably reflect preference strength. This approach enables ResponseRank to learn utility differences robustly without explicitly modeling the full DDM or the exact, complex relationship between absolute RTs and preference strength across diverse settings and individuals. Moreover, Gates et al. [43] suggest that relative utility differences can often be inferred from RTs without needing strict assumptions about the decision model.

## F.3 Learning strength in classification-like settings

Learning utility functions with choice models like the Bradley–Terry model effectively reduces preference learning to a classification task, where preference strength can be reflected in the classifier's confidence. Techniques in supervised learning that address label certainty or aim to modulate learning based on a kind of 'strength' can offer conceptual parallels to learning preference strength.

Label smoothing is a common technique to prevent overfitting in classification by relaxing hard labels to a target distribution, typically mixing the ground-truth label with a uniform distribution [52]. While this can mitigate overconfidence in predictions, its fixed and static nature implies that the model is encouraged to predict this smoothed target distribution regardless of the input, which might be suboptimal in some cases [53].

Label relaxation provides an alternative by *relaxing* the classification target to a range of permissible values, rather than a single fixed distribution [54]. This approach allows the model to predict labels based on its own learned confidence for a given input, relying on its inductive biases to distinguish between easy (high confidence) and hard (low confidence) examples. The model is not forced to predict a specific distribution but can learn to assign higher confidence (analogous to stronger preference) to certain outcomes based on the data.

While these techniques are not directly about learning from RTs, they address the challenge of moving beyond binary outcomes to incorporate a notion of strength or confidence in a learning process. This shares a conceptual similarity with ResponseRank's goal of extracting preference strength information from relative RTs, rather than solely relying on binary preference choices.

### F.4 Second-order preference learning

ResponseRank's approach to learning preference strength is grounded in the concept of second-order preferences. A *second-order preference relation* is a preference over comparisons themselves, rather than just a preference over individual items. For instance, the preference for $a_1$ over $b_1$ might be judged stronger than the preference for $a_2$ over $b_2$. Assuming an underlying utility function $u(x)$, this implies an ordering over utility *differences*: $|u(a_1) - u(b_1)| > |u(a_2) - u(b_2)|$.

In ResponseRank, we leverage response times (RTs) to induce a ranking over a set of pairwise comparisons. This RT-derived ranking (see Section 2 and Table 1) serves as an empirical proxy for such a second-order preference relation, with faster RTs (for a given choice direction) indicating stronger preference.

The theoretical basis for learning from second-order preferences is well-established. Suppes and Winet [14] demonstrate that, under certain assumptions concerning a second-order preference relation (which they term a *difference relation*), information about the ordering of utility differences is sufficient to identify an underlying utility function $u(x)$ up to a positive linear transformation. ResponseRank aims to recover this cardinal utility function. Presupposing the existence of $u(x)$ (a common starting point in RLHF), these assumptions from Suppes and Winet [14] translate to two main categories of conditions for our context:

1. **Reliable RT Proxy:** Human-generated RTs must consistently reflect the ordering of true latent utility differences. Since RTs are real numbers, any quantitative strength measure derived from them will be numerically comparable and transitive. The core assumption is that this derived order is a faithful proxy for the true order of utility differences.

2. **Structural Integrity of True Latent Preferences:** The underlying true utility function $u(x)$ and its differences are assumed to possess structural properties ensuring they behave like measures on a continuous, well-ordered scale. Key among these (following Suppes and Winet [14]'s Axioms A6, A9, A10, A11) are that:

   - The set of items $K$ and the utility function $u$ are sufficiently rich (e.g., to allow for approximate midpoints and continuity).
   - Utility differences combine additively and consistently.
   - All non-zero utility differences are Archimedean (i.e., commensurable, without infinitely small or large differences relative to others). While RT-derived strengths are inherently Archimedean (due to RTs being real numbers), the foundational assumption is that this reflects an Archimedean nature of the *true underlying preferences*.

In essence, these structural assumptions ensure that subjective 'preference strength' corresponds to a well-behaved cardinal utility scale. ResponseRank, by learning from RT-derived rankings, operates on the premise that RTs are a sufficiently faithful proxy to allow recovery of a utility model $u_\theta(x)$ that reflects these underlying principles of $u(x)$.

A second-order preference relation subsumes a first-order one by $a \succeq_1 b \iff ab \succeq_2 aa$. ResponseRank realizes this by: (1) using the first-order preference $p_i$ to normalize comparisons into a $(w_i, l_i)$ format, and (2) ranking the resultant utility differences $s_\theta(w_i, l_i)$ against a *virtual anchor*. This anchor serves as the conceptual $aa$ element, with $s_\theta(aa) = 0$ (or more precisely, its score in the Plackett-Luce list is 0). Note that this element does not really refer to any object being compared; the score is entirely virtual and not related to any comparison. An anchor element serving a comparable role appears in the multiclass classification approach of Fürnkranz et al. [55], where an 'artificial calibration label' separates relevant from irrelevant labels. Ranking $(w_i, l_i)$ pairs above this zero-score anchor directly ensures $s_\theta(w_i, l_i) > 0$, thereby establishing the correct ordinal preference $u_\theta(w_i) > u_\theta(l_i)$. The predictor's inherent antisymmetry ($s_\theta(x, x) = 0$, $s_\theta(x, y) = -s_\theta(y, x)$) then correctly orients $s_\theta(l_i, w_i)$ automatically. Thus, a single Plackett-Luce loss, operating on these RT-ordered, correctly-signed utility differences $s_\theta(w_i, l_i)$ and the anchor, learns both preference order and strength simultaneously.

## G   Proof of reduction to BT

*Proof for Theorem 1.* The theorem considers the special case where an ResponseRank stratum contains only a single comparison. Let this comparison be $q$, where item $a$ is preferred to item $b$. Following the ResponseRank methodology (as described in Section 2), this comparison $q$ is normalized to $(a, b)$ (winner first, loser second). Its RT-derived ranking places it above the virtual anchor element, $\lambda_0$. Thus, the target ranking for the Plackett-Luce (PL) model is the ordered list $[q, \lambda_0]$.

The PL model assigns scores to the elements being ranked.

- The score for the comparison $q = (a, b)$ (representing its preference strength) is the predicted utility difference $s_\theta(q) = u_\theta(a) - u_\theta(b)$. For brevity, let's denote this as $s_{ab}$.

- The score for the virtual anchor $\lambda_0$ is fixed at $s(\lambda_0) = 0$.

The probability of observing the target ranking $[q, \lambda_0]$ under the Plackett-Luce model is calculated as the product of probabilities of selecting each item in order from the remaining set:

$$P_{\text{PL}}([q, \lambda_0]) = \underbrace{\frac{\exp(s_{ab})}{\exp(s_{ab}) + \exp(s(\lambda_0))}}_{\text{Prob. of } q \text{ chosen first from } \{q, \lambda_0\}} \cdot \underbrace{\frac{\exp(s(\lambda_0))}{\exp(s(\lambda_0))}}_{\text{Prob. of } \lambda_0 \text{ chosen next from } \{\lambda_0\}} \tag{3}$$

$$= \frac{\exp(s_{ab})}{\exp(s_{ab}) + \exp(0)} \cdot 1 \tag{4}$$

$$= \frac{\exp(s_{ab})}{\exp(s_{ab}) + 1} \tag{5}$$

Now, substituting $s_{ab} = u_\theta(a) - u_\theta(b)$ (the predicted utility difference):

$$P_{\text{PL}}([q, \lambda_0]) = \frac{\exp(u_\theta(a) - u_\theta(b))}{1 + \exp(u_\theta(a) - u_\theta(b))} \tag{6}$$

$$= \frac{\exp(u_\theta(a))/\exp(u_\theta(b))}{1 + (\exp(u_\theta(a))/\exp(u_\theta(b)))} \tag{7}$$

$$= \frac{\exp(u_\theta(a))}{\exp(u_\theta(b)) + \exp(u_\theta(a))} \tag{8}$$

This final expression (8) is precisely the probability that item $a$ is preferred to item $b$ (denoted $a \succ b$) under the BT model, $P_{\text{BT}}(a \succ b)$.

Optimizing the parameters $\theta$ to maximize likelihood is equivalent to maximizing the log-likelihood (LL). For a single observed preference, $y \in \{0, 1\}$, where $y = 1$ if $a \succ b$ and $y = 0$ if $b \succ a$, and $p = P_{\text{BT}}(a \succ b)$, the negative log-likelihood is:

$$\text{NLL} = -[y \log(p) + (1 - y) \log(1 - p)]. \tag{9}$$

This NLL is, by definition, the binary cross-entropy between the true (observed) preference distribution (represented by $y$) and the model's predicted probability distribution (represented by $p$). Thus, maximizing $P_{\text{PL}}([q, \lambda_0])$ is equivalent to minimizing the binary cross-entropy loss for $P_{\text{BT}}(a \succ b)$.

This completes the proof. $\qquad\qquad\qquad\qquad\qquad\qquad\qquad\qquad\qquad\qquad\qquad\qquad\qquad\square$

# H Extended discussion of the Pearson distance correlation

This appendix provides a more detailed discussion of the Pearson Distance Correlation (PDC) metric, complementing Section 3 in the main text. We elaborate on its motivation, desirable properties, its validation via synthetic experiments, estimation from finite data, formal proofs of its properties, and its limitations.

## H.1 Reliance on true utilities

A practical challenge for PDC is its reliance on true utilities $u$ (yielding true differences $\Delta U$), which are often unavailable in real-world datasets. When a large dataset is accessible, we propose an approximation: train a BT model on its *entirety*, using the derived utility function $u_{\text{BT}}$ as a proxy ground-truth. This $u_{\text{BT}}$ then allows computing an approximate PDC for other models (e.g., ResponseRank) trained on *smaller data subsets*. This approach is justified because BT utilities are identified up to a constant additive shift in log space [24]; thus, with sufficient data, the absolute utility differences will closely approximate the true $\Delta U$ (further discussed in Appendix A). For model training it would be preferable to directly use the model trained on more data, but this method enables PDC-based evaluation of distance-learning on real-world data during method development. This setup specifically tests how much preference strength can be extracted from (in this case artificially) limited data using informative signals.

## H.2 Motivation for a dedicated preference strength metric

Common metrics like calibration error (e.g., True Calibration Error, TCE, based on $l_1$ distance between predicted confidences and true likelihoods [25, 56, 57]) might seem relevant, especially when evaluating models like Bradley-Terry that link utility differences to choice probabilities (akin to classification confidence). However, such metrics are often insufficient for the specific purpose of isolating how well a model has learned *cardinal distance* information (i.e., preference strength). They typically lack key properties desirable for this task:

- They often do not possess *affine invariance*. Scaling utilities directly changes choice probabilities derived from them, thereby affecting calibration metrics. However, since only relative preference *distances* (up to a positive scaling factor) influence expected utility maximization, the evaluation of learned distance should be robust to such utility transformations. Intuitively, if we rank policies by expected total reward, affine transformations of these preserve ordering and therefore do not change the optimal policy in an RLHF context.

- They usually lack *ordinal independence*, as they tend to conflate ordinal accuracy (which item is preferred) with the accuracy of probability estimates (which are tied to utility differences). A model with perfect ordinal accuracy but poor probability estimation (due to miscalibrated distances) would be penalized, making it hard to isolate the quality of distance learning.

- They typically do not provide a clear *baseline* for zero distance learning that is independent of ordinal performance.

To address these limitations, a metric for preference strength should ideally satisfy several key properties. The Pearson Distance Correlation (PDC), as introduced in Section 3, is designed with these in mind:

1. **Distance Sensitivity.** The metric should increase monotonically as the model's predictions more accurately capture information about the true differences (distances) in utility. It should specifically reflect the quality of the learned magnitudes of these differences.

2. **Affine Invariance.** The metric should be unchanged by positive affine transformations of the utility function (i.e., $u'(x) = au(x) + b$ for $a > 0$). This is important because such

transformations preserve the underlying preference structure (and thus relative distances), and only these relative distances are relevant for expected utility maximization, the ultimate goal of RLHF. Hence, a metric should not penalize a different, but equally good, scaling of utilities. Further, scaling alters choice probabilities derived from the utilities, as this derivation generally involves arbitrary scaling factors (e.g., temperature in a softmax function), factors that should not influence the assessment of learned distance.

3. **Known Baseline and Scale.** The metric must have well-defined values corresponding to no learning of utility distances (e.g., 0, indicating predicted magnitudes are uncorrelated with true ones) and perfect learning (e.g., 1), respectively. This enables hypothesis testing about the presence of distance learning and standardized comparison across models, independent of their ordinal accuracy.

4. **Ordinal Independence.** The metric's baseline value (indicating no distance learning) should be achievable even if the model perfectly predicts ordinal preferences (i.e., correctly identifies which item is preferred) but contains no information about the magnitude of these differences. This property is critical for distinguishing models that genuinely capture utility distances from those that merely predict accurate ordinal rankings.

PDC's design allows it to reveal insights, such as a Bradley-Terry model implicitly learning significant distance information from ordinal feedback, which would be difficult to ascertain robustly with standard calibration metrics due to their conflation of ordinal and cardinal performance.

**Related approaches.** Gleave et al. [58] similarly leverage Pearson correlation's affine invariance to compare reward functions, correlating either canonically shaped rewards (EPIC) or episode returns (ERC). While sharing much of the motivation, PDC differs by correlating absolute utility differences, explicitly discarding ordinal information to achieve ordinal independence and supplement existing accuracy metrics.

### H.3 Formal properties and proofs

This section provides more formal statements and outlines the proofs for the properties of PDC, elaborating on  from the main text.

**Theorem 2** (PDC Properties - Formal Statement). *Given true utility function $u$ and predicted utility function $\hat{u}$, let $X, X'$ be independent items sampled i.i.d. from a data distribution $P_{\text{data}}$. Define the true absolute utility differences as $\Delta U = |u(X) - u(X')|$ and the predicted absolute utility differences as $\Delta \hat{U} = |\hat{u}(X) - \hat{u}(X')|$. Assuming $\mathbb{V}[\Delta U] > 0$ and $\mathbb{V}[\Delta \hat{U}] > 0$, the Pearson Distance Correlation (PDC), $\rho_{\text{PDC}}(u, \hat{u}) = \text{Corr}(\Delta U, \Delta \hat{U})$, satisfies:*

1. ***Affine Invariance:** For any constants $c_1, c_2 \in \mathbb{R}$ and $s_1, s_2 > 0$: Let $u'(X) = s_1 u(X) + c_1$ and $\hat{u}'(X) = s_2 \hat{u}(X) + c_2$. Then $\Delta U' = s_1 \Delta U$ and $\Delta \hat{U}' = s_2 \Delta \hat{U}$. It follows that $\rho_{\text{PDC}}(u', \hat{u}') = \rho_{\text{PDC}}(u, \hat{u})$.*

2. ***Distance Sensitivity:** Let $\hat{u}_1$ and $\hat{u}_2$ be two predicted utility functions, yielding $\Delta \hat{U}_1 = |\hat{u}_1(X) - \hat{u}_1(X')|$ and $\Delta \hat{U}_2 = |\hat{u}_2(X) - \hat{u}_2(X')|$. If $\Delta \hat{U}_1$ provides a "better" positive linear approximation to $\Delta U$ than $\Delta \hat{U}_2$ does, then $\rho_{\text{PDC}}(u, \hat{u}_1) \geq \rho_{\text{PDC}}(u, \hat{u}_2)$. "Better" here means that the variance of the residuals from an optimal linear regression of $\Delta \hat{U}_k$ on $\Delta U$ (i.e., $\mathbb{V}[\Delta \hat{U}_k - (a_k \Delta U + b_k)]$ for optimal $a_k > 0, b_k$) is smaller for $k = 1$ than for $k = 2$, relative to $\mathbb{V}[\Delta \hat{U}_1]$ and $\mathbb{V}[\Delta \hat{U}_2]$ respectively.*

3. ***Known Baseline and Scale:** $\rho_{\text{PDC}}(u, \hat{u}) = 0$ if $\Delta U$ and $\Delta \hat{U}$ are uncorrelated. $\rho_{\text{PDC}}(u, \hat{u}) = 1$ if $\Delta \hat{U}$ is a perfect positive linear transformation of $\Delta U$ (i.e., $\Delta \hat{U} = a \Delta U + b$ with $a > 0$). $\rho_{\text{PDC}}(u, \hat{u}) = -1$ if $\Delta \hat{U}$ is a perfect negative linear transformation of $\Delta U$ (i.e., $\Delta \hat{U} = a \Delta U + b$ with $a < 0$).*

4. ***Ordinal Independence:** Let $\hat{u}_1$ and $\hat{u}_2$ be two predicted utility functions. If, for any pair $(X, X')$ drawn from $P_{\text{data}} \times P_{\text{data}}$, it holds that $|\hat{u}_1(X) - \hat{u}_1(X')| = |\hat{u}_2(X) - \hat{u}_2(X')|$ (i.e., the magnitude predictions are identical), then $\rho_{\text{PDC}}(u, \hat{u}_1) = \rho_{\text{PDC}}(u, \hat{u}_2)$.*

5. ***Range** $-1 \leq \rho_{\text{PDC}}(u, \hat{u}) \leq 1$.*

*Outline of Proofs for Theorem 2.* The PDC is defined as $\mathrm{Corr}(\Delta U, \Delta \hat{U})$. Its properties thus largely follow directly from standard properties of the Pearson correlation coefficient.

1. **Affine Invariance:** Given $u'(X) = s_1 u(X) + c_1$ and $\hat{u}'(X) = s_2 \hat{u}(X) + c_2$ with $s_1, s_2 > 0$. The transformed absolute differences are $\Delta U' = |u'(X) - u'(X')| = |s_1(u(X) - u(X'))| = s_1 \Delta U$ (since $s_1 > 0$), and similarly $\Delta \hat{U}' = s_2 \Delta \hat{U}$. The Pearson correlation coefficient $\mathrm{Corr}(A, B)$ is invariant to positive linear scaling of its arguments, i.e., $\mathrm{Corr}(s_1 A, s_2 B) = \mathrm{Corr}(A, B)$ for $s_1, s_2 > 0$ [cf. 59]. Thus, $\rho_{\mathrm{PDC}}(u', \hat{u}') = \mathrm{Corr}(s_1 \Delta U, s_2 \Delta \hat{U}) = \mathrm{Corr}(\Delta U, \Delta \hat{U}) = \rho_{\mathrm{PDC}}(u, \hat{u})$.

2. **Distance Sensitivity:** The PDC, being $\mathrm{Corr}(\Delta U, \Delta \hat{U}_k)$, directly reflects the goodness of the linear fit between these two variables. The square of the Pearson correlation is the coefficient of determination ($R^2$), which for a regression of $\Delta \hat{U}_k$ on $\Delta U$ is $1 - \mathbb{V}[\text{residuals}]/\mathbb{V}[\Delta \hat{U}_k]$. The coefficient of determination quantifies the proportion of the variance in one variable that is predictable from the other variable in a linear regression, thus reflecting the goodness of linear fit [60, 61]. A "better" positive linear approximation as defined in Theorem 2 (i.e., a smaller relative residual variance, with assumed regression coefficient $a_k > 0$) implies a higher $R^2$. Consequently, this leads to a higher $\rho_{\mathrm{PDC}}(u, \hat{u}_k)$ value.

3. **Known Baseline and Scale:** These are direct consequences of standard Pearson correlation properties: $\mathrm{Corr}(A, B) = 0$ if $A$ and $B$ are uncorrelated. $\mathrm{Corr}(A, B) = 1$ if $B = aA + b$ with $a > 0$, and $\mathrm{Corr}(A, B) = -1$ if $B = aA + b$ with $a < 0$ [62]. PDC inherits these directly with $A = \Delta U$ and $B = \Delta \hat{U}$.

4. **Ordinal Independence:** If $|\hat{u}_1(X) - \hat{u}_1(X')| = |\hat{u}_2(X) - \hat{u}_2(X')|$ for all pairs $(X, X')$, then the random variables for predicted absolute differences, $\Delta \hat{U}_1$ and $\Delta \hat{U}_2$, are identical. By definition, $\rho_{\mathrm{PDC}}(u, \hat{u}_1) = \mathrm{Corr}(\Delta U, \Delta \hat{U}_1)$ and $\rho_{\mathrm{PDC}}(u, \hat{u}_2) = \mathrm{Corr}(\Delta U, \Delta \hat{U}_2)$. Since $\Delta \hat{U}_1 = \Delta \hat{U}_2$, it immediately follows that $\rho_{\mathrm{PDC}}(u, \hat{u}_1) = \rho_{\mathrm{PDC}}(u, \hat{u}_2)$.

5. **Range:** The Pearson correlation coefficient $\mathrm{Corr}(A, B)$ always lies in the interval $[-1, 1]$ [62]. As PDC is defined as such a correlation, it directly inherits this range.

$\square$

**Corollary 1** (PDC baseline for no distance learning). *If predicted utility difference magnitudes $|\hat{u}(X) - \hat{u}(X')|$ are statistically independent of true utility difference magnitudes $|u(X) - u(X')|$ (even if signs of the utility differences are preserved), then the PDC satisfies:*

$$\rho_{\mathrm{PDC}}(u, \hat{u}, p_{\mathrm{data}}) = 0. \tag{10}$$

*This establishes a baseline of 0 when no meaningful distance information is learned, providing a basis for hypothesis tests.*

### H.4  Synthetic experiment validating PDC properties

Figure 2 in the main text (Section 3) compares PDC and TCE under simulated prediction degradation to validate PDC's properties. The experiment starts with perfect predictions ($\hat{u} = u$) based on synthetic utility data ($N = 1000$ items, random utilities with a normal distribution). Degradation is applied by:

1. Randomly flipping the sign of the predicted utility difference for a fraction $f_{\mathrm{sign}}$ of item pairs (introducing ordinal errors, y-axis of the heatmaps).

2. Shuffling the magnitudes (absolute values) of the predicted utility differences for a fraction $f_{\mathrm{mag}}$ of item pairs (destroying distance information, x-axis of the heatmaps).

This process is performed for both the original utilities $u$ and affine-transformed utilities $u' = 2u + 5$.

The figure demonstrates PDC's properties as follows:

- **Ordinal Independence:** Observed along the left edge ($f_{\mathrm{mag}} = 0$) of the PDC heatmaps. Even as $f_{\mathrm{sign}}$ increases (moving up the y-axis, introducing more ordinal errors), PDC remains

at its maximum value (blue), indicating perfect correlation of magnitudes. This demonstrates that PDC's assessment of distance learning is unaffected by purely ordinal errors when true distance information is perfectly preserved. This contrasts sharply with TCE, which worsens (increases) with $f_{\text{sign}}$.

- **Distance Sensitivity:** Observed as one moves from left ($f_{\text{mag}} = 0$) to right ($f_{\text{mag}} = 1$) across the PDC heatmaps. PDC systematically decreases (plots become less blue/more red) as $f_{\text{mag}}$ increases, reflecting its sensitivity to the progressive loss of true magnitude information caused by shuffling.

- **Affine Invariance:** Demonstrated by comparing the left PDC panel (original $u$) with the right PDC panel (affine-scaled $u' = 2u + 5$). The heatmaps are nearly identical, confirming that PDC's evaluation is robust to positive affine transformations of the underlying utility function. TCE, conversely, shows different patterns for $u$ and $u'$, highlighting its lack of affine invariance. Note that technically the desirable property is invariance to affine transformations of the utility *distance*. Scaling of utilities directly translates to scaling of distances, however, shifting utilities cancels out, and shifting of distances is not meaningful due to the anchoring at 0.

- **Known Baseline and Scale:** The PDC heatmaps show a clear scale: optimal (blue, approximately 1) when $f_{\text{mag}} = 0$ (perfect distance information, irrespective of $f_{\text{sign}}$), and a baseline (red, approximately 0) when $f_{\text{mag}} = 1$ (all distance information destroyed, irrespective of $f_{\text{sign}}$). This aligns with the theoretical properties.

- **Contrast with TCE:** The TCE plots in Figure 2 clearly lack these desirable properties. TCE increases (worsens) with *both* sign flips ($f_{\text{sign}} > 0$) and magnitude shuffling ($f_{\text{mag}} > 0$), thereby conflating ordinal and distance errors. Furthermore, its sensitivity to affine scaling (evident from the differing left and right TCE panels) and its lack of a clear, ordinally-independent baseline for distance learning make interpretations of TCE regarding learned *strength* ambiguous.

This empirical validation highlights the PDC's ability to specifically and robustly quantify learned utility distance information, fulfilling the design goals outlined in Appendix H.2 and Section 3.

### H.5 Estimation of PDC from sample data

In practical scenarios, the PDC (Definition 1) is estimated from a finite dataset of pairwise comparisons. Let $\mathcal{D}_{\text{pairs}} = \{(a_i, b_i)\}_{i=1}^{N}$ denote $N$ item-pairs sampled i.i.d. from $p_{\text{data}} \times p_{\text{data}}$. The *sample PDC* is computed as the sample Pearson correlation coefficient:

$$\hat{\rho}_{\text{PDC}}(u, \hat{u}; \mathcal{D}_{\text{pairs}}) = \frac{\sum_{i=1}^{N} \left( \Delta U_i - \overline{\Delta U} \right) \left( \Delta \hat{U}_i - \overline{\Delta \hat{U}} \right)}{\sqrt{\sum_{i=1}^{N} \left( \Delta U_i - \overline{\Delta U} \right)^2 \sum_{i=1}^{N} \left( \Delta \hat{U}_i - \overline{\Delta \hat{U}} \right)^2}}, \tag{11}$$

where $\Delta U_i = |u(a_i) - u(b_i)|$ and $\Delta \hat{U}_i = |\hat{u}(a_i) - \hat{u}(b_i)|$ are the observed true and predicted absolute utility differences for the $i$-th pair, and $\overline{\Delta U}, \overline{\Delta \hat{U}}$ denote their respective sample means. While the sample PDC, $\hat{\rho}_{\text{PDC}}$, is not generally an unbiased estimator of the population PDC, $\rho_{\text{PDC}}$, it is a consistent estimator under standard assumptions [62, Section 9.7].

## I  Baselines

This appendix details the baseline and control methods used for comparison against ResponseRank in the experiments (Section 4).

**Bradley–Terry (BT).** This baseline represents standard RLHF approaches that learn utility functions solely from preference labels, without considering response times. As a *response-time unaware* baseline, we use the Bradley–Terry model [20] to learn the utility function from the preference labels $p_i$ alone. The Bradley–Terry model is a special case of the Plackett–Luce model discussed in Section 2 for two alternatives. Unlike ResponseRank, which ranks entire comparisons, the BT model as used here ranks individual items (e.g., $a_i$ or $b_i$). The model learns a utility function $u$ that maps

each item to a scalar utility value. It assumes that the probability of choosing item $a_i$ over item $b_i$ is given by the softmax function

$$p(a_i \succ b_i) = \frac{e^{u(a_i)}}{e^{u(a_i)} + e^{u(b_i)}}. \tag{12}$$

This baseline helps establish the performance level achievable using only the ordinal preference information inherent in choices.

**Response Time Regression (RtRegression).** The RtRegression baseline represents methods that *directly model RTs* by assuming a specific, known, and global functional relationship between RT and preference strength. It is a regression-based approach that directly models the response times $\tau_i$ as a function of the utility differences. It assumes $\tau_i = \mathrm{link}(u(a_i) - u(b_i))$, where $\mathrm{link}$ is a specific function mapping utility differences to response times. It further requires $\mathrm{link}$ to be known and invertible, which is a strong assumption often not met in practice. Given this assumption, the response times $\tau_i$ give access to the unsigned utility differences via $|u(a_i) - u(b_i)| = \mathrm{link}^{-1}(\tau_i)$. To model cardinal preference strength *and* direction jointly, we convert this to a signed difference using the preference label $p_i$. We define $\mathrm{sign}(p_i) = +1$ if $a_i$ is preferred (e.g., $p_i = 0$), and $\mathrm{sign}(p_i) = -1$ if $b_i$ is preferred (e.g., $p_i = 1$). The regression target is then:

$$\Delta \hat{u}_i = \mathrm{sign}(p_i) \cdot \mathrm{link}^{-1}(\tau_i). \tag{13}$$

This signed difference $\Delta \hat{u}_i$ forms the regression target. A neural network $u_\theta$ is trained with an MSE loss to predict these signed differences. The network's prediction for a pair $(a_i, b_i)$ is the strength predictor $s_\theta(a_i, b_i) = u_\theta(a_i) - u_\theta(b_i)$. Minimizing $\sum_i \big(s_\theta(a_i, b_i) - \Delta \hat{u}_i\big)^2$ yields the parameters $\theta$ for our utility function $u_\theta$. In our experiments, we assume a hyperbolic link function (Equation (14)) that *exactly matches* the data-generating process in some synthetic datasets.

$$\mathrm{link}(\Delta u; \mathrm{rt}_{\min}, \mathrm{rt}_{\max}) := \mathrm{rt}_{\min} + \frac{\mathrm{rt}_{\max} - \mathrm{rt}_{\min}}{|\Delta u| + 1}. \tag{14}$$

This specific link function is invertible (if the sign of $\Delta u$ is known), inversely proportional to $|\Delta u|$, and bounded between $\mathrm{rt}_{\min}$ and $\mathrm{rt}_{\max}$. While informed by the general observation of an inverse RT-strength relationship [43], the specific choice of Equation (14) is illustrative and by no means unique; the requirement to assume *some* explicit link function is a primary weakness of this baseline approach.

Critically, in contrast to ResponseRank, this RtRegression approach relies on the *absolute* RT values (processed via the assumed inverse link function). It makes strong global assumptions about the *exact* functional form relating RT and preference strength. ResponseRank avoids these assumptions by using only the *relative* order of RTs within carefully constructed strata (Section 2). This reliance on relative ranks within strata provides robustness when the exact global RT-strength relationship is unknown or varies significantly across contexts or individuals.

**Permutation Controls (ResponseRank-Perm and RtRegression-Perm).** These controls verify that performance gains stem from informative RT signals, rather than from the structure of the loss or model itself. The core mechanism involves randomly permuting the response times $\tau_i$ across the dataset to get $\tau_{\pi(i)}$, where $\pi$ is a random permutation. This isolates the contribution of RT magnitude information while preserving the ordinal preference direction. We include two such controls based on our primary RT-aware methods:

- **ResponseRank-Perm:** Applies the ResponseRank ranking method but uses the permuted times $\tau_{\pi(i)}$ when constructing the magnitude component of the sort keys $k_i$. The preference label $p_i$ still determines the key's directional component.

- **RtRegression-Perm:** Applies the RtRegression method but uses the permuted times $\tau_{\pi(i)}$ as input to the inverse link function, i.e., the regression target becomes $\Delta \hat{u}_i' = \mathrm{sign}(p_i) \cdot \mathrm{link}^{-1}(\tau_{\pi(i)})$. Similarly here the sign of the inferred utility strength remains untouched by the permutation.

A significant drop in performance (especially in PDC, Section 3) for these controls compared to their non-permuted versions (ResponseRank and RtRegression) confirms that the original methods successfully leveraged meaningful information encoded in the timing of human responses.

**Stratification ablation (ResponseRank-Pool).** This ablation investigates the specific benefit of *stratification* within the ResponseRank framework. We consider ResponseRank-Pool, a variant of ResponseRank that ablates the stratification step. This variant omits stratification based on metadata $m_i$ (Section 2). Instead, comparisons are grouped into random batches of size $b$ for ranking. All other aspects of the ResponseRank method (e.g., learning utility differences via Plackett-Luce loss on comparison ranks) remain the same. By comparing the performance of ResponseRank to ResponseRank-Pool, we can isolate the performance impact specifically attributable to the strategic use of stratification to control for confounders. If ResponseRank significantly outperforms ResponseRank-Pool, it validates the effectiveness of the stratification strategy.

# J  Extended discussion of related work

This appendix provides further context and detailed comparisons related to the main paper's discussion of related work, covering foundational perspectives on response times and prior computational methods leveraging RTs or other strength signals.

## J.1  Detailed comparison with prior RT modeling approaches

Section 7 briefly introduced prior methods integrating response times (RTs) into preference learning. Here, we provide a more detailed comparison highlighting the technical distinctions between these methods and our ResponseRank approach, particularly concerning their underlying assumptions and applicability to large-scale RLHF.

Prior methods that integrate RTs often employ explicit cognitive process models like the Drift Diffusion Model (DDM) [e.g., 21, 2] or make strong assumptions about a direct, *global* relationship between RT magnitude and preference strength [e.g., 22].

## J.2  Ranking-based utility learning

Methods like SeqRank [63] and LiRE [64] also leverage ranking structures in RLHF. Both construct preference rankings of items from pairwise feedback and derive overlapping preference pairs (e.g., $A \succ B$, $B \succ C$, $A \succ C$) to train with Bradley–Terry loss. This imposes soft constraints on utilities and thereby conveys implicit strength information [64].

ResponseRank differs in two key aspects: (1) it ranks *comparisons* of items rather than items themselves, and (2) it uses external strength signals rather than preference feedback to structure the rankings. This reflects different objectives: SeqRank and LiRE improve feedback efficiency through querying strategies and data augmentation, requiring control over the queries posed to the annotator, while ResponseRank focuses on loss formulation for leveraging auxiliary strength information.

## J.3  Explicit strength specification

While ResponseRank leverages implicit signals of preference strength, other works take the approach of explicitly eliciting strength information from users through modified feedback mechanisms.

Several approaches directly request strength annotation during preference collection. Wilde et al. [5] proposes *scale feedback*, where users provide a slider-based response $\psi \in [-1, 1]$ for each comparison, where the magnitude encodes preference strength proportional to reward difference. Similarly, Jansen et al. [22] propose a *strength elicitation* method that requires users to explicitly assign preference strength labels during comparisons.

Other approaches ask users to specify the *relative* strength of preferences between multiple comparisons. DistQ [65] requires users to select which of two comparisons is easier to distinguish, then judge that pair. While conceptually related to ResponseRank, DistQ requires explicit meta-cognition about comparison difficulty and uses a composite pairwise loss instead of our joint ranking loss. Future work could explore combining ResponseRank's loss with DistQ's explicit distinguishability queries or vice versa. Another approach, taken by Papadimitriou and Brown [66], asks users to group items into preference tiers (e.g., good, acceptable, dangerous) and then explicitly specify the relative margins between these tiers (e.g., acceptable – dangerous is twice as large as good – acceptable).

All these approaches recognize the importance of preference strength, but they all require explicit user annotation of strength. ResponseRank instead focuses on leveraging *implicit* response metadata without modifications to the feedback collection process, making it broadly applicable and able to reuse existing datasets.

## J.4 Adaptive losses

While ResponseRank leverages explicit external strength signals, complementary work explores using adaptive losses to allow the learner to more robustly learn preference strength implicitly. Hong et al. [67] propose Ada-Pref, which learns instance-specific loss scaling factors for each preference pair via distributionally robust optimization. Pairs with ambiguous preferences receive smaller scaling factors, allowing the reward model to learn smaller reward differences; pairs with clear preferences receive larger factors, enabling larger learned differences. This adaptivity emerges implicitly, similar to implicit strength learning in non-adaptive BT discussed in Appendix A.

Ada-Pref and ResponseRank address complementary scenarios. ResponseRank excels when domain-specific strength signals are readily available and locally valid. Ada-Pref suits settings where no reliable strength signal exists, but preference data contains implicit variation in confidence or difficulty. Future work may explore combining these two approaches, though ResponseRank's ranking construction may already capture much of the implicit learning Ada-Pref achieves.

## J.5 Approaches based on explicit cognitive models

A significant line of research incorporates RTs by employing explicit cognitive process models, primarily variants of the DDM. These models aim to provide a mechanistic account of how latent utility differences jointly influence generated choices and RTs, enabling learning from both signals.

Shvartsman et al. [21], for example, propose integrating a DDM within a Gaussian Process (GP) framework for preference learning, primarily targeting the low-data regime and reporting significant improvements in sample efficiency. Their method links the GP's latent utility difference to the DDM's drift rate. A key challenge is that the joint choice-RT likelihood of the DDM is analytically intractable, and common numerical approximations are non-differentiable. They tackle this challenge by approximating the DDM using moment-matching with a family of parametric skewed distributions, which allows them to optimize the GP hyperparameters by minimizing the approximate negative log marginal likelihood of both choices and RTs within a DDM-inspired likelihood framework.

Similarly grounded in cognitive modeling, Li et al. [2] use the simplified, computationally tractable dEZDM variant [68] to incorporate RTs into preference-based linear bandits, focusing on fixed-budget best-arm identification. Their core mechanism relies on an analytical property of the dEZDM linking the ratio of *expected* choice and *expected decision time* directly to the scaled utility difference, where the utility function is assumed linear within their bandit framework. They estimate these scaled linear parameters via linear regression, using empirical averages of choices ($\bar{C}_x$) and decision times ($\bar{t}_x$) computed over multiple *identical query repetitions* as the target value $\bar{C}_x/\bar{t}_x$, and integrate this into the Generalized Successive Elimination (GSE) algorithm. This approach requires averaging over *repeated identical queries* to estimate these expectations ($\bar{C}_x$, $\bar{t}_x$) and assumes *utility linearity* within a bandit setting, using $\bar{C}_x/\bar{t}_x$ as the target for linear regression to estimate scaled linear parameters.

ResponseRank contrasts significantly with these DDM-based approaches. Firstly, it entirely avoids specifying or fitting an explicit cognitive model of the RT generation process. This sidesteps the complexities associated with DDM likelihood calculations, their approximations, and the potential fragility of assuming a specific cognitive process accurately describes diverse human behavior across different tasks and contexts. Secondly, the RT signal used is fundamentally different. ResponseRank relies solely on the relative rank order of total observed RTs within specific strata as a noisy, non-parametric indicator of preference strength rank. It does not attempt to link RT magnitudes directly to a model parameter like drift rate, nor does it need to isolate a theoretical "decision time" from the measured total RT. Consequently, ResponseRank circumvents the restrictive data requirements seen in some prior work, such as the need for repeated identical queries or assumptions of utility linearity [2], making it directly applicable to standard RLHF datasets typically consisting of unique comparisons.

### J.6 Other RT integration strategies

Other methods avoid fitting full cognitive models directly or integrate RTs in different ways.

Shvartsman et al. [21], in addition to their DDM-based approach, also propose *prediction stacking*. This is a two-stage approach where a separate model is first trained to predict RTs based on item features. These predicted RTs then serve as additional *input features* for a choice model (a final GP model that predicts choice probabilities), potentially informing it about decision difficulty without relying on the DDM's generative assumptions for the choice model itself. RT predictions might act as a form of regularization, modulating the influence of choices based on predicted difficulty. ResponseRank differs as it uses observed (not predicted) RTs and integrates their rank information directly into the training objective via the ranking loss, rather than using RTs as input features.

Aiming to elicit complex preference systems ($[A, R_1, R_2]$), with $R_2$ representing preference strength, over a *fixed item set $A$*, Jansen et al. [22] propose *time elicitation*. After collecting ordinal preferences ($R_1$) via pairwise comparisons, they measure the user's response time for each pair. Crucially, the method then *deterministically constructs* the cardinal relation ($R_2$) representing relative preference strength. This construction is based directly on the RT *magnitudes*, assuming these perfectly represent preference exchange strength. This direct construction of a *preference system* from assumed clean response times over a fixed set $A$ differs fundamentally from *learning a utility function* from potentially noisy, total response times over structured objects in an open world, as is typical in RLHF settings. Nonetheless, their direct construction of $R_2$ partially motivates our construction of the target ranking relation in ResponseRank.

### J.7 Core distinctions

While the methods discussed above offer diverse ways to leverage RTs, ResponseRank is distinct in its minimal and localized assumptions. Primarily, ResponseRank diverges in the assumed scope of the RT-strength correlation. Unlike prior works that often implicitly or explicitly assume a *global* correlation (faster RTs consistently indicate stronger preferences across the dataset), ResponseRank posits a weaker, *local* assumption. This local view holds that the inverse RT-strength correlation is reliable only *within* specific, carefully constructed strata (e.g., per-annotator or per-session), which are designed to control for confounding factors like annotator effects or task complexity. This local approach is crucial for robustness, enabling effective learning from pooled data across diverse annotators and contexts where global RT comparisons would be unreliable.

Furthermore, ResponseRank does not specify or fit an explicit cognitive model (like DDM) to describe the joint generation of choices and RTs. This design choice avoids the complexities of DDM likelihood calculations and their approximations (cf. 21, 2), and the potential fragility of assuming a single cognitive process accurately describes diverse human behavior. Consequently, rather than using RT magnitudes directly as model inputs, as features (e.g., in prediction stacking; 21), or for deterministic rule construction from raw RTs (e.g., in time elicitation; 22), ResponseRank utilizes the relative rank order of total observed RTs within these strata. This ordinal information from RTs serves as a noisy, non-parametric proxy for the rank order of preference strength, without attempting to model precise RT values or link them to specific model parameters like a DDM's drift rate.

These characteristics enable ResponseRank to circumvent many restrictive data requirements common in prior work, such as the need for repeated identical queries [2], assumptions of utility linearity [2], operations on a fixed item set [22], or assumptions about 'clean' RT magnitudes [22]. By leveraging *relative RT ranks within strata* via a ranking loss (like Plackett-Luce), ResponseRank is designed to bypass these requirements. This offers robustness and direct applicability to typical large-scale RLHF data, which often involves unique comparisons, non-linear utilities, and noisy total response times from diverse sources. The method aims to filter out systemic variations and noise from confounding factors (individual speed, fatigue, task complexity) without explicit modeling of these factors or relying on pristine RT magnitudes.

## K    Compute resources

We conducted the synthetic experiments on a single compute node with 8 CPU cores and 32 GB of RAM. They take approximately 17 hours to complete in parallel on this hardware. The MultiPref

experiments were run on A100 and H100 GPUs, with a full-size (full training set fraction) run taking approximately 1.5h on a single H100 GPU.

The RL experiments were conducted on a compute cluster. The control experiments had four separate stages: (1) Training of the baseline RL models for collecting checkpoints, (2) Generating feedback datasets based on rollouts from checkpoints, (3) Training of reward models with feedback datasets, and (4) training of downstream RL models with reward models. Steps 1,2, and 4 were conducted on CPU nodes, with each individual run being allocated 2 CPU cores and 16GB RAM. Training of RL agents takes around 1 hour, collection of feedback around 30 minutes. Reward model training was performed on nodes with access to a A100 and H100 GPU shared across four parallel runs. Reward model training took approximately 30 minutes. In total, 20 baseline runs and 20 feedback datasets were generated. To reproduce the results, 240 reward model trainings and RL down-stream training runs are required (with shared GPUs) this results in approx. 30 GPU hours and 480 CPU hours.

## L   Software and data assets

This work uses the following external assets:

- **MultiPref** dataset (version 1.0) [15]: `https://huggingface.co/datasets/allenai/multipref`, licensed under ODC-BY.
- **Llama-3.1-8B-Instruct** [35]: `https://huggingface.co/meta-llama/Llama-3.1-8B-Instruct`, licensed under the Llama 3.1 Community License.
- **MuJoCo** v5 environments (HalfCheetah, Swimmer, Walker2d) [37]: `https://github.com/google-deepmind/mujoco`, licensed under Apache 2.0.
- **highway-env** merge-v0 [38]: `https://github.com/Farama-Foundation/HighwayEnv`, licensed under MIT.

We release the following assets at `https://github.com/timokau/response-rank`, documented in the repository README:

- Experiment source code, licensed under MIT.
- Numerical results for synthetic, RL, and LLM experiments.

