# OpenReview forum: "ResponseRank: Data-Efficient Reward Modeling through Preference Strength Learning"
_NeurIPS.cc/2025/Conference — NeurIPS 2025 poster_

### Official Review · Reviewer_WFhw · 2025-06-26

**Clarity:** 4
**Significance:** 3
**Originality:** 3
**Rating:** 5
**Confidence:** 3

**Summary:**

This paper introduces RtRank, a method for learning *preference strength* from response times (RTs) in the context of reinforcement learning from human feedback (RLHF). Traditional RLHF relies on binary preferences: user specifies which item is preferred, not by how much. RtRank proposes to incorporate such an additional signal without requiring further annotation (a binary preference is still elicited) by leveraging response time (RT) of the preference elicitation process. The core hypothesis is that low(er) RT corresponds to a stronger preference; because RTs are typically noisy, the authors propose leveraging *relative* RTs within homogeneous strata (e.g., defined by annotator or session) to infer the strength of preferences. The method uses a Plackett–Luce-based model to learn utility differences and introduces a new metric, Pearson Distance Correlation (PDC), designed to measure learned utility magnitude correctness, independent of the preference-prediction correctness. The paper presents empirical results on synthetic datasets under various noise and generation conditions.

**Questions:**

1. Can you comment on the feasibility and challenges of applying RtRank to real-world datasets? Are you planning or conducting such evaluations? What pitfalls do you expect when scaling RtRank to real-world data?
2. How sensitive is the method to poor stratification or suboptimal strata sizes? Could automated or adaptive stratification be a viable extension? The backbone of the work is an assumption that correlation between RT and preference-strenght is high in “optimal” stratas; I believe it would strengthen the work if such discussion is included.
3. Beyond RTs, have you though about other implicit indicators of preference strength? Would it make sense to extend RtRank to take into account multiple indicators (assuming they exist in the dataset)?

**Ethical Concerns:**

["NO or VERY MINOR ethics concerns only"]

**Final Justification:**

The authors have addressed my general questions and concerns with clarifications and new experiments. While these additions improved my understanding of the method, I believe further validation on real-world data is needed to fully establish its impact. I will therefore maintain my score.

**Limitations:**

Yes.

**Paper Formatting Concerns:**

I have not noticed major formatting issues.

**Quality:**

3

**Strengths And Weaknesses:**

### Strengths

- The idea of stratified RT-based ranking for capturing preference strength is well-motivated and novel in this context (as per my current knowledge of the relevant literature).
- The proposed PDC metric is compelling and, when used in conjunction with additional metrics, may provide more granular insight into the learned models.
- Preliminary results on synthetic datasets are likewise compelling.
- Paper is clearly written and easy to follow.

### Weaknesses

- It would be great to see real-world evaluations, though it is understandable that the paper presents a first proof-of-concept.
- The method’s performance depends on strata quality; deeper analysis of stratification choices would strengthen the work (see also my question below).

---

> ### Author Response · Authors · 2025-08-01
> **Rebuttal**
>
> Thank you for your review and constructive feedback. Due to inclusion of the two new experiments, we unfortunately failed to finalize all responses before the original rebuttal deadline. We apologize for the delay and hope you will consider our reply nonetheless.
>
> > **Weakness 1: It would be great to see real-world evaluations, though it is understandable that the paper presents a first proof-of-concept.**
>
> We agree real-world evaluations would strengthen the paper. We are constrained by limited publicly available response time datasets: Existing datasets from prior work are typically small with single evaluators, unsuited for demonstrating our method's strengths. As an approximation, we report MultiPref results using weighted inter-annotator agreement for strength rankings.
>
> **MultiPref experiments.** We trained language reward models on MultiPref and evaluated on RewardBench v1 and v2, with v2 showing strong correlation with downstream performance [1]. More details in our response to jccL. Results across training sizes (10 seeds, differences to BT baseline in parentheses):
>
> | Train samples (fraction) | Test set accuracy | RewardBench v1  | RewardBench v2  |
> |-------------------------:|------------------:|----------------:|----------------:|
> | 3271 (0.4)               | 74.7 **(+3.0)**   | 77.4 **(+5.4)** | 49.0 **(+3.9)** |
> | 4907 (0.6)               | 74.6 **(+3.2)**   | 77.1 **(+0.8)** | 54.4 **(+4.8)** |
> | 6540 (0.8)               | 74.1 **(+2.5)**   | 79.3 **(+1.6)** | 56.1 **(+3.8)** |
> | 7359 (0.9)               | 74.2 **(+3.6)**   | 76.0 (-1.8)     | 52.3 (-0.3)     |
> | 8179 (1.0)               | 74.7 **(+3.7)**   | 78.1 (-1.4)     | 56.1 **(+1.3)** |
>
> **RL experiments.** We conducted experiments on classical RL tasks (HalfCheetah, Swimmer, Walker2d, Highway merge), with RtRank commonly outperforming the BT baseline, particularly in low-noise scenarios. Detailed results in response to reviewer jccL.
>
>
> > **Weakness 2: The method’s performance depends on strata quality; deeper analysis of stratification choices would strengthen the work (see also my question below).**
>
> We agree, but unfortunately this is limited by real-world data availability. Both new experiments do not rely on data-based stratification, as this is unnecessary for agreement-based or oracle-based preference strength rankings. We encourage future work on datasets supporting more realistic experiments.
>
> > **Question 1: Can you comment on the feasibility and challenges of applying RtRank to real-world datasets? Are you planning or conducting such evaluations? What pitfalls do you expect when scaling RtRank to real-world data?**
>
> We believe real-world application is highly feasible. Computationally, RtRank could scale to large-scale LLM posttraining pipelines. Key challenges include stratification techniques and implementation constraints: entire rankings must be co-located in single on-device batches, limiting maximum stratum size and requiring non-iid batches. We will include a discussion of this in the paper.
>
> As demonstrated by our new experiments, these technical limitations are manageable: short rankings convey substantial strength information, and non-iid batches are already common for grouping similar-length texts. The primary requirement is access to high-quality response time datasets.
>
> > **Question 2: How sensitive is the method to poor stratification or suboptimal strata sizes? Could automated or adaptive stratification be a viable extension? The backbone of the work is an assumption that correlation between RT and preference-strength is high in “optimal” stratas; I believe it would strengthen the work if such discussion is included.**
>
> The RtRank-Perm baseline addresses this by shuffling response times, simulating poor stratification with no strength signal. Results show that while performance degrades, the model still learns ordinal preferences reasonably well. Suboptimal stratification would likely provide more signal than -Perm baseline. We will add discussion of stratum quality dependence to the paper.
>
> **Stratification robustness**: Our method includes built-in protections: (1) single comparisons reduce to Bradley-Terry (Theorem 1), preserving preference learning and (2) Plackett-Luce loss handles ranking noise robustly.
>
> **Adaptive stratification** could be viable! Future work in this direction may prove promising.
>
> > **Question 3: Beyond RTs, have you thought about other implicit indicators of preference strength? Would it make sense to extend RtRank to take into account multiple indicators (assuming they exist in the dataset)?**
>
> Certainly! RtRank is flexible and can learn from any relative preference strength information. Response time is natural but could be extended by other factors such as mouse movement or gaze patterns, or replaced by other decision difficulty estimates.
>
> *[1] Malik et al., "RewardBench 2: Advancing Reward Model Evaluation", arXiv preprint, 2025.*

---

> > ### Comment · Reviewer_WFhw · 2025-08-01
> >
> > I thank the authors for their response. The added experiments and clarifications improved my understanding, and I will take them into account in my final assessment.

---

> > > ### Author Response · Authors · 2025-08-04
> > >
> > > Thank you again for your constructive feedback and for taking the time to review our rebuttal.

---

### Official Review · Reviewer_LCxR · 2025-07-01

**Clarity:** 3
**Significance:** 3
**Originality:** 3
**Rating:** 5
**Confidence:** 4

**Summary:**

This paper seeks to use response times during RLHF to learn a better utility model. The main idea is that response times can be ranked to provide a ranking loss that gives a signal about a user's strength of preference. The paper proposes RtRank that looks at relative response times within different groups (strata) of preference annotations to better model preference strength. Results show that the proposed approach outperforms methods that do not use response time or stratified modeling.

**Questions:**

Missing related work on learning based on preference strength: Papadimitriou et al.. "Bayesian Constraint Inference from User Demonstrations Based on Margin-Respecting Preference Models." ICRA 2024.

Line 92: How do you ensure that task difficulty is relatively constant? It seems that for any user, the task difficulty will vary by query.

Is it possible to also include a measure of fatigue when learning from response times? For example, what if the metadata for each preference contained the order it was answered or how long the annotator had been working before they supplied a preference label.

It is not clear from the text why you need an anchor element. Would the proposed approach work as long as the learned utility differences satisfy the relative ranking?

Fig 2: How is PCE unaffected by ordinal error? Shouldn't it perform worse with more noisy data

How do you know the learned rewards are better? Prior work (e.g., Tien et al. "Causal Confusion and Reward Misidentification in Preference-Based Reward Learning." ICLR. 2023.) has shown that accuracy on a held out set is not always indicative of downstream RL performance.

**Ethical Concerns:**

["NO or VERY MINOR ethics concerns only"]

**Final Justification:**

The authors have addressed most of my concerns.  I still think doing actual experiments with real human response times is needed to understand the true benefits of this approach, but I agree that the simulated results are promising and I appreciate the new results described in the authors rebuttal. The paper studies an interesting topic and has nice simulation results which is common for NeurIPS so I will keep my score.

**Limitations:**

I would have liked to see actual human experiments. It seems that the focus is very much human factors-oriented (focusing on response times, strata, etc) but then all the experiments are synthetic making the merit of the work unclear.

**Quality:**

3

**Strengths And Weaknesses:**

Strengths:

+Good motivation for learning from response times

+good synthetic experiments that compare with baselines and test ablations, robustness, and sensitivity

+well written paper with nice figures and tables

Weaknesses:

-This paper focuses on human response times but contains no actual human data.

-No justification for the synthetic data creation or why it is a good model of human data

-No results showing down-stream performance using the learned reward models.

---

> ### Author Response · Authors · 2025-08-01
> **Rebuttal**
>
> Thank you for recognizing the strength of our contribution and highlighting areas for improvement! Due to inclusion of the two new experiments, we failed to finalize all responses before the original rebuttal deadline. We apologize for the delay and hope you will consider our reply nonetheless.
>
> **Weaknesses**
>
> > **This paper focuses on human response times but contains no actual human data.**
>
> We acknowledge this limitation. To partially address this, we conducted a new experiment using the MultiPref dataset - initially chosen as the only publicly available large-scale preference dataset with response time information. However, the reported response times are conflated with extensive metadata annotation (graded preferences across helpful/truthful/harmless categories plus justifications). We therefore generated strength rankings from multi-annotator agreement weighted by self-reported preference strength. While not using response times directly, this experiment is a proof of concept of our method's applicability to real language modeling data. We encourage future research with cleanly collected response times.
>
> **Language experiments.** We trained language reward models on MultiPref and evaluated on RewardBench v1 and v2, with v2 showing strong correlation with downstream performance [1]. We use inter-annotator agreement for strength rankings. Results across training sizes (10 seeds, differences to BT baseline in parentheses):
>
>
> | Train samples (fraction) | Test set accuracy | RewardBench v1 | RewardBench v2 |
> |-------------------------:|------------------:|----------------:|----------------:|
> | 3271 (0.4) | 74.7 **(+3.0)** | 77.4 **(+5.4)** | 49.0 **(+3.9)** |
> | 4907 (0.6) | 74.6 **(+3.2)** | 77.1 **(+0.8)** | 54.4 **(+4.8)** |
> | 6540 (0.8) | 74.1 **(+2.5)** | 79.3 **(+1.6)** | 56.1 **(+3.8)** |
> | 7359 (0.9) | 74.2 **(+3.6)** | 76.0 (-1.8) | 52.3 (-0.3) |
> | 8179 (1.0) | 74.7 **(+3.7)** | 78.1 (-1.4) | 56.1 **(+1.3)** |
>
>
> *Experimental details*: Training on MultiPref subsets after withholding test sets, filtering texts >1024 tokens (~1400 test samples). Fine-tuned from Llama-3.1-8B-Instruct with optimal hyperparameters: 3 epochs, batch size 64, learning rate 15e-6, linear LR scheduler (0.05 warmup), gradient clipping 1.0, weight decay 0.1. For RtRank, strength rankings computed via annotator agreement weighted by reported strength, constructing balanced strata with no ties (mostly size 4).
>
>
> > **No justification for the synthetic data creation or why it is a good model of human data**
>
> Our method makes minimal assumptions about human behavior. It requires only that (1) preferences approximately follow some utility function (standard Bradley-Terry assumption) and (2) we can approximately rank preference groups by strength. **The only assumption about response times is an inverse monotonic relationship with preference strength.**
>
> Our synthetic data reflects these properties: preferences generated from utility functions and response times following inverse monotonic relationships (supported by psychology literature [2]). One of our settings follows the drift-diffusion model, which has extensive psychological validation [3], providing a model of RT that goes beyond the monotonic relationship. We choose base utility functions randomly, which while not aligned with specific human preferences, is reasonable since the model should learn any utility function matching our two key assumptions.
>
> We will clarify this in the paper.
>
> > **No results showing down-stream performance using the learned reward models.**
>
> We agree such results would strengthen the paper. The MultiPref experiments partially address this, as RewardBench v2 performance is strongly correlated with down-stream tasks [1]. For a more direct evaluation, we added RL experiments probing downstream performance on control tasks.
>
> **RL experiments.** We conducted control domain experiments using established ground-truth reward functions to synthesize preferences and strength rankings. We tested three Mujoco environments (HalfCheetah, Swimmer, Walker2d) and Highway merge-v0, comparing Bradley-Terry baseline with RtRank using 5000 queries per training run (5 seeds each). RtRank often outperforms BT models, achieving higher accuracy by utilizing additional information. While performance degrades slightly under preference and response time noise, RtRank generally demonstrates robust performance, showing that reward models trained with RtRank often translate to better downstream performance. Detailed results are in our response to reviewer jccL.
>
> *[1] Malik et al., "RewardBench 2: Advancing Reward Model Evaluation", arXiv preprint, 2025.*
>
> *[2] Milosavljevic et al., "The Drift Diffusion Model can account for the accuracy and reaction time of value-based choices under high and low time pressure." Judgment and Decision Making, 2010.*
>
> *[3] Ratcliff and McKoon, "The Diffusion Decision Model: Theory and Data for Two-Choice Decision Tasks." Neural computation, 2008.*

---

> > ### Author Response · Authors · 2025-08-01
> >
> > **Response to questions**
> >
> > > **Missing related work on learning based on preference strength: Papadimitriou et al.. "Bayesian Constraint Inference from User Demonstrations Based on Margin-Respecting Preference Models." ICRA 2024.**
> >
> > Thank you for bringing this to our attention! This is certainly related, as part of their focus is also learning preference strength. While the goal is shared, the setting is different, requiring direct margin specification and collecting preferences over groups of items. We will discuss this in our related work.
> >
> >
> > > **Line 92: How do you ensure that task difficulty is relatively constant? It seems that for any user, the task difficulty will vary by query.**
> >
> > Task difficulty is very difficult to measure. Inter-user differences can be captured by stratification, which can be hierarchical (first on annotator, then on estimated difficulty). This leaves the problem of estimating this difficulty, however. Luckily **rtrank does not require a numerical estimate or even an ordering of difficulty, just a clustering valid for a single annotator**. Semantical embedding similarity may serve as a proxy for this, though in general stratification will likely be domain-dependent.
> >
> >
> > > **Is it possible to also include a measure of fatigue when learning from response times? For example, what if the metadata for each preference contained the order it was answered or how long the annotator had been working before they supplied a preference label.**
> >
> > Absolutely! The rtrank loss makes no assumptions on how strata are created, and much domain knowledge can be encoded this way. Estimated fatigue could be one factor in stratum creation.
> >
> >
> > > **It is not clear from the text why you need an anchor element. Would the proposed approach work as long as the learned utility differences satisfy the relative ranking?**
> >
> > **The anchor element encodes preference information.** Rankings capture preference *strength*, ordering utility differences by magnitude. The anchor element ensures all comparisons rank above zero utility, pushing their utilities positive. Since we normalize comparisons to (chosen, rejected) format, positive utility differences align with user choices.
> >
> > Without anchors, the model could theoretically minimize loss by inverting all preferences while preserving difference ordering. We will clarify this motivation in the paper.
> >
> >
> > > **Fig 2: How is PCE unaffected by ordinal error? Shouldn't it perform worse with more noisy data**
> >
> > PDC isolates distance information by taking absolute differences, discarding ordinal information. This may be slightly unintuitive, but results in a metric with interpretable extreme values, ideally complementing the information already provided by the accuracy: 0 (no distance information) and 1 (perfect distance information).
> >
> >
> > > **How do you know the learned rewards are better? Prior work (e.g., Tien et al. "Causal Confusion and Reward Misidentification in Preference-Based Reward Learning." ICLR. 2023.) has shown that accuracy on a held out set is not always indicative of downstream RL performance.**
> >
> > You raise an important point about generalization failures in preference-based reward learning. We expect RtRank to help address these issues as **strength information acts as regularization**, downweighting weak (potentially noisy) preferences while emphasizing strong ones. This regularization effect should promote more generalizable and robust reward models by reducing the influence of inconsistent or misleading preference data that could lead to reward misidentification. Our language modeling experiments further validate this, as RtRank improves RewardBench v2 performance which has been shown to strongly correlate with downstream tasks [4].
> >
> > *[4] Malik et al., "RewardBench 2: Advancing Reward Model Evaluation", arXiv preprint, 2025.*

---

> > > ### Comment · Reviewer_LCxR · 2025-08-04
> > >
> > > Thank you for your responses. I still think doing actual experiments with real human response times is needed to understand the true benefits of this approach, but I agree that the simulated results are promising and I appreciate the new results described in the authors rebuttal. The paper studies an interesting topic and has nice simulation results which is common for NeurIPS so I will keep my score.

---

> > > > ### Author Response · Authors · 2025-08-05
> > > >
> > > > Thank you for considering our response and for maintaining your positive evaluation. We appreciate your recognition that our simulation results are promising and that the paper addresses an interesting topic.
> > > >
> > > > We fully agree that real human response time data would provide valuable additional validation, and we hope such data will become available in the future. In the meantime, we believe our simulation experiments demonstrate the method's potential, particularly given its flexibility to adapt to other strength signals beyond RT.
> > > >
> > > > We are grateful for your feedback and continued support.

---

### Official Review · Reviewer_jccL · 2025-07-03

**Clarity:** 2
**Significance:** 2
**Originality:** 2
**Rating:** 4
**Confidence:** 2

**Summary:**

This paper utilizes the response time of the human labeler to leverage the strength of the preferences. The authors say that the short response time indicates the stronger preference, and using such preference strength, the utility model is trained to quantify the preference strength. Furthermore, by introducing the Pearson Distance Correlation to evaluate how the model learns the preference strength, in the experiment, the proposed method shows higher PDC and accuracy than other baselines.

**Questions:**

Already mentioned in the Weaknesses section.

**Ethical Concerns:**

["NO or VERY MINOR ethics concerns only"]

**Final Justification:**

Since the authors addressed my concern, I increase my score to 4.

**Limitations:**

Yes

**Quality:**

2

**Strengths And Weaknesses:**

Strengths

1. This paper uses additional information, response time, to quantify the strength of the user preference, and show that the effectiveness of proposed method compared to the standard Bradley-Terry model.


Weaknesses

1. First, the reason for the effectiveness of leveraging the response time to quantify the preference strength is not clear. Why the response time can indicate the preference strength? Is there any toy experiments on showing the effectiveness of use response?

2. There is no reinforcement learning experiment using the proposed method. Since the high accuracy or PDC do not directly reflect the better performance of RL agent, it would be better to show the performance of RL agent using the proposed reward model.

3. More comparison should be performed with other state-of-the-art baselines such as sequential pair-wise comparison methods [1] or list-wise comparison methods [2]. The authors compared the proposed method only with standard models.

[1] Hwang et. al., Sequential Preference Ranking for Efficient Reinforcement Learning from Human Feedback, NeurIPS, 2023

[2] Choi et. al., Listwise Reward Estimation for Offline Preference-based Reinforcement Learning, ICML, 2024

---

> ### Author Rebuttal · Authors · 2025-07-31
>
> Thank you for your review and your insightful comments, which will help us improve the paper. We appreciate that you recognize the strength of our method, but also appreciate the weaknesses identified which we will discuss in the following.
>
> > **1. First, the reason for the effectiveness of leveraging the response time to quantify the preference strength is not clear. Why the response time can indicate the preference strength? Is there any toy experiments on showing the effectiveness of use response?**
>
> **Foundational support for RT.** Psychology research demonstrates that response time is inversely correlated with preference strength [1]. When options are similar in utility, humans require more time to analyze differences and estimate each option's worth accurately. This principle underlies the drift-diffusion model, which has extensive empirical support. We discuss this thoroughly in Appendix A.1 (third paragraph) and Section 5 ("Learning from RTs"), with additional details in Appendix G.1.
>
> **Caveats of RT.** Response times require clean data and careful stratification. We initially planned experiments using MultiPref dataset response times but found them unusable due to confounding time taken for metadata annotation. Unfortunately, we lack access to clean real-world RT datasets. However, our MultiPref experiments using agreement-based strength rankings provide strong support for real-world applicability when preference strength rankings are available.
>
> **RtRank beyond RT.** Crucially, our core contribution, the rtrank loss, is not restricted to response times. It can incorporate any secondary information about preference strength, requiring only ordinal and local comparisons (no globally comparable values). Beyond response times, we could use mouse movement, gaze information, domain knowledge, or directly stated strength information commonly collected but underutilized in preference datasets.
>
> > **2. There is no reinforcement learning experiment using the proposed method. Since the high accuracy or PDC do not directly reflect the better performance of RL agent, it would be better to show the performance of RL agent using the proposed reward model.**
>
> **RL experiments.** We conducted additional control domain experiments using established ground-truth reward functions to synthesize preferences and strength rankings. Following recent methodology [2], we tested three Mujoco environments (HalfCheetah, Swimmer, Walker2d) and Highway merge-v0, comparing Bradley-Terry baseline with RtRank using 5000 queries per training run (5 seeds each). Here, we report both max. and final scores (in brackets the comparison with the BT baseline):
>
>
> | Environment | Max Reward | Final Reward |
> |------------:|-----------:|-------------:|
> | HalfCheetah-v5 | 5866.9 **(+352.3)** | 5215.2 (-219.3) |
> | Swimmer-v5 | 110.0 **(+66.1)** | 98.3 **(+77.1)** |
> | Walker2d-v5 | 3905.1 **(+1088.1)** | 2679.7 **(+313.2)** |
> | merge-v0 | 12.2 **(+1.2)** | 11.4 **(+1.0)** |
>
> RtRank generally outperforms BT models. For the Mujoco environments, we find that RtRank reward models also achieve higher accuracy, in line with improved downstream performance. For merge-v0, RtRank consistently outperforms despite similar reward model accuracy, suggesting better reward shaping.
>
> **Noise robustness.** The results above were measured with perfect environment rewards. To measure robustness to noise, we also benchmarked with different noise levels (according to the methodology outlined in [2]), which perturbs the underlying ground-truth reward and in turn extracted response times and preferences.
> We find that RtRank outperforms the BT-model in low-noise scenarios, and only underperforms when overall RL performance deteriorates significantly. In environments that are generally robust to noise (such as merge-v0 or HalfCheetah), RtRank also stays stable.
>
> | Environment | Max Reward | Final Reward |
> |------------:|------------:|------------:|
> | HalfCheetah-v5 noise=0.0 | 5866.9 **(+352.3)** | 5215.2 (-219.3) |
> | HalfCheetah-v5 noise=0.1 | 5208.3 (-124.3) | 4922.7 (-387.7) |
> | HalfCheetah-v5 noise=0.25 | 5850.7 **(+418.2)** | 5481.4 **(+201.4)** |
> | HalfCheetah-v5 noise=0.5 | 5433.3 **(+155.5)** | 5433.3 **(+439.5)** |
> | Swimmer-v5 noise=0.0 | 110.0 **(+66.1)** | 98.3 **(+77.1)** |
> | Swimmer-v5 noise=0.1 | 94.6 **(+48.9)** | 81.8 **(+49.6)** |
> | Swimmer-v5 noise=0.25 | 36.7 (-4.8) | 9.5 (-12.6) |
> | Swimmer-v5 noise=0.5 | 47.4 (-29.1) | 17.0 (-49.7) |
> | Walker2d-v5 noise=0.0 | 3905.1 **(+1088.1)** | 2679.7 **(+313.2)** |
> | Walker2d-v5 noise=0.1 | 3126.1 **(+298.1)** | 2742.2 **(+216.0)** |
> | Walker2d-v5 noise=0.25 | 2567.7 (-1344.8) | 1919.1 (-1616.6) |
> | Walker2d-v5 noise=0.5 | 1754.5 (-1387.6) | 1621.9 (-1037.9) |
> | merge-v0 noise=0.0 | 12.2 **(+1.2)** | 11.4 **(+1.0)** |
> | merge-v0 noise=0.1 | 12.1 **(+0.9)** | 11.4 **(+0.9)** |
> | merge-v0 noise=0.25 | 11.8 **(+0.5)** | 11.7 **(+1.1)** |
> | merge-v0 noise=0.5 | 12.4 **(+0.8)** | 11.5 **(+0.3)** |
>
>
> **MultiPref experiments.** To demonstrate real-world applicability, we trained language reward models on MultiPref using inter-annotator agreement for strength rankings (lacking reliable RT data). While not a perfect test for RtRank with response times, this serves as proof of concept for our method's applicability to language modeling settings. Results across training sizes:
>
> | Train samples (fraction) | Test set accuracy | RewardBench v1  | RewardBench v2  |
> |-------------------------:|------------------:|----------------:|----------------:|
> | 3271 (0.4)                | 74.7 **(+3.0)** | 77.4 **(+5.4)** | 49.0 **(+3.9)** |
> | 4907 (0.6)                | 74.6 **(+3.2)** | 77.1 **(+0.8)** | 54.4 **(+4.8)** |
> | 6540 (0.8)                | 74.1 **(+2.5)** | 79.3 **(+1.6)** | 56.1 **(+3.8)** |
> | 7359 (0.9)                | 74.2 **(+3.6)** | 76.0 (-1.8)     | 52.3 (-0.3)     |
> | 8179 (1.0)                | 74.7 **(+3.7)** | 78.1 (-1.4)     | 56.1 **(+1.3)** |
>
> We fine-tuned from Llama-3.1-8B-Instruct with optimal hyperparameters found for both variants. Strength rankings used weighted annotator agreement (clear preference: weight 1.0, slight: 0.5). Results demonstrate consistent improvements across most settings.
>
> All sizes are subsets of the total MultiPref dataset after withholding a test set. We filter both sets for texts exceeding 1024 tokens, resulting in ~1400 test samples. We shuffle the dataset with different seeds and report mean performance across 10 seeds. Columns show performance on in-distribution (MultiPref) test set and two out-of-distribution sets: RewardBench v1 and v2 [4,5] (measured per official methodology). Both RtRank and BT use the same optimal hyperparameters: 3 epochs, effective batch size 64 (16 on-device, 4 accumulation steps), learning rate 15e-6, linear LR scheduler with 0.05 warmup ratio, gradient clipping at 1.0, weight decay 0.1, Adam optimizer (0.9/0.999/1e-08). For RtRank, we compute strength rank by weighted annotator agreement, then construct balanced strata with no ties (most strata size 4).
>
> We will include detailed descriptions of these experiments in the paper.
>
> These preliminary results highlight RtRank's potential for downstream tasks. Particularly RewardBench v2 has been shown to correlate strongly with downstream performance [4], further validating our approach.
>
> > **3. More comparison should be performed with other state-of-the-art baselines such as sequential pair-wise comparison methods [1] or list-wise comparison methods [2]. The authors compared the proposed method only with standard models.**
>
> Thank you for highlighting these relevant works! Both Choi et al. and Hwang et al. use transitivity to construct item rankings from pairwise comparisons, then apply Bradley-Terry loss to generated pairs. Their rankings do communicate strength information similar to ours.
>
> However, they differ in two crucial aspects: (1) They construct rankings over *items* while we rank *comparisons*. (2) They **require specialized sampling procedures** while rtrank works with any preference data given local strength judgments (e.g., response times).
>
> This makes direct comparison difficult since their contribution lies primarily in querying procedures and data augmentation, while ours focuses on novel loss formulation. We cannot apply their methods unchanged to our dataset, and implementing their querying strategy would make approaches quite similar. Importantly, the methods are **inherently complementary**.
>
> We will **add discussion** of these approaches, their impact on preference strength learning, and potential complementary usage to our related work section.
>
> ---
>
> *[1] Milosavljevic et al., "The Drift Diffusion Model can account for the accuracy and reaction time of value-based choices under high and low time pressure." Judgment and Decision Making, 2010.*
>
> *[2] Metz et al., Reward Learning from Multiple Feedback Types, ICLR2025*
>
> *[3] Lambert et al., "RewardBench: Evaluating Reward Models for Language Modeling", NAACL findings, 2025.*
>
> *[4] Malik et al., "RewardBench 2: Advancing Reward Model Evaluation", arXiv preprint, 2025.*

---

> > ### Comment · Reviewer_jccL · 2025-08-04
> > **Response to the comment**
> >
> > Thank you for your reply.
> >
> > In the RL experiment, how did you measure the response time? Did you utilize the ground-truth rewards to make the preference label?
> >
> > It would be better to specify the way to train the reward model in RL experiment.

---

> > > ### Author Response · Authors · 2025-08-04
> > >
> > > Thank you for engaging with our rebuttal! To clarify: yes, we use ground-truth environment reward functions to synthesize both preference labels and strength indicators.
> > >
> > > **Preference and strength.** Given a trajectory pair annotated with (optionally perturbed) environment rewards, we mark the higher-reward trajectory as preferred and use the negative reward difference as a pseudo-response-time. Clearer preference differences correspond to lower simulated response times. Note that since only the order of pseudo RTs matters, negative values are allowed, and any preference strength estimate could substitute for response time. We then randomly construct rankings of size 16 (matching our on-device batch size) for reward model training using the RtRank loss.
> > >
> > > **Noise.** To test robustness to reward noise, we also experimented with perturbed rewards. Here, we sample rewards from a truncated Gaussian distribution with mean equal to the retrieved environment reward and scale $\sigma = \alpha \cdot \sigma_r$, where $\alpha$ is the noise factor (from tables) and  $\sigma_r$ is the dataset reward standard deviation. Truncation ensures samples stay within the observed reward range, following prior work [2], though it's not essential to our approach. Higher noise levels reduce both preference accuracy and strength reliability.
> > >
> > > **Training.** We follow methodology from [2]. Expert models (trained via standard RL on ground-truth rewards) generate trajectories sampled across multiple checkpoints, ensuring diverse policy coverage and skill levels for our offline dataset. We sample 5000 segment pairs of length 50 (truncated at episode boundaries) for reward model training. Our reward models are 6-layer MLPs with 256 hidden units, processing concatenated state and action vectors (one-hot encoded for discrete action spaces). We use Adam optimization with weight decay and early stopping on a validation holdout set. For downstream RL evaluation, we use the same hyperparameters from Stable Baselines3 [5] Zoo across all experimental conditions, with frozen reward models serving as predictors.
> > >
> > > *[2] Metz et al., Reward Learning from Multiple Feedback Types, ICLR2025* (numbering from the original rebuttal)
> > > *[5] Raffin et al., Stable-Baselines3: Reliable Reinforcement Learning Implementations, JMLR 2021.*

---

> > > > ### Comment · Reviewer_jccL · 2025-08-05
> > > > **Reply to the comment**
> > > >
> > > > Thank you for giving details on the additional experiment.
> > > >
> > > > After carefully reading the experiment results, the advantage of using response time (RT) with preference label seems effective on RL experiment.
> > > >
> > > > Though the empirical evidence on using RT is still weak in the motivation section, the authors mainly resolve my concern on the reinforcement experiment.
> > > >
> > > > Based on the authors' comment, I will increase the score to 4.

---

> > > > > ### Author Response · Authors · 2025-08-05
> > > > >
> > > > > We are glad that the experiments were able to address your concern on downstream RL applicability. Thank you for again for your openness and engagement!

---

### Official Review · Reviewer_7tw8 · 2025-07-05

**Clarity:** 2
**Significance:** 3
**Originality:** 3
**Rating:** 3
**Confidence:** 3

**Summary:**

This paper considers the degree of preference strength in preference learning. Specifically, the article uses the time intervals during data annotation as an indicator of preference strength, assuming that stronger preferences lead to shorter reaction times. After grouping the annotated data accordingly, they model it using Plackett-Luce. To measure how well reward models capture this degree information, the authors propose the Pearson Distance Correlation (PDC) metric. On synthetic datasets, their approach outperforms the baseline (BT model).

**Questions:**

1. What does the preference pair normalization mentioned in Figure 1 refer to, why is normalization necessary, and how should real data be processed?
2. There are many possible scenarios in real annotation settings. Has the author considered real-world scenarios, such as how to screen for outliers, including more practical data cleaning issues like different comprehension times for different questions?

**Ethical Concerns:**

["NO or VERY MINOR ethics concerns only"]

**Limitations:**

yes

**Quality:**

2

**Strengths And Weaknesses:**

Strengths：
1. The authors use annotation time intervals as an indicator of preference strength, which allows incorporating preference degrees into the scoring information and should theoretically yield better results.
2. Meanwhile, the authors focus on addressing the lack of metrics for evaluating how well preference models capture preference degrees, introducing a measure based on the correlation coefficient between true scores and trained scores.

Weakness：
1. Lacks training and testing results in real-world scenarios. The experimental section of the paper is based on synthetic data, which inherently involves strong assumptions. The actual effectiveness in real preference learning scenarios remains unknown.
2. The PDC metric is difficult to utilize. The use of the PDC metric in real-world scenarios seems to conflict with the method itself. Since we use BT as an approximation for the golden truth in real scenarios, this implies the assumption that BT performs best on large-scale data. However, the paper's method shows better performance than the BT model on synthetic data, which contradicts this assumption.

---

> ### Author Response · Authors · 2025-08-01
> **Rebuttal**
>
> Thank you for your thoughtful review and for recognizing both contributions as strengths. Due to including two new experiments, we unfortunately failed to finalize this response for the original rebuttal deadline. We apologize for the delay and hope you will consider our response addressing each of your points below.
>
> **Response to weaknesses**
>
> > **1. Lacks training and testing results in real-world scenarios. The experimental section of the paper is based on synthetic data, which inherently involves strong assumptions. The actual effectiveness in real preference learning scenarios remains unknown.**
>
> Thank you for raising this point, which has also been echoed by other reviewers. In response, we have conducted two new experiments: (1) Training a reward model on the MultiPref dataset and (2) training a reinforcement learning agent on control environments.
>
> **Multipref.** This dataset provides text preferences with rich metadata and four annotations per comparison. We initially chose this dataset as the only publicly available large-scale preference dataset with response time information. Unfortunately, total response time measures not only preference selection time, but also time for fine-grained graded preferences across multiple categories (helpful, truthful, harmless) and justifications. As this conflates the strength signal, we instead generated strength rankings from multi-annotator agreement rates weighed by self-reported preference strength. While this evaluation does not use RTs, it is grounded in real data and serves as proof of concept for our method's applicability to language modeling settings. We strongly encourage future research with response times collected in more controlled settings.
>
> We trained reward models on MultiPref and evaluated on a held-out test set as well as RewardBench v1 and v2, with v2 correlating strongly with downstream performance [1]. Hyperparameters were tuned for RewardBench v2. Further details in response to reviewer jccL. We observed that optimizing BT involves a tradeoff between RewardBench and test-set accuracy. RtRank seemed more robust, potentially due to regularizing effects of strength. Results across training sizes (10 seeds for different random subsets, mean performance with differences to BT baseline in parentheses):
>
> | Train samples (fraction) | Test set accuracy | RewardBench v1 | RewardBench v2 |
> |-------------------------:|------------------:|----------------:|----------------:|
> | 3271 (0.4) | 74.7 **(+3.0)** | 77.4 **(+5.4)** | 49.0 **(+3.9)** |
> | 4907 (0.6) | 74.6 **(+3.2)** | 77.1 **(+0.8)** | 54.4 **(+4.8)** |
> | 6540 (0.8) | 74.1 **(+2.5)** | 79.3 **(+1.6)** | 56.1 **(+3.8)** |
> | 7359 (0.9) | 74.2 **(+3.6)** | 76.0 (-1.8) | 52.3 (-0.3) |
> | 8179 (1.0) | 74.7 **(+3.7)** | 78.1 (-1.4) | 56.1 **(+1.3)** |
>
> **RL experiments.** We conducted control domain experiments using ground-truth reward functions to synthesize preferences and strength rankings. We tested three Mujoco environments (HalfCheetah, Swimmer, Walker2d) and Highway merge-v0, comparing Bradley-Terry baseline with RtRank using 5000 queries per training run (5 seeds each). We find RtRank often outperforms BT models, particularly in low-noise scenarios. For the Mujoco environments, we find that RtRank reward models also achieve higher accuracy, in line with improved downstream performance. For merge-v0, RtRank consistently outperforms the baseline despite similar reward model accuracy, suggesting better reward shaping. Detailed results are in our response to reviewer jccL.
>
> > **2. The PDC metric is difficult to utilize. The use of the PDC metric in real-world scenarios seems to conflict with the method itself. Since we use BT as an approximation for the golden truth in real scenarios, this implies the assumption that BT performs best on large-scale data. However, the paper's method shows better performance than the BT model on synthetic data, which contradicts this assumption.**
>
> Using BT models on large datasets as PDC ground-truth does not conflict with our method. The Bradley-Terry model theoretically identifies utilities up to shift [2]. With (infinitely) large datasets, it learns “correct” utilities up to shift, which suffices for PDC. RtRank's goal is not greater accuracy with infinite data, but extracting more information from limited data. This motivates training BT on large datasets, then testing sample-efficiency of methods like RtRank on small subsets evaluated against this ground-truth. We will clarify this in the paper.
>
> We nonetheless agree that this can be difficult to use in practice. It suits method development rather than production model evaluation. For final training, you would use all available data. Nonetheless, PDC provides valuable development insights.
>
> *[1] Malik et al., "RewardBench 2: Advancing Reward Model Evaluation", arXiv preprint, 2025.*
>
> *[2] Train, Discrete Choice Methods with Simulation, Cambridge University Press, 2009.*

---

> > ### Author Response · Authors · 2025-08-01
> >
> > **Response to questions**
> >
> > > **1. What does the preference pair normalization mentioned in Figure 1 refer to, why is normalization necessary, and how should real data be processed?**
> >
> > We normalize presentation order so the chosen item always appears first. This requires only preference information and is not specific to synthetic data. It is necessary because the RtRank loss function uses an anchor element to encourage all-positive utility differences. With normalized ordering, it encourages utility differences agreeing with preference labels. The combination of normalization and the anchor element encodes preferences in the loss (comparison rankings additionally encode strength). As stated in Theorem 1, this is equivalent to standard Bradley-Terry loss when only one comparison and anchor element are in the ranking.
> >
> > > **2. There are many possible scenarios in real annotation settings. Has the author considered real-world scenarios, such as how to screen for outliers, including more practical data cleaning issues like different comprehension times for different questions?**
> >
> > **Outlier robustness.** RtRank is robust to outliers since we use only ordinal response time information. Whether response time is twice or 100 times larger than another makes no difference. An outlier can at worst be misplaced in rankings, and Plackett-Luce loss handles such noise robustly, just as Bradley-Terry handles preference noise. Additionally, since our loss reduces to Bradley-Terry with single comparisons, outliers can be screened using standard techniques and placed in single-element strata, effectively using only preference information while discarding response time.
> >
> > **Implementation considerations.** Our loss operates at partition level, requiring entire partitions in single batches. This limits partition size by batch size and necessitates non-iid sampling. To maintain fixed batch sizes, multiple partitions must be packed per batch, potentially requiring partition splitting. These constraints are manageable, as evidenced by our MultiPref and control experiments: rankings of size 8-16 (typical on-device batch sizes) convey substantial information. Non-iid batches are already common practice for grouping similar-length sequences [5, Section 5.1]. Since partitions are placed into batches IID and gradient accumulation ensures multiple partitions per effective batch, training impact is minimal.
> >
> > Thank you for raising these points, we will include a discussion of this in the paper.
> >
> > *[5] Vaswani et al., “Attention Is All You Need”, NeurIPS 2017.*

---

### Comment · Area_Chair_yhVL · 2025-08-04

Dear Reviewers,

Please kindly note that the author-reviewer discussion period has started. Please take a look at the rebuttals and acknowledge the receipt of it (if you have not done so). Meanwhile, you are encouraged to initiate further discussions with the reviewers.

Best,

Your AC

---

### Note · Authors · 2025-08-12

Dear AC and Reviewers,

We are grateful for the constructive reviews and discussion. These final remarks summarize key strengths, outcomes, and manuscript changes addressing reviewer concerns.

**Key strengths recognized by reviewers**
- Clear writing and presentation [**WFhw, LCxR**]
- Novel, well-motivated RtRank method [**WFhw**] using additional information for better preference models [**7tw8, jccL**]
- Strong synthetic experiments with good baselines and ablations [**LCxR**]
- PDC metric addressing evaluation gaps for preference degrees [**7tw8, WFhw**]

**Key improvements addressing concerns**

*Real-world evaluation:* Responding to concerns about synthetic-only validation, we conducted new experiments in **language reward model training** and **reinforcement learning for control tasks** (detailed in rebuttal to jccL). The LLM experiments demonstrate RtRank's effectiveness on real, complex preference data and its adaptability to other preference strength signals. The RL experiments demonstrate translation to better downstream policies and applicability to control domains.

*Acknowledged limitations:* We recognize the lack of real response time experiments due to unavailability of suitable data. However, our LLM experiments demonstrate RtRank can learn from real preference data and real strength indicators (based on agreement). While RT's effectiveness as a strength indicator requires further validation in this context, prior psychological work supports this hypothesis. Importantly, RtRank can be used with any strength indicator, not just response times.

*Other questions:* We have responded to all the reviewer's individual questions (e.g., on preference pair normalization [**7tw8**], response time motivation [**jccL**], the anchor element [**LCxR**], and practical challenges [**WFhw**]), for which we refer to the individual responses.

**Conclusion**
Our new experiments strengthen the contribution and we are grateful for the reviewer's time and feedback. We are glad that reviewer jccL improved their assessment in response to our rebuttal (4), while LCxR and WFhw (initially 5) indicated satisfaction with our response. Although reviewer 7tw8 hasn't changed their borderline reject score, we hope that our new experimental results (shared in official rebuttal to jccL, then referenced to 7tw8) and elaborations have adequately addressed their concerns.

We will incorporate all feedback in the revision. Three of four reviewers now favor acceptance.

---

### Decision · Program_Chairs · 2025-09-17

**Decision:**

Accept (poster)

**Comment:**

This work proposes a new way that takes response time into account to model preference strength. To handle the noisy response time data, this paper introduces RtRank. RtRank leverages relative RT differences to rank pairwise comparisons by their inferred preference strength. The paper provides empirical validations for the proposed method.

The method offers a new perspective for evaluating the strength of a response, which is critical in preference alignment. The idea of using response time is reasonable and the authors also notice the noises therein. The reviewers initially concerns about the lack of experimental results with real-world human response time data, but in the rebuttal, the authors manage to provide some proxy results as a response.

Meanwhile, the current experiment results have room for improvement as public datasets usually do not perfectly capture response time. Overall, I believe this work brings an interesting direction for future study and it has the potential to further improve preference alignment.